# UNMASKING TRANSFORMERS: A THEORETICAL APPROACH TO DATA RECOVERY VIA ATTENTION WEIGHTS

## ABSTRACT

In the realm of deep learning, transformers have emerged as a dominant architecture, particularly in both natural language processing and computer vision tasks. However, with their widespread adoption, concerns regarding the security and privacy of the data processed by these models have arisen. In this paper, we address a pivotal question: Can the data fed into transformers be recovered using their attention weights and outputs? We introduce a theoretical framework to tackle this problem. Specifically, we present an algorithm that aims to recover the input data $X \in \mathbb{R}^{d \times n}$ from given attention weights $W = QK^\top \in \mathbb{R}^{d \times d}$ and output $B \in \mathbb{R}^{n \times n}$ by minimizing the loss function $L(X)$. This loss function captures the discrepancy between the expected output and the actual output of the transformer. Our findings have significant implications for preventing privacy leakage from attacking open-sourced model weights, suggesting potential vulnerabilities in the model's design from a security and privacy perspective - you may need only a few steps of training to force LLMs to tell their secrets.

## 1 INTRODUCTION

In the intricate and constantly evolving domain of deep learning, the transformer architecture has emerged as a game-changing innovation Vaswani et al. (2017). This novel architecture has propelled the state-of-the-art performance in a myriad of tasks, and its potency lies in the underlying mechanism known as the "attention mechanism". The essence of this mechanism can be distilled into its unique interaction between three distinct matrices: the **Query** ($Q$), the **Key** ($K$), and the **Value** ($V$), where the **Query** matrix ($Q$) represents the questions or the aspects we're interested in, the **Key** matrix ($K$) denotes the elements against which these questions are compared or matched, and the **Value** matrix ($V$) encapsulates the information we want to retrieve based on the comparisons. These matrices are not just mere multidimensional arrays; they play vital roles in encoding, comparing, and extracting pertinent information from the data.

Given this context, the attention mechanism can be mathematically captured as follows:

**Definition 1.1** (Attention matrix computation). *Let $Q, K \in \mathbb{R}^{n \times d}$ be two matrices that respectively represent the query and key. Similarly, for a matrix $V \in \mathbb{R}^{n \times d}$ denoting the value, the attention matrix is defined as*

$$\mathrm{Att}(Q, K, V) := D^{-1}AV,$$

*In this equation, two matrices are introduced: $A \in \mathbb{R}^{n \times n}$ and $D \in \mathbb{R}^{n \times n}$, defined as:*

$$A := \exp(QK^\top) \ \text{ and } \ D := \mathrm{diag}(A\mathbf{1}_n).$$

Here, the matrix $A$ represents the relationship scores between the query and key, and $D$ ensures normalization. The computation hence, deftly combines these relationships with the value matrix to output the final attended representation.

In practical large-scale language models ChatGPT (2022); OpenAI (2023), there might be multi-levels of the attention computation. For those multi-level architecture, the feed-forward training can be represented as

$$X_{\ell+1}^\top \leftarrow D(X_\ell)^{-1} \exp(X_\ell^\top Q_\ell K_\ell X_\ell) X_\ell^\top V_\ell$$

---

**Algorithm 1** Sketch of inverse attack to transformer-based models

---

**Input:** Ideal model prediction $B \in \mathbb{R}^{n \times d}$
**Parameters:** Model function $f$, pretrained weights $W$, training steps $T$
**Output:** Leaked input $X \in \mathbb{R}^{n \times d}$ for output $B$
**procedure** INVERSEATTACK($B, f, W, T$)
    Initialize each entry of $X_0 \in \mathbb{R}^{n \times d}$ from Gaussian distribution $\mathcal{N}(0,1)$.
    $t \leftarrow 1$
    **for** $t < T$ **do**
        Compute loss by some specific metric $\ell(\cdot, \cdot)$, such that $L_t := \ell(f(W, X_{t-1}), B)$
        Compute gradient $g_t := \nabla_{X_{t-1}} L_t$
        Compute update for $X$ via first-order or second order algorithm using $g_t$, denote $\Delta X$
        Update $X_t \leftarrow X_{t-1} - \Delta X$
        $t \leftarrow t + 1$
    **end for**
    **return** $X_T$ with guaranteed $L_t \leq \epsilon$ (Theorem 4.3 and Theorem 4.4)
**end procedure**

---

where $X_\ell$ is the input of $\ell$-th layer, and $X_{\ell+1}$ is the output of $\ell$-th layer, and $Q_\ell, K_\ell, V_\ell$ are the attention weights in $\ell$-th layer.

This architecture has particularly played a pivotal role in driving progress across various sub-disciplines of natural language processing (NLP) Firat et al. (2016); Choi et al. (2018); Usama et al. (2020); Naseem et al. (2020); Martin et al. (2019); ChatGPT (2022); OpenAI (2023). This trajectory of influence is most prominently embodied by the creation and widespread adoption of Large Language Models (LLMs) like GPT-4 and Claude-3. These models are hallmarks due to their staggering number of parameters and complex architectural designs.

Yet, the very complexity and architectural sophistication that propel the success of transformers come with a host of consequential challenges, making their effective and responsible usage nontrivial. Prominent among these challenges is the overarching imperative of ensuring data security and privacy Pan et al. (2020); Brown et al. (2022); Kandpal et al. (2022). Within the corridors of the research community, an increasingly pertinent question is emerging regarding the inherent vulnerabilities of these architectures. Specifically,

> *is it possible to know the input data by analyzing the attention weights and model outputs?*

To put it in mathematical terms, given a language model represented as $B = f(W; X)$, if one has access to the output $B$ and the attention weights $W$, is it possible to mathematically invert the model to obtain the original input data $X$?

Addressing this line of inquiry extends far beyond the realm of academic speculation; it has direct and significant implications for practical, real-world applications. This is especially true when these transformer models interact with data that is either sensitive in nature, like personal health records Cascella et al. (2023), or proprietary, as in the financial sector Wu et al. (2023). With the broader deployment of Large Language Models into environments that adhere to stringent data confidentiality regulations, the mandate for achieving data security becomes essential. In this work, we aim to delve deeply into this issue, striving to offer a nuanced understanding of these potential vulnerabilities while suggesting pathways for ensuring safety in the development, training, and utilization of transformer technologies.

This paper addresses a distinct *attention-based regression model* that differs from the conventional task of finding optimal weights for a given input and output. Specifically, we assume that the weights are already known, and our objective is to invert the output to recover the original data. The key focus of our investigation lies in *identifying the conditions* under which successful inversion of the original input is feasible. This problem holds significant relevance in the context of addressing security concerns associated with attention networks.

**Our contribution** In this paper, we formulate the formal regression model for the inverse attack on the soft-max attention layer. Utilizing simplified notations of the loss function, we are able to

calculate a close-form representation of its Hessian. By assuming bounded parameters and adding a moderate regularizer, we prove the smoothness (Lipschitz continuity) and strongly-convexity (Positive Semi-definiteness) of our regression problem, which leads to the convergence of gradient-based and Hessian-based methods that approach the approximate optimal. Therefore, we apply these algorithms to invert the attention weights to the input data. We provided numerical experiments to verify the reliability of our methods.

**Roadmap.** We arrange the rest of our paper as follows. In Section 2 we present some works related our topic. In Section 3, we state an overview of our techniques, summarizing the method we use to recover data via attention weights. We state our main theories in Section 4. We provide our experiment results in Section 5. We conclude our work in Section 6.

## 2 Related Works

This section discusses related works in the LLM community. We summarize the current research on LLM security and inversion attack in Section 2.1. We concern about attention computation theory and LLM-based regression theory in Section 2.2.

### 2.1 LLM Security

**Security concerns about LLM.** Amid LLM advancements, concerns about misuse have arisen Pan et al. (2020); Brown et al. (2022); Kandpal et al. (2022); Kirchenbauer et al. (2023); Vyas et al. (2023); Chu et al. (2023a); Xu et al. (2023); Gao et al. (2023d); Kirchenbauer et al. (2023); He et al. (2022a;b); Gao et al. (2023f); Shen et al. (2023a). Pan et al. (2020) assesses the privacy risks of capturing sensitive data with eight models and introduces defensive strategies, balancing performance and privacy. Brown et al. (2022) asserts that current methods fall short in guaranteeing comprehensive privacy for language models, recommending training on publicly intended text. Kandpal et al. (2022) reveals that the vulnerability of large language models to privacy attacks is significantly tied to data duplication in training sets, emphasizing that deduplicating this data greatly boosts their resistance to such breaches. Kirchenbauer et al. (2023) devised a way to watermark LLM output without compromising quality or accessing LLM internals. Meanwhile, Vyas et al. (2023) introduced near access-freeness (NAF), ensuring generative models, like transformers and image diffusion models, don't closely mimic copyrighted content by over $k$-bits.

**Inverting the neural network.** Originating from the explosion of deep learning, there have been a series of works focused on inverting the neural network Jensen et al. (1999); Lu et al. (1999); Mahendran & Vedaldi (2015); Dosovitskiy & Brox (2016); Zhang et al. (2020d). Jensen et al. (1999) surveys various techniques for neural network inversion, which involves finding input values that produce desired outputs, and highlights its applications in query-based learning, sonar performance analysis, power system security assessment, control, and codebook vector generation. Lu et al. (1999) presents a method for inverting trained neural networks by formulating the problem as a mathematical programming task, enabling various network inversions and enhancing generalization performance.. Mahendran & Vedaldi (2015) explores the reconstruction of image representations, including CNNs, to assess the extent to which it's possible to recreate the original image, revealing that certain layers in CNNs retain accurate visual information with varying degrees of geometric and photometric invariance. Zhang et al. (2020d) presents a novel generative model-inversion attack method that can effectively reverse deep neural networks, particularly in the context of face image reconstruction, and explores the connection between a model's predictive ability and vulnerability to such attacks while noting limitations in using differential privacy for defense.

**Attacking the Neural Networks.** During the development of artificial intelligence, there have been many works on attaching the neural networks Zhu et al. (2019); Wei et al. (2020); Rigaki & Garcia (2020); Huang et al. (2020); Yin et al. (2021); Huang et al. (2021b); Gao et al. (2023c). Several studies Zhu et al. (2019); Wei et al. (2020); Rigaki & Garcia (2020); Yin et al. (2021) have warned that local training data can be compromised using only exchanged gradient information. These methods start with dummy data and gradients, and through gradient descent, they empirically show that the original data can be fully reconstructed. A follow-up study Zhao et al. (2020) specifically focuses on classification tasks and finds that the real labels can also be accurately recovered. Other

types of attacks include membership and property inference Shokri et al. (2017); Melis et al. (2019), the use of Generative Adversarial Networks (GANs) Hitaj et al. (2017); Goodfellow et al. (2014), and additional machine-learning techniques McPherson et al. (2016); Papernot et al. (2016). A recent paper Wang et al. (2023) uses tensor decomposition for gradient leakage attacks but is limited by its inefficiency and focus on over-parametrized networks.

## 2.2 ATTENTION COMPUTATION AND REGRESSION

**Attention Computation Theory.**    Following the rise of LLM, numerous studies have emerged on attention computation Kitaev et al. (2020); Tay et al. (2020); Chen et al. (2021); Zandieh et al. (2023); Tarzanagh et al. (2023); Sanford et al. (2023); Panigrahi et al. (2023a); Zhang et al. (2020a); Arora & Goyal (2023); Tay et al. (2021); Deng et al. (2023b); Xia et al. (2023); Kacham et al. (2023). LSH techniques approximate attention, and based on them, the KDEformer offers a notable dot-product attention approximation Zandieh et al. (2023). Recent works Alman & Song (2023); Brand et al. (2023); Deng et al. (2023c) explored diverse attention computation methods and strategies to enhance model efficiency. On the optimization front, Zhang et al. (2020b) highlighted that adaptive methods excel over SGD due to heavy-tailed noise distributions. Other insights include the emergence of the KTIW property Snell et al. (2021) and various regression problems inspired by attention computation Gao et al. (2023a); Li et al. (2023c;b), revealing deeper nuances of attention models.

**Theoretical Approaches to Understanding LLMs.**    Recent strides have been made in understanding and optimizing regression models using various activation functions. Research on over-parameterized neural networks has examined exponential and hyperbolic activation functions for their convergence properties and computational efficiency Gao et al. (2023a); Li et al. (2023c); Deng et al. (2023b); Gao et al. (2023d); Li et al. (2023a); Gao et al. (2023e); Song et al. (2023); Sinha et al. (2023); Chu et al. (2023a;b); Shen et al. (2023b). Modifications such as regularization terms and algorithmic innovations, like a convergent approximation Newton method, have been introduced to enhance their performance Li et al. (2023c); Deng et al. (2022). Studies have also leveraged tensor tricks to vectorize regression models, allowing for advanced Lipschitz and time-complexity analyses Gao et al. (2023b); Deng et al. (2023a). Simultaneously, the field is seeing innovations in optimization algorithms tailored for LLMs. Techniques like block gradient estimators have been employed for huge-scale optimization problems, significantly reducing computational complexity Cai et al. (2021). Unique approaches like Direct Preference Optimization bypass the need for reward models, fine-tuning LLMs based on human preference data Rafailov et al. (2023). Additionally, advancements in second-order optimizers have relaxed the conventional Lipschitz Hessian assumptions, providing more flexibility in convergence proofs Liu et al. (2023). Also, there is a series of work on understanding fine-tuning Malladi et al. (2023a;b); Panigrahi et al. (2023b). Collectively, these theoretical contributions are refining our understanding and optimization of LLMs, even as they introduce new techniques to address challenges such as non-guaranteed Hessian Lipschitz conditions.

**Optimization and Convergence of Deep Neural Networks.**    Prior research Li & Liang (2018); Du et al. (2018); Allen-Zhu et al. (2019a;b); Arora et al. (2019a;b); Song & Yang (2019); Cai et al. (2019); Zhang et al. (2019); Cao & Gu (2019); Zou & Gu (2019); Oymak & Soltanolkotabi (2020); Ji & Telgarsky (2019); Lee et al. (2020); Huang et al. (2021a); Zhang et al. (2020c); Brand et al. (2020); Zhang et al. (2020a); Song et al. (2021); Alman et al. (2023); Munteanu et al. (2022); Zhang (2022); Gao et al. (2023a); Li et al. (2023c); Qin et al. (2023) on the optimization and convergence of deep neural networks has been crucial in understanding their exceptional performance across various tasks. These studies have also contributed to enhancing the safety and efficiency of AI systems. In Gao et al. (2023a) they define a neural function using an exponential activation function and apply the gradient descent algorithm to find optimal weights. In Li et al. (2023c), they focus on the exponential regression problem inspired by the attention mechanism in large language models. They address the non-convex nature of standard exponential regression by considering a regularization version that is convex. They propose an algorithm that leverages input sparsity to achieve efficient computation. The algorithm has a logarithmic number of iterations and requires nearly linear time per iteration, making use of the sparsity of the input matrix.

## 3 RECOVERING DATA VIA ATTENTION WEIGHTS

In this section, we propose our theoretical method to recover the training data from trained transformer weights and outputs. In Section 3.1, we provide a detailed description of our approach. In Section 3.2, we introduce our simplified notations to calculate the Hessian of the loss function. In Section 3.3, we state the decomposed expression of the Hessian.

### 3.1 TRAINING OBJECTIVE OF ATTENTION INVERSION ATTACK

In this study, we propose a novel technique for inverting the attention weights of a transformer model using Hessian-based algorithms. We consider the single-layer soft-max attention function

$$f(W; X) := D(X)^{-1} \exp(X^\top W X) V$$

, where $W = KQ^\top \in \mathbb{R}^{d \times d}$ represents the attention weights and $D(X) = \mathrm{diag}(\exp(X^\top W X)) \in \mathbb{R}^{n \times n}$ is the diagonal matrix for normalization.

Our aim is to find the input $X \in \mathbb{R}^{d \times n}$ that minimizes the Frobenius norm of the difference between $f(W; X)$ and the output $B$. Here, dimension $d$ denotes the length of a token, dimension $n$ denotes the total number of the tokens in $X$. To achieve this, we introduce an algorithm that minimizes the loss function $L(X)$, defined as follows:

**Definition 3.1** (Regression model). *Given the attention weights $W = KQ^\top \in \mathbb{R}^{d \times d}$, $V \in \mathbb{R}^{d \times d}$ and output $B \in \mathbb{R}^{n \times d}$, the goal is find $X \in \mathbb{R}^{d \times n}$ such that*

$$L(X) := \|D(X)^{-1} \exp(X^\top W X) X^\top V - B\|_F^2 + L_{\mathrm{reg}}, \tag{1}$$

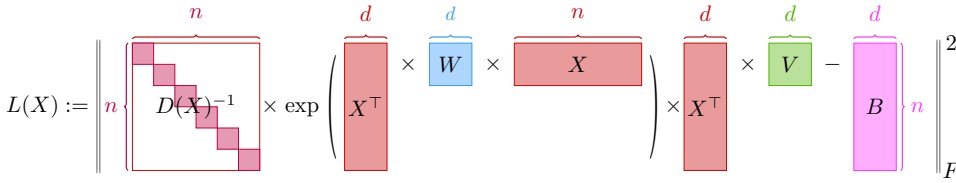

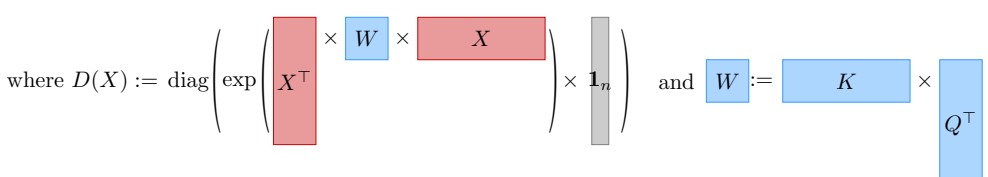

Figure 1: Visualization of our loss function.

$L_{\mathrm{reg}}$ captures the additional regularization terms which we introduce later. This loss function quantifies the discrepancy between the expected output and the actual output of the transformer.

In our approach, we leverage Hessian decomposition to efficiently compute the Hessian matrix and apply a second-order method to approximate the optimal input $X$. Utilizing the Hessian, we can gain insights into the curvature of the loss function, which improves the efficiency of finding the approximate optimal solution.

By integrating Hessian decomposition and second-order optimization techniques (Anstreicher (2000); Lee et al. (2019); Cohen et al. (2019); Jiang et al. (2021); Huang et al. (2022); Gu & Song (2022); Gu et al. (2023)), our proposed algorithm provides a promising approach for addressing the challenging task of inverting attention weights in transformer models.

### 3.2 MODEL SIMPLIFICATION

Due to the complexity of the loss function (Eq. (1)), it is challenging to give the explicit formula of its Hessian. To simplify the computation, we introduce several notations (See Figure 2 for visualization):

$$\text{Exponential Function: } u(X)_i := \exp(X^\top W X_{*,i})$$

$$\text{Sum of Softmax: } \alpha(X)_i := \langle u(X)_i, \mathbf{1}_n \rangle$$

$$\text{Softmax Probability: } f(X)_i := \alpha(X)_i^{-1} u(X)_i$$

$$\text{Value Function: } h(X)_j := X^\top V_{*,j}$$

$$\text{One-entry Loss Function: } c(X)_{i,j} := \langle f(X)_i, h(X)_j \rangle - b_{i,j}.$$

(a) Exponential Function    (b) Sum of Softmax    (c) Softmax Probability

(d) Value Function    (e) One-unit Loss Function

Figure 2: Visualization of Notations We Defined

Using these terms, we can express the loss function $L(X)$ as the sum over the loss in each entry as below, which allows us to break down the computation into several steps. $L(X) = \sum_{i=1}^{n} \sum_{j=1}^{d} (c(X)_{i,j})^2$.

### 3.3 HESSIAN DECOMPOSITION

This section provides our technique to decompose the Hessian. By decomposing the Hessian into several cases, we can give a close-form expression, which enables us to comprehend and analyze the Hessian. We use variables $i_k \in [n], j_k \in [d], k = 1, 2, 3$ to denote the indexes.

Now, we split $\frac{\mathrm{d}c(X)_{i_0,j_0}}{\mathrm{d}x_{i_1,j_1}}$ (the gradient of $c(X)_{i_0,j_0}$) into two cases:

- *Case 1:* The situation when $i_0 = i_1$.
- *Case 2:* The situation when $i_0 \neq i_1$.

Similar, we break down the computation of $\frac{\mathrm{d}^2 c(X)_{i_0,j_0}}{\mathrm{d}x_{i_1,j_1}\mathrm{d}x_{i_2,j_2}}$ into five cases to handle different scenarios:

- *Case 1:* The situation when $i_0 = i_1 = i_2$.
- *Case 2:* The situation when $i_0 = i_1 \neq i_2$.
- *Case 3:* The situation when $i_0 = i_2 \neq i_1$.
- *Case 4:* The situation when $i_0 \neq i_1$, $i_0 \neq i_2$ and $i_1 = i_2$.
- *Case 5:* The situation when $i_0 \neq i_1$, $i_0 \neq i_2$ and $i_1 \neq i_2$.

It is worth mentioning that the second case and the third case are equivalent by switching indexes. By considering these cases, we can calculate the Hessian for each element in $X$. This allows us to gain further insights into the curvature of the loss function and optimize the parameters more effectively.

Since our decision variable $X$ is a $n \times d$ matrix, we define the Hessian of $c(X)_{i_0,j_0}$ by considering its Hessian with respect to $x = \mathrm{vec}(X)$. This means that, $\nabla^2 c(X)_{i_0,j_0}$ is a $nd \times nd$ matrix with its $i_1 \cdot j_1, i_2 \cdot j_2$-th entry being $\frac{\mathrm{d}c(X)_{i_0,j_0}}{\mathrm{d}x_{i_1,j_2}x_{i_2,j_2}}$. Leveraging the split of different scenarios, we decompose the Hessian into a partition of square matrices.

**Definition 3.2** (Hessian split). *We use $H_k^{(i_1,i_2)} \in \mathbb{R}^{d \times d}$ to represent the square matrix corresponding to the $k$-th case in Hessian computation. Notice that the $j_1, j_2$-th entry of $H_k^{(i_1,i_2)}$ is $\frac{\mathrm{d}c(X)_{i_0,j_0}}{\mathrm{d}x_{i_1,j_2}x_{i_2,j_2}}$. Then, the Hessian of the loss is a matrix partition consists of matrices of the above five cases. The formal representation can be found in Appendix D.1.*

The reason we introduce the Hessian split is that the square matrices of the same type share the similar formula. Therefore, we can compute the expression of each type (see detailed calculation in Section D) to derive $\frac{\mathrm{d}c(X)_{i_0,j_0}}{\mathrm{d}x_{i_1,j_2}x_{i_2,j_2}}$. This gives us the information of the Hessian of the loss function.

# 4    MAIN RESULTS

Now, we state the analysis of the correctness of our inversion attack strategy. Assuming the parameters are bounded, we verify the Hession of our loss function is Lipschitz continuous and PSD lower-bounded. Therefore, gradient-based and Hessian-based methods are used to solve the regularized regression model. We defer the proofs to the Appendix.

**Properties of the Hessian**    We assume an unified upper bound for all parameters in our model, including the weight $W$, the value $V$, the output $B$, and the decision variable $X$.

**Assumption 4.1** (Bounded Parameters, Informal version of Assumption F.1). *We assume $\|W\| \leq R, \|V\| \leq R, \|X\| \leq R, b_{i,j} \leq R^2$, where $\|\cdot\|$ is the matrix 2-norm and $R > 1$ is some constant.*

Next, we state the bounds for the Hessian of the loss function in terms of $\mathrm{poly}(n, d, R)$.

**Theorem 4.2** (Properties of the Hessian, Informal version of Theorem G.12 and Theorem H.2). *We assume that Assumption 4.1 holds. Then, the Hessian of $L(X)$ is Lipschitz continuous with Lipschitz constant being $O(n^{3.5}d^{3.5}R^{10})$. Also, it has PSD lower bound: $L(X) \succeq -O(ndR^8) \cdot \mathbf{I}_{nd}$.*

Therefore, we define the regularization term to be $L_{\mathrm{reg}} := O(ndR^8) \cdot \|\mathrm{vec}(X)\|_2^2$ to have the PSD guarantee for our regression problem.

**Convergence analysis**    With above properties of the loss function, we have the convergence results stated as follows. Theorem 4.3 shows the correctness of the gradient-based method. Theorem 4.3 shows the correctness of the Hessian-based method. The algorithm for approximating PSD matrices in Deng et al. (2022) can be applied to approximate the Hessian efficiently.

**Theorem 4.3** (First-Order Main Result, Informal version of Theorem I.2). *We assume that Assumption 4.1 holds. Let $X^*$ denote the optimal point of the regularized regression model defined in Definition 3.1. Then, for any accuracy parameter $\epsilon \in (0, 0.1)$, an algorithm based on the gradient-descent method can be employed to recover the initial data. It outputs a matrix $\widetilde{X} \in \mathbb{R}^{d \times n}$ satisfying $\|\widetilde{X} - X^*\|_F \leq \epsilon$. The algorithm runs $T = O(\mathrm{poly}(n, d, R) \cdot \log(\|X_0 - X^*\|_F/\epsilon))$ iterations, with execution time for each iteration being $\mathrm{poly}(n, d)$, where the degree of $d$ depends on the current matrix computation time.*

**Theorem 4.4** (Second-Order Main Result, Informal version of Theorem I.3). *We assume that Assumption 4.1 holds. Let $X^*$ denote the optimal point of the regularized regression model defined in Definition 3.1. Suppose we choose an initial point $X_0$ such that $M \cdot \|X_0 - X^*\|_F \leq O(ndR^8)$ where $M = O(n^3d^3R^{10})$. Then, for any accuracy parameter $\epsilon \in (0, 0.1)$ and any failure probability $\delta \in (0, 0.1)$, an algorithm based on the approximation-Newton method can be employed to recover the initial data. It outputs a matrix $\widetilde{X} \in \mathbb{R}^{d \times n}$ satisfying $\|\widetilde{X} - X^*\|_F \leq \epsilon$ with a probability at least $1 - \delta$. The algorithm runs $T = O(\log(|X_0 - X^*|_F/\epsilon))$ iterations, with execution time for each iteration being $\mathrm{poly}(n, d, \log(1/\delta))$, where the degree of $d$ depends on the current matrix computation time.*

These theorems show that we can utilize first-order method and second-order method to search an $\epsilon$-optimal approximation to the real input data $X$ within a preferable running time.

| step | recovering text | loss |
|------|-----------------|------|
| 0 | GrapeJUST once received cancer treatment at this hospital. | 4.74 |
| 2500 | precious quoted once received cancer treatment at this hospital. | 4.61 |
| 5000 | grass Tradable once received cancer treatment at this hospital. | 4.50 |
| 6500 | acrylic Bob once received cancer treatment at this hospital. | 4.29 |
| 7500 | Alan Bob once received cancer treatment at this hospital. | 2.27 |

Table 1: Visualization of the results. Here, the original target text is **Alan Bob once received cancer treatment at this hospital.** We mask the sensitive data **Alan Bob** and run the gradient-descent inverse attack to recover. The blue-colored texts are the outputs in each iteration. The column on the right shows the value of the cross-entropy loss. It can be seen that the original data is leaked after 7500 steps, which echoes our convergence analysis.

## 5 EXPERIMENT

In this section, to verify the accuracy of our theory, we conducted a simple experiment to evaluate how our approach recovers data from the pre-trained weights in the LLM. In Section 5.1, we provide the setup and the design of our data-attack experiment. Next, we discuss our results in Section 5.2. Supplementary experimental details are provided in Appendix J.

### 5.1 EXPERIMENT DESIGN AND SETUP

We use the pre-trained language model GPT-2-small Radford et al. (2019). For the dataset, we utilize GPT-4 Achiam et al. (2023); Bubeck et al. (2023) to help us create hundreds of text data containing virtual information. This can be viewed as the toy or the synthetic dataset. Then, we use the synthetic dataset to fine-tune the pre-trained GPT-2-small with Adam optimizer Kingma & Ba (2014).

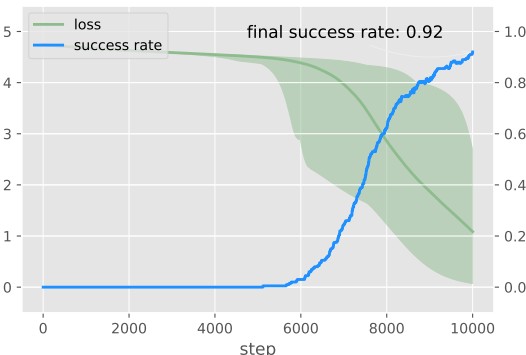

Figure 3: Training record of our inversion recovery attack. We state the maximum, mean, and minimum loss during 10000 updates. We also present the success rate of attack in 1000 repeated experiments.

For the recovery part, we first choose one text from the dataset and convert it into one-hot vectors through the model's vocabulary, denoted by $S^* \in \mathbb{R}^{n \times N}$ where $N$ is the vocabulary size. Notice that GPT-2-small is trained to conduct next-token prediction by causal mask, namely, it uses the information of the first $k$ words to predict the $(k + 1)$-th word. Therefore, we split $S^*$ to the masked part $S_1 \in \mathbb{R}^{m \times N}$ and the unmasked part $S_2 \in \mathbb{R}^{(n-m) \times N}$. Then, we use $S_2$ as part of the initial input and we introduce our inversion attack approach to recover $S_1$.

We initialize our recovery by a random matrix $X^0 \in \mathbb{R}^{m \times N}$ where each entry is sampled from $\mathcal{N}(0, 1)$. We compute $S_1^0 \in \mathbb{R}^{m \times N} := \mathsf{softmax}(X^0)$, and concatenate it with $S_2$ to form $S^0 \in \mathbb{R}^{n \times N} = \begin{bmatrix} S_1^0 \\ S_2^0 \end{bmatrix}$, then input it into the model. We denote the GPT-2-small model by a mapping $F : \mathbb{R}^{n \times N} \to \mathbb{R}^{n \times N}$. For any input matrix $A \in \mathbb{R}^{n \times N}$, the output of GPT-2-small $F(A) \in \mathbb{R}^{n \times N}$ will consist of row-wise soft-max vectors since we add a soft-max operation to the output of the last

layer to compute the probability distribution. We use $S^t \in \mathbb{R}^{n \times N}$ to represent the matrix of soft-max vectors we recover at the $t$-th timestamp for integer $t \geq 0$ by minimizing the loss.

We define our problem as minimizing the cross-entropy loss which is calculated as $L(F(S^t), S^t) := \sum_{i=1}^{n-1} \sum_{j=1}^{N} -S_{i+1,j}^t \cdot \log(F(S^t)_{i,j})$.

**Remark 5.1.** *We use the cross-entropy loss here instead since it is commonly used in the training of current LLMs. Note that our approach to analyze the canonical softmax loss regression can be modified to show the correctness of the cross-entropy loss regression. Similar topics have been discussed in other LLM-related literature, e.g. Gao et al. (2023c).*

We use the gradient-descent method to conduct the attack. The update rule is defined as:

$$X_{t+1} \leftarrow X_t - \eta \nabla_{X_t} L(F(S^t), S^t),$$

where we use $X_t$ to denote the recovering input at $t$-th timestamp for integer $t \geq 0$. Note that $\eta$ denotes the learning rate. $S^t$ is computed by $X_t$ as we mentioned above.

The training involves Adam optimizer, and all the hyper-parameters are set to be defaults. Totally, we trained 10000 steps for the input recovery. All the experiments are repeated 1000 times to ensure reliability.

## 5.2 RESULTS

We state our results of recovery in Figure 3. We recorded the mean, maximum, and minimum loss during the training. We also recorded the success rate at each stage in the 10000 updates. Notice that the success rate at the $k$-th update is computed by the count of successful experiments (i.e., the masked input data is recovered) at the $k$-th update divided by 1000, which is the repeated time. It's noteworthy that after 5000 steps, the success rate greatly increases, eventually, it demonstrates a high value of 0.92. This result verifies our attacking method has a high probability of recovery training, especially for private and sensitive data from open-source weights of language models.

Furthermore, we showcase one example of the recovery attacks in Table 1, where we create fake data "Alan Bob once received cancer treatment at this hospital.". Accordingly, the name "Alan Bob" in the context is private and masked. We cut these two words and converted the sentence " once received cancer treatment at this hospital." into one-hot vectors as $S_2$ in Section 5.1. Next, we run the inverse attack and record the output and loss value at each step. We use blue text to represent the text that is predicted by our algorithm. As we can see from Table 1, the recovering text is initially GrapeJUST with the cross-entropy loss 4.74 at the beginning. Then, at the 6500-th step of recovering, our algorithm outputs acrylic Bob, where the word "Bob" is successfully recovered. Finally, at the 7500-th step, our algorithm successfully recovers the target text Alan Bob.

## 6 CONCLUSION

In this study, we have presented a theoretical approach for conducting the inverse recovery on the input data using weights and outputs.

We propose the mathematical framework of the attention-inspired mechanism regression model. Our theoretical analysis part consists of the efficient calculation of the Hessian and the verification of its smoothness and strongly-convexity. With the aim of these properties, we introduce gradient-based and Hessian-based to do the inverse recovery. Then, we show the reliability of our proposed method by experiments on text reconstruction using GPT-2-small.

The insights gained from this research are intended to deepen our understanding and facilitate the development of more secure and robust transformer models. By doing so, we strive to foster responsible and ethical advancements in the field of deep learning. This work lays the groundwork for future research and development aimed at fortifying transformer technologies against potential threats and vulnerabilities.

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

**Roadmap.**    We arrange the appendix as follows. In Section A we provide details of computing the gradients. In Section B and Section C we provide detail of computing Hessian for two cases. In Section D we show how to split the Hessian matrix. In Section E we combine the results before and compute the Hessian for the loss function. In Section F we bound the basic functions to be used later. In Section G we provide proof for the Lipschitz property of the Hessian of the loss function. In Section H, we provide the proof for the PSD bound of the Hessian. In Section I, we provide the convergence analysis for our proposed methods. In Section J, we provide additional details for our experiment.

## A    GRADIENTS

Here in this section, we provide analysis for the gradient computation. In Section A.1 we state some facts to be used. In Section A.2 we provide some definitions. In Sections A.3, A.4, A.5, A.6, A.7, A.8 and A.9 we compute the gradient for the terms defined respectively. Finally in Section A.10 we compute the gradient for $L(X)$.

### A.1    FACTS

**Fact A.1** (Basic algebra).  *We have*

- $\langle u, v \rangle = \langle v, u \rangle = u^\top v = v^\top u$.

- $\langle u \circ v, w \rangle = \langle u \circ v \circ w, \mathbf{1}_n \rangle$

- $u^\top (v \circ w) = u^\top \operatorname{diag}(v) w$

**Fact A.2** (Basic calculus rule).  *We have*

- $\frac{\mathrm{d}\langle f(x), g(x)\rangle}{\mathrm{d}t} = \langle \frac{\mathrm{d}f(x)}{\mathrm{d}t}, g(x)\rangle + \langle f(x), \frac{\mathrm{d}g(x)}{\mathrm{d}t}\rangle$ *(here t can be any variable)*

- $\frac{\mathrm{d}y^z}{\mathrm{d}x} = z \cdot y^{z-1} \frac{\mathrm{d}y}{\mathrm{d}x}$

- $u \cdot v = v \cdot u$

- $\frac{\mathrm{d}x}{\mathrm{d}x_j} = e_j$ *where $e_j$ is a vector that only $j$-th entry is 1 and zero everywhere else.*

- *Let $x \in \mathbb{R}^d$, let $y \in \mathbb{R}$ be independent of $x$, we have $\frac{\mathrm{d}x}{\mathrm{d}y} = \mathbf{0}_d$.*

- *Let $f(x), g(x) \in \mathbb{R}$, we have $\frac{\mathrm{d}(f(x)g(x))}{\mathrm{d}t} = \frac{\mathrm{d}f(x)}{\mathrm{d}t} g(x) + f(x) \frac{\mathrm{d}g(x)}{\mathrm{d}t}$*

- *Let $x \in \mathbb{R}$, $\frac{\mathrm{d}}{\mathrm{d}x} \exp(x) = \exp(x)$*

- *Let $f(x) \in \mathbb{R}^n$, we have $\frac{\mathrm{d}\exp(f(x))}{\mathrm{d}t} = \exp(f(x)) \circ \frac{\mathrm{d}f(x)}{\mathrm{d}t}$*

### A.2    DEFINITIONS

**Definition A.3** (Simplified notations).  *We have following definitions*

- *We use $u(X)_{i_0, i_1}$ to denote the $i_1$-th entry of $u(X)_{i_0}$.*

- *We use $f(X)_{i_0, i_1}$ to denote the $i_1$-th entry of $f(X)_{i_0}$.*

- *We define $W_{j_1, *}$ to denote the $j_1$-th row of $W$. (In the proof, we treat $W_{j_1, *}$ as a column vector).*

- *We define $W_{*, j_1}$ to denote the $j_1$-th column of $W$.*

- *We define $w_{j_1, j_0}$ to denote the scalar equals to the entry in $j_1$-th row, $j_0$-th column of $W$.*

- *We define $V_{*, j_1}$ to denote the $j_1$-th column of $V$.*

- *We define $v_{j_1, j_0}$ to denote the scalar equals to the entry in $j_1$-th row, $j_0$-th column of $V$.*

- *We define $X_{*,i_0}$ to denote the $i_0$-th column of $X$.*

- *We define $x_{i_1,j_1}$ to denote the scalar equals to the entry in $i_1$-**th column**, $j_1$-**th row** of $X$.*

**Definition A.4** (Exponential function $u$). *If the following conditions hold*

- *Let $X \in \mathbb{R}^{d \times n}$*

- *Let $W \in \mathbb{R}^{d \times d}$*

*For each $i_0 \in [n]$, we define $u(X)_{i_0} \in \mathbb{R}^n$ as follows*

$$u(X)_{i_0} = \exp(X^\top W X_{*,i_0})$$

**Definition A.5** (Sum function of softmax $\alpha$). *If the following conditions hold*

- *Let $X \in \mathbb{R}^{d \times n}$*

- *Let $u(X)_{i_0}$ be defined as Definition A.4*

*We define $\alpha(X)_{i_0} \in \mathbb{R}$ for all $i_0 \in [n]$ as follows*

$$\alpha(X)_{i_0} = \langle u(X)_{i_0}, \mathbf{1}_n \rangle$$

**Definition A.6** (Softmax probability function $f$). *If the following conditions hold*

- *Let $X \in \mathbb{R}^{d \times n}$*

- *Let $u(X)_{i_0}$ be defined as Definition A.4*

- *Let $\alpha(X)_{i_0}$ be defined as Definition A.5*

*We define $f(X)_{i_0} \in \mathbb{R}^n$ for each $i_0 \in [n]$ as follows*

$$f(X)_{i_0} := \alpha(X)_{i_0}^{-1} u(X)_{i_0}$$

**Definition A.7** (Value function $h$). *If the following conditions hold*

- *Let $X \in \mathbb{R}^{d \times n}$*

- *Let $V \in \mathbb{R}^{d \times d}$*

*We define $h(X)_{j_0} \in \mathbb{R}^n$ for each $j_0 \in [n]$ as follows*

$$h(X)_{j_0} := X^\top V_{*,j_0}$$

**Definition A.8** (One-unit loss function $c$). *If the following conditions hold*

- *Let $f(X)_{i_0}$ be defined as Definition A.6*

- *Let $h(X)_{j_0}$ be defined as Definition A.7*

*We define $c(X) \in \mathbb{R}^{n \times d}$ as follows*

$$c(X)_{i_0,j_0} := \langle f(X)_{i_0}, h(X)_{j_0} \rangle - b_{i_0,j_0}, \forall i_0 \in [n], j_0 \in [d]$$

**Definition A.9** (Overall function $L$). *If the following conditions hold*

- *Let $c(X)_{i_0,j_0}$ be defined as Definition A.8*

*We define $L(X) \in \mathbb{R}$ as follows*

$$L(X) := \sum_{i_0=1}^{n} \sum_{j_0=1}^{d} (c(X)_{i_0,j_0})^2$$

## A.3   Gradient for each column of $X^\top W X_{*,i_0}$

**Lemma A.10.** *We have*

- **Part 1.** *Let $i_0 = i_1 \in [n]$, $j_1 \in [d]$*

$$\underbrace{\frac{\mathrm{d} X^\top W X_{*,i_0}}{\mathrm{d} x_{i_1,j_1}}}_{n \times 1} = \underbrace{e_{i_0}}_{n \times 1} \cdot \underbrace{\langle W_{j_1,*}, X_{*,i_0} \rangle}_{\text{scalar}} + \underbrace{X^\top}_{n \times d} \underbrace{W_{*,j_1}}_{d \times 1}$$

- **Part 2** *Let $i_0 \neq i_1 \in [n]$, $j_1 \in [d]$*

$$\underbrace{\frac{\mathrm{d} X^\top W X_{*,i_0}}{\mathrm{d} x_{i_1,j_1}}}_{n \times 1} = \underbrace{e_{i_1}}_{n \times 1} \cdot \underbrace{\langle W_{j_1,*}, X_{*,i_0} \rangle}_{\text{scalar}}$$

*Proof.* **Proof of Part 1.**

$$\underbrace{\frac{\mathrm{d} X^\top W X_{*,i_0}}{\mathrm{d} x_{i_1,j_1}}}_{n \times 1} = \underbrace{\frac{\mathrm{d} X^\top}{\mathrm{d} X_{i_1,j_1}}}_{n \times d} \underbrace{W}_{d \times d} \underbrace{X_{*,i_0}}_{d \times 1} + \underbrace{X^\top}_{n \times d} \underbrace{W}_{d \times d} \underbrace{\frac{\mathrm{d} X_{*,i_0}}{\mathrm{d} X_{i_1,j_1}}}_{d \times 1}$$

$$= \underbrace{e_{i_1}}_{n \times 1} \underbrace{e_{j_1}^\top}_{1 \times d} \underbrace{W}_{d \times d} \underbrace{X_{*,i_0}}_{d \times 1} + \underbrace{X^\top}_{n \times d} \underbrace{W}_{d \times d} \underbrace{e_{j_1}}_{d \times 1}$$

$$= \underbrace{e_{i_1}}_{n \times 1} \cdot \underbrace{\langle W_{j_1,*}, X_{*,i_0} \rangle}_{\text{scalar}} + \underbrace{X^\top}_{n \times d} \underbrace{W_{*,j_1}}_{d \times 1}$$

$$= \underbrace{e_{i_0}}_{n \times 1} \cdot \underbrace{\langle W_{j_1,*}, X_{*,i_0} \rangle}_{\text{scalar}} + \underbrace{X^\top}_{n \times d} \underbrace{W_{*,j_1}}_{d \times 1}$$

where the 1st step follows from Fact A.2, the 2nd step follows from simple derivative rule, the 3rd is simple algebra, the 4th step ie because $i_0 = i_1$.

**Proof of Part 2**

$$\underbrace{\frac{\mathrm{d} X^\top W X_{*,i_0}}{\mathrm{d} x_{i_1,j_1}}}_{n \times 1} = \underbrace{\frac{\mathrm{d} X^\top}{\mathrm{d} x_{i_1,j_1}}}_{n \times d} \underbrace{W}_{d \times d} \underbrace{X_{*,i_0}}_{d \times 1} + \underbrace{X^\top}_{n \times d} \underbrace{W}_{d \times d} \underbrace{\frac{\mathrm{d} X_{*,i_0}}{\mathrm{d} x_{i_1,j_1}}}_{d \times 1}$$

$$= \underbrace{e_{i_1}}_{n \times 1} \underbrace{e_{j_1}^\top}_{1 \times d} \underbrace{W}_{d \times d} \underbrace{X_{*,i_0}}_{d \times 1} + \underbrace{X^\top}_{n \times d} \underbrace{W}_{d \times d} \underbrace{\mathbf{0}_d}_{d \times 1}$$

$$= \underbrace{e_{i_1}}_{n \times 1} \cdot \underbrace{\langle W_{j_1,*}, X_{*,i_0} \rangle}_{\text{scalar}}$$

where the 1st step follows from Fact A.2, the 2nd step follows from simple derivative rule, the 3rd is simple algebra. $\square$

## A.4   Gradient for $u(X)_{i_0}$

**Lemma A.11.** *Under following conditions*

- *Let $u(X)_{i_0}$ be defined as Definition A.4*

*We have*

- **Part 1.** *For each $i_0 = i_1 \in [n]$, $j_1 \in [d]$*

$$\underbrace{\frac{\mathrm{d} u(X)_{i_0}}{\mathrm{d} x_{i_1,j_1}}}_{n \times 1} = u(X)_{i_0} \circ (e_{i_0} \cdot \langle W_{j_1,*}, X_{*,i_0} \rangle + X^\top W_{*,j_1})$$

- **Part 2** *For each $i_0 \neq i_1 \in [n]$, $j_1 \in [d]$*

$$\underbrace{\frac{\mathrm{d}u(X)_{i_0}}{\mathrm{d}x_{i_1,j_1}}}_{n \times 1} = \underbrace{u(X)_{i_0}}_{n \times 1} \circ (e_{i_1} \cdot \langle W_{j_1,*}, X_{*,i_0} \rangle)$$

*Proof.*

**Proof of Part 1**

$$\underbrace{\frac{\mathrm{d}u(X)_{i_0}}{\mathrm{d}x_{i_1,j_1}}}_{n \times 1} = \underbrace{\frac{\mathrm{d}\exp(X^\top W X_{*,i_0})}{\mathrm{d}x_{i_1,j_1}}}_{n \times 1}$$

$$= \exp(\underbrace{X^\top}_{n \times d} \underbrace{W}_{d \times d} \underbrace{X_{*,i_0}}_{d \times 1}) \circ \underbrace{\frac{\mathrm{d}X^\top W X_{*,i_0}}{\mathrm{d}x_{i_1,j_1}}}_{n \times 1}$$

$$= \underbrace{u(X)_{i_0}}_{n \times 1} \circ \underbrace{\frac{\mathrm{d}X^\top W X_{*,i_0}}{\mathrm{d}x_{i_1,j_1}}}_{n \times 1}$$

$$= \underbrace{u(X)_{i_0}}_{n \times 1} \circ (\underbrace{e_{i_0}}_{n \times 1} \cdot \underbrace{\langle W_{j_1,*}, X_{*,i_0} \rangle}_{\text{scalar}} + \underbrace{X^\top}_{n \times d} \underbrace{W_{*,j_1}}_{d \times 1})$$

where the 1st step and the 3rd step follow from Definition of $u(X)_{i_0}$ (see Definition A.4), the 2nd step follows from Fact A.2, the 4th step follows by Lemma A.10.

**Proof of Part 2**

$$\underbrace{\frac{\mathrm{d}u(X)_{i_0}}{\mathrm{d}x_{i_1,j_1}}}_{n \times 1} = \underbrace{\frac{\mathrm{d}\exp(X^\top W X_{*,i_0})}{\mathrm{d}x_{i_1,j_1}}}_{n \times 1}$$

$$= \exp(\underbrace{X^\top}_{n \times d} \underbrace{W}_{d \times d} \underbrace{X_{*,i_0}}_{d \times 1}) \circ \underbrace{\frac{\mathrm{d}X^\top W X_{*,i_0}}{\mathrm{d}x_{i_1,j_1}}}_{n \times 1}$$

$$= \underbrace{u(X)_{i_0}}_{n \times 1} \circ \underbrace{\frac{\mathrm{d}X^\top W X_{*,i_0}}{\mathrm{d}x_{i_1,j_1}}}_{n \times 1}$$

$$= \underbrace{u(X)_{i_0}}_{n \times 1} \circ (\underbrace{e_{i_1}}_{n \times 1} \cdot \underbrace{\langle W_{j_1,*}, X_{*,i_0} \rangle}_{\text{scalar}})$$

where the 1st step and the 3rd step follow from Definition of $u(X)_{i_0}$ (see Definition A.4), the 2nd step follows from Fact A.2, the 4th step follows by Lemma A.10.

$\square$

### A.5 GRADIENT COMPUTATION FOR $\alpha(X)_{i_0}$

**Lemma A.12** (A generalization of Lemma 5.6 in Deng et al. (2023b))**.** *If the following conditions hold*

- *Let $\alpha(X)_{i_0}$ be defined as Definition A.5*

*Then, we have*

- **Part 1.** *For each $i_0 = i_1 \in [n]$, $j_1 \in [d]$*

$$\underbrace{\frac{\mathrm{d}\alpha(X)_{i_0}}{\mathrm{d}x_{i_1,j_1}}}_{\text{scalar}} = u(X)_{i_0,i_0} \cdot \langle W_{j_1,*}, X_{*,i_0} \rangle + \langle u(X)_{i_0}, X^\top W_{*,j_1} \rangle$$

- **Part 2.** *For each $i_0 \neq i_1 \in [n]$, $j_1 \in [d]$*

$$\underbrace{\frac{d\alpha(X)_{i_0}}{dx_{i_1,j_1}}}_{\text{scalar}} = u(X)_{i_0,i_1} \cdot \langle W_{j_1,*}, X_{*,i_0} \rangle$$

*Proof.* **Proof of Part 1.**

$$\underbrace{\frac{d\alpha(X)_{i_0}}{dx_{i_1,j_1}}}_{\text{scalar}} = \underbrace{\frac{d\langle u(X)_{i_0}, \mathbf{1}_n \rangle}{dx_{i_1,j_1}}}_{\text{scalar}}$$

$$= \langle \underbrace{\frac{du(X)_{i_0}}{dx_{i_1,j_1}}}_{n \times 1}, \underbrace{\mathbf{1}_n}_{n \times 1} \rangle$$

$$= \langle \underbrace{u(X)_{i_0}}_{n \times 1} \circ (e_{i_0} \cdot \langle W_{j_1,*}, X_{*,i_0} \rangle + X^\top W_{*,j_1}), \underbrace{\mathbf{1}_n}_{n \times 1} \rangle$$

$$= \langle \underbrace{u(X)_{i_0}}_{n \times 1} \circ e_{i_0}, \mathbf{1}_n \rangle \cdot \langle W_{j_1,*}, X_{*,i_0} \rangle + \langle u(X)_{i_0} \circ (X^\top W_{*,j_1}), \underbrace{\mathbf{1}_n}_{n \times 1} \rangle$$

$$= \langle \underbrace{u(X)_{i_0}, e_{i_0}}_{n \times 1} \rangle \cdot \langle W_{j_1,*}, X_{*,i_0} \rangle + \langle u(X)_{i_0}, X^\top W_{*,j_1} \rangle$$

$$= u(X)_{i_0,i_0} \cdot \langle W_{j_1,*}, X_{*,i_0} \rangle + \langle u(X)_{i_0}, X^\top W_{*,j_1} \rangle$$

where the 1st step follows from the definition of $\alpha(X)_{i_0}$ (see Definition A.5), the 2nd step follows from Fact A.2, the 3rd step follows from Lemma A.11, the 4th step is rearrangement, the 5th step is derived by Fact A.1, the last step is by the definition of $U(X)_{i_0,i_0}$.

**Proof of Part 2.**

$$\underbrace{\frac{d\alpha(X)_{i_0}}{dx_{i_1,j_1}}}_{\text{scalar}} = \underbrace{\frac{d\langle u(X)_{i_0}, \mathbf{1}_n \rangle}{dx_{i_1,j_1}}}_{\text{scalar}}$$

$$= \langle \underbrace{\frac{du(X)_{i_0}}{dx_{i_1,j_1}}}_{n \times 1}, \underbrace{\mathbf{1}_n}_{n \times 1} \rangle$$

$$= \langle \underbrace{u(X)_{i_0}}_{n \times 1} \circ (e_{i_1} \cdot \langle W_{j_1,*}, X_{*,i_0} \rangle), \underbrace{\mathbf{1}_n}_{n \times 1} \rangle$$

$$= \langle \underbrace{u(X)_{i_0}}_{n \times 1} \circ e_{i_1}, \underbrace{\mathbf{1}_n}_{n \times 1} \rangle \cdot \langle W_{j_1,*}, X_{*,i_0} \rangle$$

$$= \underbrace{u(X)_{i_0,i_1}}_{\text{scalar}} \cdot \langle W_{j_1,*}, X_{*,i_0} \rangle$$

where the 1st step follows from the definition of $\alpha(X)_{i_0}$ (see Definition A.5), the 2nd step follows from Fact A.2, the 3rd step follows from Lemma A.11, the 4th step is rearrangement, the 5th step is derived by Fact A.1.

$\square$

### A.6 GRADIENT COMPUTATION FOR $\alpha(X)_{i_0}^{-1}$

**Lemma A.13** (A generalization of Lemma 5.6 in Deng et al. (2023b))**.** *If the following conditions hold*

- *Let $\alpha(X)_{i_0}$ be defined as Definition A.5*

*we have*

- **Part 1.** *For $i_0 = i_1 \in [n]$, $j_1 \in [d]$*

$$\underbrace{\frac{\mathrm{d}\alpha(X)_{i_0}^{-1}}{\mathrm{d}x_{i_1,j_1}}}_{\text{scalar}} = -\alpha(X)_{i_0}^{-1} \cdot (f(X)_{i_0,i_0} \cdot \langle W_{j_1,*}, X_{*,i_0} \rangle + \langle f(X)_{i_0}, X^\top W_{*,j_1} \rangle \rangle)$$

- **Part 2.** *For $i_0 \neq i_1 \in [n]$, $j_1 \in [d]$*

$$\underbrace{\frac{\mathrm{d}\alpha(X)_{i_0}^{-1}}{\mathrm{d}x_{i_1,j_1}}}_{\text{scalar}} = -\alpha(X)_{i_0}^{-1} \cdot f(X)_{i_0,i_1} \cdot \langle W_{j_1,*}, X_{*,i_0} \rangle$$

*Proof.* **Proof of Part 1.**

$$\underbrace{\frac{\mathrm{d}\alpha(X)_{i_0}^{-1}}{\mathrm{d}x_{i_1,j_1}}}_{\text{scalar}} = \underbrace{-1}_{\text{scalar}} \cdot \underbrace{(\alpha(X)_{i_0})^{-2}}_{\text{scalar}} \cdot \underbrace{\frac{\mathrm{d}(\alpha(X)_{i_0})}{\mathrm{d}x_{i_1,j_1}}}_{\text{scalar}}$$

$$= -\underbrace{(\alpha(X)_{i_0})^{-2}}_{\text{scalar}} \cdot (u(X)_{i_0,i_0} \cdot \langle W_{j_1,*}, X_{*,i_0} \rangle + \langle u(X)_{i_0}, X^\top W_{*,j_1} \rangle)$$

$$= -\alpha(X)_{i_0}^{-1} \cdot (f(X)_{i_0,i_0} \cdot \langle W_{j_1,*}, X_{*,i_0} \rangle + \langle f(X)_{i_0}, X^\top W_{*,j_1} \rangle)$$

where the 1st step follows from Fact A.2, the 2nd step follows by Lemma A.12.

**Proof of Part 2.**

$$\underbrace{\frac{\mathrm{d}\alpha(X)_{i_0}^{-1}}{\mathrm{d}x_{i_1,j_1}}}_{\text{scalar}} = \underbrace{-1}_{\text{scalar}} \cdot \underbrace{(\alpha(X)_{i_0})^{-2}}_{\text{scalar}} \cdot \underbrace{\frac{\mathrm{d}(\alpha(X)_{i_0})}{\mathrm{d}x_{i_1,j_1}}}_{\text{scalar}}$$

$$= -\underbrace{(\alpha(X)_{i_0})^{-2}}_{\text{scalar}} \cdot u(X)_{i_0,i_1} \cdot \langle W_{j_1,*}, X_{*,i_0} \rangle$$

$$= -\alpha(X)_{i_0}^{-1} \cdot f(X)_{i_0,i_1} \cdot \langle W_{j_1,*}, X_{*,i_0} \rangle$$

where the 1st step follows from Fact A.2, the 2nd step follows from result from Lemma A.12.

$\square$

### A.7    GRADIENT FOR $f(X)_{i_0}$

**Lemma A.14.** *If the following conditions hold*

- *Let $f(X)_{i_0}$ be defined as Definition A.6*

*Then, we have*

- **Part 1.** *For all $i_0 = i_1 \in [n]$, $j_1 \in [d]$*

$$\underbrace{\frac{\mathrm{d}f(X)_{i_0}}{\mathrm{d}x_{i_1,j_1}}}_{n \times 1} = -\underbrace{f(X)_{i_0}}_{n \times 1} \cdot \underbrace{(f(X)_{i_0,i_0} \cdot \langle W_{j_1,*}, X_{*,i_0} \rangle + \langle f(X)_{i_0}, X^\top W_{*,j_1} \rangle)}_{\text{scalar}}$$

$$+ \underbrace{f(X)_{i_0} \circ (e_{i_0} \cdot \langle W_{j_1,*}, X_{*,i_0} \rangle + X^\top W_{*,j_1})}_{n \times 1}$$

- **Part 2.** *For all $i_0 \neq i_1 \in [n]$, $j_1 \in [d]$*

$$\underbrace{\frac{\mathrm{d}f(X)_{i_0}}{\mathrm{d}x_{i_1,j_1}}}_{n \times 1} = -\underbrace{f(X)_{i_0}}_{n \times 1} \cdot \underbrace{f(X)_{i_0,i_1} \cdot \langle W_{j_1,*}, X_{*,i_0} \rangle}_{\text{scalar}}$$

$$+ \underbrace{f(X)_{i_0} \circ (e_{i_1} \cdot \langle W_{j_1,*}, X_{*,i_0} \rangle)}_{n \times 1}$$

*Proof.* **Proof of Part 1.**

$$\underbrace{\frac{\mathrm{d}f(X)_{i_0}}{\mathrm{d}x_{i_1,j_1}}}_{n \times 1} = \underbrace{\frac{\mathrm{d}\alpha(X)_{i_0}^{-1} u(X)_{i_0}}{\mathrm{d}x_{i_1,j_1}}}_{n \times 1}$$

$$= \underbrace{u(X)_{i_0}}_{n \times 1} \cdot \underbrace{\frac{\mathrm{d}}{\mathrm{d}x_{i_1,j_1}} \alpha(X)_{i_0}^{-1}}_{\text{scalar}} + \underbrace{\alpha(X)_{i_0}^{-1}}_{\text{scalar}} \cdot \underbrace{\frac{\mathrm{d}}{\mathrm{d}x_{i_1,j_1}} u(X)_{i_0}}_{n \times 1}$$

$$= - \underbrace{u(X)_{i_0}}_{n \times 1} \cdot \underbrace{(\alpha(X)_{i_0})^{-1} \cdot (f(X)_{i_0,i_0} \cdot \langle W_{j_1,*}, X_{*,i_0} \rangle + \langle f(X)_{i_0}, X^\top W_{*,j_1} \rangle)}_{\text{scalar}}$$

$$+ \underbrace{\alpha(X)_{i_0}^{-1}}_{\text{scalar}} \cdot \underbrace{\frac{\mathrm{d}}{\mathrm{d}x_{i_1,j_1}} u(X)_{i_0}}_{n \times 1}$$

$$= - \underbrace{u(X)_{i_0}}_{n \times 1} \cdot \underbrace{(\alpha(X)_{i_0})^{-1} \cdot (f(X)_{i_0,i_0} \cdot \langle W_{j_1,*}, X_{*,i_0} \rangle + \langle f(X)_{i_0}, X^\top W_{*,j_1} \rangle)}_{\text{scalar}}$$

$$+ \underbrace{\alpha(X)_{i_0}^{-1}}_{\text{scalar}} \cdot \underbrace{(u(X)_{i_0} \circ (e_{i_0} \cdot \langle W_{j_1,*}, X_{*,i_0} \rangle + X^\top W_{*,j_1}))}_{n \times 1}$$

$$= - \underbrace{f(X)_{i_0}}_{n \times 1} \cdot \underbrace{(f(X)_{i_0,i_0} \cdot \langle W_{j_1,*}, X_{*,i_0} \rangle + \langle f(X)_{i_0}, X^\top W_{*,j_1} \rangle)}_{\text{scalar}}$$

$$+ \underbrace{f(X)_{i_0} \circ (e_{i_0} \cdot \langle W_{j_1,*}, X_{*,i_0} \rangle + X^\top W_{*,j_1})}_{n \times 1}$$

where the 1st step follows from the definition of $f(X)_{i_0}$ (see Definition A.6), the 2nd step follows from Fact A.2, the 3rd step follows from Lemma A.13, the 4th step follows from result from Lemma A.11, the 5th step from the definition of $f(X)_{i_0}$ (see Definition A.6).

**Proof of Part 2.**

$$\underbrace{\frac{\mathrm{d}f(X)_{i_0}}{\mathrm{d}x_{i_1,j_1}}}_{n \times 1} = \underbrace{\frac{\mathrm{d}\alpha(X)_{i_0}^{-1} u(X)_{i_0}}{\mathrm{d}x_{i_1,j_1}}}_{n \times 1}$$

$$= \underbrace{u(X)_{i_0}}_{n \times 1} \cdot \underbrace{\frac{\mathrm{d}}{\mathrm{d}x_{i_1,j_1}} \alpha(X)_{i_0}^{-1}}_{\text{scalar}} + \underbrace{\alpha(X)_{i_0}^{-1}}_{\text{scalar}} \cdot \underbrace{\frac{\mathrm{d}}{\mathrm{d}x_{i_1,j_1}} u(X)_{i_0}}_{n \times 1}$$

$$= - \underbrace{u(X)_{i_0}}_{n \times 1} \cdot \underbrace{(\alpha(X)_{i_0})^{-2} \cdot u(X)_{i_0,i_1} \cdot \langle W_{j_1,*}, X_{*,i_0} \rangle}_{\text{scalar}}$$

$$+ \underbrace{\alpha(X)_{i_0}^{-1}}_{\text{scalar}} \cdot \underbrace{\frac{\mathrm{d}}{\mathrm{d}x_{i_1,j_1}} u(X)_{i_0}}_{n \times 1}$$

$$= - \underbrace{u(X)_{i_0}}_{n \times 1} \cdot \underbrace{(\alpha(X)_{i_0})^{-2} \cdot u(X)_{i_0,i_1} \cdot \langle W_{j_1,*}, X_{*,i_0} \rangle}_{\text{scalar}}$$

$$+ \underbrace{\alpha(X)_{i_0}^{-1}}_{\text{scalar}} \cdot \underbrace{(u(X)_{i_0} \circ (e_{i_1} \cdot \langle W_{j_1,*}, X_{*,i_0} \rangle))}_{n \times 1}$$

$$= - \underbrace{f(X)_{i_0}}_{n \times 1} \cdot \underbrace{f(X)_{i_0,i_1} \cdot \langle W_{j_1,*}, X_{*,i_0} \rangle}_{\text{scalar}}$$

$$+ e_{i_1} \cdot \underbrace{f(X)_{i_0,i_1} \cdot \langle W_{j_1,*}, X_{*,i_0} \rangle)}_{\text{scalar}}$$

where the 1st step follows from the definition of $f(X)_{i_0}$ (see Definition A.6), the 2nd step follows from Fact A.2, the 3rd step follows from Lemma A.13, the 4th step follows from result from Lemma A.11, the 5th step from the definition of $f(X)_{i_0}$ (see Definition A.6). $\qquad\square$

## A.8 GRADIENT FOR $h(X)_{j_0}$

**Lemma A.15.** *If the following conditions hold*

- *Let $h(X)_{j_0}$ be defined as Definition A.7*

*Then, for all $i_1 \in [n]$, $j_0, j_1 \in [d]$, we have*

$$\underbrace{\frac{\mathrm{d}h(X)_{j_0}}{\mathrm{d}x_{i_1,j_1}}}_{n \times 1} = e_{i_1} \cdot v_{j_1,j_0}$$

*Proof.*

$$\underbrace{\frac{\mathrm{d}h(X)_{j_0}}{\mathrm{d}x_{i_1,j_1}}}_{n \times 1} = \underbrace{\frac{\mathrm{d}X^\top V_{*,j_0}}{\mathrm{d}x_{i_1,j_1}}}_{n \times 1}$$

$$= \underbrace{\frac{\mathrm{d}X^\top}{\mathrm{d}x_{i_1,j_1}}}_{n \times d} \cdot \underbrace{V_{*,j_0}}_{d \times 1}$$

$$= \underbrace{e_{i_1}}_{n \times 1} \cdot \underbrace{e_{j_1}^\top}_{1 \times d} \cdot \underbrace{V_{*,j_0}}_{d \times 1}$$

$$= \underbrace{e_{i_1}}_{n \times 1} \cdot \underbrace{v_{j_1,j_0}}_{\text{scalar}}$$

where the first step is by definition of $h(X)_{j_0}$ (see Definition A.7), the 2nd and the 3rd step are by differentiation rules, the 4th step is by simple algebra. $\qquad\square$

## A.9 GRADIENT FOR $c(X)_{i_0,j_0}$

**Lemma A.16.** *If the following conditions hold*

- *Let $c(X)_{i_0}$ be defined as Definition A.8*

- *Let $s(X)_{i_0,j_0} := \langle f(X)_{i_0}, h(X)_{j_0} \rangle$*

*Then, we have*

- **Part 1.** *For all $i_0 = i_1 \in [n]$, $j_0, j_1 \in [d]$*

$$\frac{\mathrm{d}c(X)_{i_0,j_0}}{\mathrm{d}x_{i_1,j_1}} = C_1(X) + C_2(X) + C_3(X) + C_4(X) + C_5(X)$$

  *where we have definitions:*

  - $C_1(X) := -s(X)_{i_0,j_0} \cdot f(X)_{i_0,i_0} \cdot \langle W_{j_1,*}, X_{*,i_0} \rangle$
  - $C_2(X) := -s(X)_{i_0,j_0} \cdot \langle f(X)_{i_0}, X^\top W_{*,j_1} \rangle$
  - $C_3(X) := f(X)_{i_0,i_0} \cdot h(X)_{j_0,i_0} \cdot \langle W_{j_1,*}, X_{*,i_0} \rangle$
  - $C_4(X) := \langle f(X)_{i_0} \circ (X^\top W_{*,j_1}), h(X)_{j_0} \rangle$
  - $C_5(X) := f(X)_{i_0,i_0} \cdot v_{j_1,j_0}$

- **Part 2.** *For all $i_0 \neq i_1 \in [n]$, $j_0, j_1 \in [d]$*

$$\frac{\mathrm{d}c(X)_{i_0,j_0}}{\mathrm{d}x_{i_1,j_1}} = C_6(X) + C_7(X) + C_8(X)$$

*where we have definitions:*

- $C_6(X) := -s(X)_{i_0,j_0} \cdot f(X)_{i_0,i_1} \cdot \langle W_{j_1,*}, X_{*,i_0} \rangle$
    * *This is corresponding to $C_1(X)$*
- $C_7(X) := f(X)_{i_0,i_1} \cdot h(X)_{j_0,i_1} \cdot \langle W_{j_1,*}, X_{*,i_0} \rangle$
    * *This is corresponding to $C_3(X)$*
- $C_8(X) := f(X)_{i_0,i_1} \cdot v_{j_1,j_0}$
    * *This is corresponding to $C_5(X)$*

*Proof.* **Proof of Part 1**

$$\underbrace{\frac{\mathrm{d}c(X)_{i_0,j_1}}{\mathrm{d}x_{i_1,j_1}}}_{\text{scalar}} = \underbrace{\frac{\mathrm{d}(\langle f(X)_{i_0}, h(X)_{j_0} \rangle - b_{i_0,j_0})}{\mathrm{d}x_{i_1,j_1}}}_{\text{scalar}}$$

$$= \underbrace{\frac{\mathrm{d}\langle f(X)_{i_0}, h(X)_{j_0} \rangle}{\mathrm{d}x_{i_1,j_1}}}_{\text{scalar}}$$

$$= \langle \underbrace{\frac{\mathrm{d}f(X)_{i_0}}{\mathrm{d}x_{i_1,j_1}}}_{n \times 1}, \underbrace{h(X)_{j_0}}_{n \times 1} \rangle + \langle \underbrace{f(X)_{i_0}}_{n \times 1}, \underbrace{\frac{\mathrm{d}h(X)_{j_0}}{\mathrm{d}x_{i_1,j_1}}}_{n \times 1} \rangle$$

$$= \langle \underbrace{\frac{\mathrm{d}f(X)_{i_0}}{\mathrm{d}x_{i_1,j_1}}}_{n \times 1}, \underbrace{h(X)_{j_0}}_{n \times 1} \rangle + \langle \underbrace{f(X)_{i_0}}_{n \times 1}, \underbrace{e_{i_1}}_{n \times 1} \cdot \underbrace{v_{j_1,j_0}}_{\text{scalar}} \rangle$$

$$= \langle - \underbrace{f(X)_{i_0}}_{n \times 1} \cdot \underbrace{(f(X)_{i_0,i_0} \cdot \langle W_{j_1,*}, X_{*,i_0} \rangle + \langle f(X)_{i_0}, X^\top W_{*,j_1} \rangle)}_{\text{scalar}}$$

$$+ \underbrace{f(X)_{i_0} \circ (e_{i_0} \cdot \langle W_{j_1,*}, X_{*,i_0} \rangle + X^\top W_{*,j_1})}_{n \times 1}, \underbrace{h(X)_{j_0}}_{n \times 1} \rangle + \langle \underbrace{f(X)_{i_0}}_{n \times 1}, \underbrace{e_{i_1}}_{n \times 1} \cdot \underbrace{v_{j_1,j_0}}_{\text{scalar}} \rangle$$

$$= - s(X)_{i_0,j_0} \cdot f(X)_{i_0,i_0} \cdot \langle W_{j_1,*}, X_{*,i_0} \rangle$$
$$- s(X)_{i_0,j_0} \cdot \langle f(X)_{i_0}, X^\top W_{*,j_1} \rangle$$
$$+ f(X)_{i_0,i_0} h(X)_{j_0,i_0} \langle W_{j_1,*}, X_{*,i_0} \rangle$$
$$+ \langle f(X)_{i_0} \circ (X^\top W_{*,j_1}), h(X)_{j_0} \rangle$$
$$+ f(X)_{i_0,i_1} v_{j_1,j_0}$$
$$:= C_1(X) + C_2(X) + C_3(X) + C_4(X) + C_5(X)$$

where the first step is by definition of $c(X)_{i_0,j_0}$ (see Definition A.8), the 2nd step is because $b_{i_0,j_0}$ is independent of $X$, the 3rd step is by Fact A.2, the 4th step uses Lemma A.15, the 5th step uses Lemma A.14, the 6th and 8th step are rearrangement of terms, the 7th step holds by the definition of $f(X)_{i_0}$ (see Definition A.6).

**Proof of Part 2**

$$\underbrace{\frac{\mathrm{d}c(X)_{i_0,j_1}}{\mathrm{d}x_{i_1,j_1}}}_{\text{scalar}} = \underbrace{\frac{\mathrm{d}(\langle f(X)_{i_0}, h(X)_{j_0} \rangle - b_{i_0,j_0})}{\mathrm{d}x_{i_1,j_1}}}_{\text{scalar}}$$

$$= \underbrace{\frac{\mathrm{d}\langle f(X)_{i_0}, h(X)_{j_0} \rangle}{\mathrm{d}x_{i_1,j_1}}}_{\text{scalar}}$$

$$
= \langle \underbrace{\frac{\mathrm{d}f(X)_{i_0}}{\mathrm{d}x_{i_1,j_1}}}_{n\times 1}, \underbrace{h(X)_{j_0}}_{n\times 1}\rangle + \langle \underbrace{f(X)_{i_0}}_{n\times 1}, \underbrace{\frac{\mathrm{d}h(X)_{j_0}}{\mathrm{d}x_{i_1,j_1}}}_{n\times 1}\rangle
$$

$$
= \langle \underbrace{\frac{\mathrm{d}f(X)_{i_0}}{\mathrm{d}x_{i_1,j_1}}}_{n\times 1}, \underbrace{h(X)_{j_0}}_{n\times 1}\rangle + \langle \underbrace{f(X)_{i_0}}_{n\times 1}, \underbrace{e_{i_1}}_{n\times 1} \cdot \underbrace{v_{j_1,j_0}}_{\text{scalar}}\rangle
$$

$$
= \langle - \underbrace{(\alpha(X)_{i_0})^{-1}}_{\text{scalar}} \cdot \underbrace{f(X)_{i_0}}_{n\times 1} \cdot \underbrace{u(X)_{i_0,i_1} \cdot \langle W_{j_1,*}, X_{*,i_0}\rangle}_{\text{scalar}}
$$

$$
+ \underbrace{f(X)_{i_0} \circ (e_{i_1} \cdot \langle W_{j_1,*}, X_{*,i_0}\rangle)}_{n\times 1}, \underbrace{h(X)_{j_0}}_{n\times 1}\rangle + \langle \underbrace{f(X)_{i_0}}_{n\times 1}, \underbrace{e_{i_1}}_{n\times 1} \cdot \underbrace{v_{j_1,j_0}}_{\text{scalar}}\rangle
$$

$$
= - \underbrace{(\alpha(X)_{i_0})^{-1} \cdot \langle f(X)_{i_0}, h(X)_{j_0}\rangle \cdot u(X)_{i_0,i_1} \cdot \langle W_{j_1,*}, X_{*,i_0}\rangle}_{\text{scalar}}
$$

$$
+ \underbrace{\langle f(X)_{i_0} \circ e_{i_1}, h(X)_{j_0}\rangle \cdot \langle W_{j_1,*}, X_{*,i_0}\rangle}_{\text{scalar}}
$$

$$
+ \langle \underbrace{f(X)_{i_0}}_{n\times 1}, \underbrace{e_{i_1}}_{n\times 1} \cdot \underbrace{v_{j_1,j_0}}_{\text{scalar}}\rangle
$$

$$
= - s(X)_{i_0,j_0} \cdot f(X)_{i_0,i_1} \cdot \langle W_{j_1,*}, X_{*,i_0}\rangle
$$

$$
+ f(X)_{i_0,i_1} \cdot h(X)_{j_0,i_1} \cdot \langle W_{j_1,*}, X_{*,i_0}\rangle
$$

$$
+ f(X)_{i_0,i_1} \cdot v_{j_1,j_0}
$$

$$
:= C_6(X) + C_7(X) + C_8(X)
$$

where the first step is by definition of $c(X)_{i_0,j_0}$ (see Definition A.8), the 2nd step is because $b_{i_0,j_0}$ is independent of $X$, the 3rd step is by Fact A.2, the 4th step uses Lemma A.15, the 5th step uses Lemma A.14, the 6th and 7th step are rearrangement of terms. $\square$

### A.10 GRADIENT FOR $L(X)$

**Lemma A.17.** *If the following holds*

- *Let $L(X)$ be defined as Definition A.9*

*For $i_1 \in [n]$, $j_1 \in [d]$, we have*

$$
\frac{\mathrm{d}L(X)}{\mathrm{d}x_{i_1,j_1}} = \sum_{i_0=1}^{n} \sum_{j_0=1}^{d} c(X)_{i_0,j_0} \cdot \frac{\mathrm{d}c(X)_{i_0,j_0}}{\mathrm{d}x_{i_1,j_1}}
$$

*Proof.* The result directly follows by chain rule. $\square$

## B HESSIAN CASE 1: $i_0 = i_1$

Here in this section, we provide Hessian analysis for the first case. In Sections B.1, B.2, B.3, B.4, B.5, B.6 and B.8, we calculate the derivative for several important terms. In Section B.9, B.10, B.11, B.12 and B.13 we calculate derivative for $C_1, C_2, C_3, C_4$ and $C_5$ respectively. Finally in Section B.14 we calculate derivative of $\frac{c(X)_{i_0,j_0}}{\mathrm{d}x_{i_1,j_1}\mathrm{d}_{i_2,j_2}}$.

Now, we list some simplified notations which will be used in following sections.

**Definition B.1.** *We have following definitions to simplify the expression.*

- $s(X)_{i,j} := \langle f(X)_i, h(X)_j\rangle$

- $w(X)_{i,j} := \langle W_{j,*}, X_{*,i}\rangle$

- $z(X)_{i,j} := \langle f(X)_i, X^\top W_{*,j}\rangle$

- $z(X)_i := WX \cdot f(X)_i$

- $w(X)_{i,*} := WX_{*,i}$

## B.1 DERIVATIVE OF SCALAR FUNCTION $w(X)_{i_0,j_1}$

**Lemma B.2.** *We have*

- **Part 1** *For $i_0 = i_1 = i_2 \in [n]$, $j_1, j_2 \in [d]$*

$$\frac{\mathrm{d}w(X)_{i_0,j_1}}{\mathrm{d}x_{i_2,j_2}} = w_{j_1,j_2}$$

- **Part 2** *For $i_0 = i_1 \neq i_2 \in [n]$, $j_1, j_2 \in [d]$*

$$\frac{\mathrm{d}w(X)_{i_0,j_1}}{\mathrm{d}x_{i_2,j_2}} = 0$$

*Proof.* **Proof of Part 1**

$$\begin{aligned}
\frac{\mathrm{d}w(X)_{i_0,j_1}}{\mathrm{d}x_{i_2,j_2}} &= \langle W_{j_1,*}, \frac{\mathrm{d}X_{*,i_0}}{\mathrm{d}x_{i_2,j_2}} \rangle \\
&= \langle W_{j_1,*}, e_{j_2} \rangle \\
&= w_{j_1,j_2}
\end{aligned}$$

where the first step and the 2nd step are by Fact A.2, the 3rd step is simple algebra.

**Proof of Part 2**

$$\begin{aligned}
\frac{\mathrm{d}w(X)_{i_0,j_1}}{\mathrm{d}x_{i_2,j_2}} &= \langle W_{j_1,*}, \frac{\mathrm{d}X_{*,i_0}}{\mathrm{d}x_{i_2,j_2}} \rangle \\
&= \langle W_{j_1,*}, \mathbf{0}_d \rangle \\
&= 0
\end{aligned}$$

where the first step is by Fact A.2, the 2nd step is because $i_0 \neq i_2$. $\square$

## B.2 DERIVATIVE OF VECTOR FUNCTION $X^\top W_{*,j_1}$

**Lemma B.3.** *We have*

- **Part 1** *For $i_0 = i_1 = i_2 \in [n]$, $j_1, j_2 \in [d]$*

$$\frac{\mathrm{d}X^\top W_{*,j_1}}{\mathrm{d}x_{i_2,j_2}} = e_{i_0} \cdot w_{j_2,j_1}$$

- **Part 2** *For $i_0 = i_1 \neq i_2 \in [n]$, $j_1, j_2 \in [d]$*

$$\frac{\mathrm{d}X^\top W_{*,j_1}}{\mathrm{d}x_{i_2,j_2}} = e_{i_2} \cdot w_{j_2,j_1}$$

*Proof.* **Proof of Part 1**

$$\begin{aligned}
\frac{\mathrm{d}X^\top W_{*,j_1}}{\mathrm{d}x_{i_2,j_2}} &= \frac{\mathrm{d}X^\top}{\mathrm{d}x_{i_2,j_2}} \cdot W_{*,j_1} \\
&= e_{i_2} e_{j_2}^\top \cdot W_{*,j_1} \\
&= e_{i_2} \cdot w_{j_2,j_1} \\
&= e_{i_0} \cdot w_{j_2,j_1}
\end{aligned}$$

where the first step and the 2nd step are by Fact A.2, the 3rd step is simple algebra, the 4th step holds since $i_0 = i_2$.

**Proof of Part 2**

$$
\frac{\mathrm{d}X^\top W_{*,j_1}}{\mathrm{d}x_{i_2,j_2}} = \frac{\mathrm{d}X^\top}{\mathrm{d}x_{i_2,j_2}} \cdot W_{*,j_1}
$$
$$
= e_{i_2} e_{j_2}^\top \cdot W_{*,j_1}
$$
$$
= e_{i_2} \cdot w_{j_2,j_1}
$$

where the first step and the 2nd step are by Fact A.2, the 3rd step is simple algebra. $\qquad \square$

## B.3  DERIVATIVE OF SCALAR FUNCTION $f(X)_{i_0,i_0}$

**Lemma B.4.** *If the following holds:*

- *Let $f(X)_{i_0}$ be defined as Definition A.6*

*We have*

- **Part 1** *For $i_0 = i_2 \in [n]$, $j_1, j_2 \in [d]$*

$$
\frac{\mathrm{d}f(X)_{i_0,i_0}}{\mathrm{d}x_{i_2,j_2}} = - f(X)_{i_0,i_0} \cdot (f(X)_{i_0,i_0} \cdot w(X)_{i_0,j_2} + \langle f(X)_{i_0}, X^\top W_{*,j_2}\rangle)
$$
$$
+ f(X)_{i_0,i_0} \cdot \langle W_{j_2,*} + W_{*,j_2}, X_{*,i_0}\rangle
$$

- **Part 2** *For $i_0 \ne i_2 \in [n]$, $j_1, j_2 \in [d]$*

$$
\frac{\mathrm{d}f(X)_{i_0,i_0}}{\mathrm{d}x_{i_2,j_2}} = -f(X)_{i_0,i_0} \cdot f(X)_{i_0,i_2} \cdot w(X)_{i_0,j_2}
$$

*Proof.* **Proof of Part 1**

$$
\frac{\mathrm{d}f(X)_{i_0,i_0}}{\mathrm{d}x_{i_2,j_2}} = (-(\alpha(X)_{i_0})^{-1} \cdot f(X)_{i_0} \cdot (u(X)_{i_0,i_0} \cdot w(X)_{i_0,j_2} + \langle u(X)_{i_0}, X^\top W_{*,j_2}\rangle)
$$
$$
+ f(X)_{i_0} \circ (e_{i_0} \cdot w(X)_{i_0,j_2} + X^\top W_{*,j_2}))_{i_0}
$$
$$
= - (\alpha(X)_{i_0})^{-1} \cdot f(X)_{i_0,i_0} \cdot (u(X)_{i_0,i_0} \cdot w(X)_{i_0,j_2} + \langle u(X)_{i_0}, X^\top W_{*,j_2}\rangle)
$$
$$
+ (f(X)_{i_0} \circ (e_{i_0} \cdot w(X)_{i_0,j_2}))_{i_0} + (f(X)_{i_0} \circ (X^\top W_{*,j_2}))_{i_0}
$$
$$
= - (\alpha(X)_{i_0})^{-1} \cdot f(X)_{i_0,i_0} \cdot (u(X)_{i_0,i_0} \cdot w(X)_{i_0,j_2} + \langle u(X)_{i_0}, X^\top W_{*,j_2}\rangle)
$$
$$
+ f(X)_{i_0,i_0} \cdot w(X)_{i_0,j_2} + f(X)_{i_0,i_0} \cdot \langle W_{*,j_2}, X_{*,i_0}\rangle
$$
$$
= - f(X)_{i_0,i_0} \cdot (f(X)_{i_0,i_0} \cdot w(X)_{i_0,j_2} + \langle f(X)_{i_0}, X^\top W_{*,j_2}\rangle)
$$
$$
+ f(X)_{i_0,i_0} \cdot w(X)_{i_0,j_2} + f(X)_{i_0,i_0} \cdot \langle W_{*,j_2}, X_{*,i_0}\rangle
$$

where the first step uses Lemma A.14 for $i_0 = i_2$, the following steps are taking the $i_0$-th entry of $f(X)_{i_0}$, the last step is by the definition of $f(X)_{i_0}$ (see Definition A.6).

**Proof of Part 2**

$$
\frac{\mathrm{d}f(X)_{i_0,i_0}}{\mathrm{d}x_{i_2,j_2}} = (-(\alpha(X)_{i_0})^{-1} \cdot f(X)_{i_0} \cdot u(X)_{i_0,i_2} \cdot w(X)_{i_0,j_2}
$$
$$
+ f(X)_{i_0} \circ (e_{i_2} \cdot w(X)_{i_0,j_2}))_{i_0}
$$
$$
= - (\alpha(X)_{i_0})^{-1} \cdot f(X)_{i_0,i_0} \cdot u(X)_{i_0,i_2} \cdot w(X)_{i_0,j_2}
$$
$$
+ (f(X)_{i_0} \circ (e_{i_2} \cdot w(X)_{i_0,j_2}))_{i_0}
$$
$$
= - (\alpha(X)_{i_0})^{-1} \cdot f(X)_{i_0,i_0} \cdot u(X)_{i_0,i_2} \cdot w(X)_{i_0,j_2}
$$
$$
= - f(X)_{i_0,i_0} \cdot f(X)_{i_0,i_2} \cdot w(X)_{i_0,j_2}
$$

where the first step uses Lemma A.14 for $i_0 \ne i_2$, the 2nd step is taking the $i_0$-th entry of $f(X)_{i_0}$, the 3rd step is because $i_0 \ne i_2$, the last step is by the definition of $f(X)_{i_0}$ (see Definition A.6). $\quad \square$

### B.4 DERIVATIVE OF SCALAR FUNCTION $h(X)_{j_0,i_0}$

**Lemma B.5.** *If the following holds:*

- *Let $h(X)_{j_0}$ be defined as Definition A.7*

*We have*

- **Part 1** *For $i_0 = i_2 \in [n]$, $j_1, j_2 \in [d]$*
$$\frac{\mathrm{d}h(X)_{j_0,i_0}}{\mathrm{d}x_{i_2,j_2}} = v_{j_2,j_0}$$

- **Part 2** *For $i_0 \neq i_2 \in [n]$, $j_1, j_2 \in [d]$*
$$\frac{\mathrm{d}h(X)_{j_0,i_0}}{\mathrm{d}x_{i_2,j_2}} = 0$$

*Proof.* **Proof of Part 1**
$$\frac{\mathrm{d}h(X)_{j_0,i_0}}{\mathrm{d}x_{i_2,j_2}} = (e_{i_2} \cdot v_{j_2,j_0})_{i_0}$$
$$= v_{j_2,j_0}$$
where the first step is by Lemma A.15, the 2nd step is because $i_0 = i_2$.

**Proof of Part 2**
$$\frac{\mathrm{d}h(X)_{j_0,i_0}}{\mathrm{d}x_{i_2,j_2}} = (e_{i_2} \cdot v_{j_2,j_0})_{i_0}$$
$$= 0$$
where the first step is by Lemma A.15, the 2nd step is because $i_0 \neq i_2$. $\qquad\square$

### B.5 DERIVATIVE OF SCALAR FUNCTION $z(X)_{i_0,j_1}$

**Lemma B.6.** *If the following holds:*

- *Let $f(X)_{i_0}$ be defined as Definition A.6*

- *Let $z(X)_{i_0,j_1} := \langle f(X)_{i_0}, X^\top W_{*,j_1} \rangle$*

- *Let $w(X)_{i_0,j_1} = \langle W_{j_1,*}, X_{*,i_0} \rangle$*

*We have*

- **Part 1** *For $i_0 = i_1 = i_2 \in [n]$, $j_1, j_2 \in [d]$*
$$\frac{\mathrm{d}z(X)_{i_0,j_1}}{\mathrm{d}x_{i_2,j_2}}$$
$$= - z(X)_{i_0,j_1} \cdot f(X)_{i_0,i_0} \cdot w(X)_{i_0,j_2}$$
$$- z(X)_{i_0,j_1} \cdot z(X)_{i_0,j_2}$$
$$+ f(X)_{i_0,i_0} \cdot \langle W_{*,j_1}, X_{*,i_0} \rangle \cdot w(X)_{i_0,j_2}$$
$$+ \langle f(X)_{i_0} \circ X^\top W_{*,j_2}, X^\top W_{*,j_1} \rangle$$
$$+ f(X)_{i_0,i_0} \cdot w_{j_2,j_1}$$

- **Part 2** *For $i_0 = i_1 \neq i_2 \in [n]$, $j_1, j_2 \in [d]$*
$$\frac{\mathrm{d}\langle f(X)_{i_0}, X^\top W_{*,j_1} \rangle}{\mathrm{d}x_{i_2,j_2}}$$
$$= - z(X)_{i_0,j_1} \cdot f(X)_{i_0,i_0} \cdot w(X)_{i_0,j_2}$$
$$+ f(X)_{i_0,i_0} \cdot w(X)_{i_0,j_2} \cdot \langle W_{*,j_1}, X_{*,i_0} \rangle$$
$$+ f(X)_{i_0,i_0} \cdot w_{j_2,j_1}$$

*Proof.* **Proof of Part 1**

$$\frac{\mathrm{d}\langle f(X)_{i_0}, X^\top W_{*,j_1}\rangle}{\mathrm{d}x_{i_2,j_2}}$$

$$= \langle \frac{\mathrm{d}f(X)_{i_0}}{\mathrm{d}x_{i_2,j_2}}, X^\top W_{*,j_1}\rangle + \langle f(X)_{i_0}, \frac{\mathrm{d}X^\top W_{*,j_1}}{\mathrm{d}x_{i_2,j_2}}\rangle$$

$$= \langle \frac{\mathrm{d}f(X)_{i_0}}{\mathrm{d}x_{i_2,j_2}}, X^\top W_{*,j_1}\rangle + \langle f(X)_{i_0}, e_{i_0} \cdot w_{j_2,j_1}\rangle$$

$$= \langle \frac{\mathrm{d}f(X)_{i_0}}{\mathrm{d}x_{i_2,j_2}}, X^\top W_{*,j_1}\rangle + f(X)_{i_0,i_0} \cdot w_{j_2,j_1}$$

$$= \langle -(\alpha(X)_{i_0})^{-1} \cdot f(X)_{i_0} \cdot (u(X)_{i_0,i_0} \cdot w(X)_{i_0,j_2} + \langle u(X)_{i_0}, X^\top W_{*,j_2}\rangle)$$
$$+ f(X)_{i_0} \circ (e_{i_0} \cdot w(X)_{i_0,j_2} + X^\top W_{*,j_2}), X^\top W_{*,j_1}\rangle + f(X)_{i_0,i_0} \cdot w_{j_2,j_1}$$

$$= \langle -f(X)_{i_0} \cdot (f(X)_{i_0,i_0} \cdot w(X)_{i_0,j_2} + \langle f(X)_{i_0}, X^\top W_{*,j_2}\rangle)$$
$$+ f(X)_{i_0} \circ (e_{i_0} \cdot w(X)_{i_0,j_2} + X^\top W_{*,j_2}), X^\top W_{*,j_1}\rangle + f(X)_{i_0,i_0} \cdot w_{j_2,j_1}$$

$$= - z(X)_{i_0,j_1} \cdot f(X)_{i_0,i_0} \cdot w(X)_{i_0,j_2}$$
$$- z(X)_{i_0,j_1} \cdot z(X)_{i_0,j_2}$$
$$+ f(X)_{i_0,i_0} \cdot \langle W_{*,j_1}, X_{*,i_0}\rangle \cdot w(X)_{i_0,j_2}$$
$$+ \langle f(X)_{i_0} \circ X^\top W_{*,j_2}, X^\top W_{*,j_1}\rangle$$
$$+ f(X)_{i_0,i_0} \cdot w_{j_2,j_1}$$

where the 1st step is by Fact A.2, the 2nd step uses Lemma B.3, the 3rd step is taking the $i_0$-th entry of $f(X)_{i_0}$, the 4th step uses Lemma A.14, the 5th step is by the definition of $f(X)_{i_0}$ (see Definition A.6).

**Proof of Part 2**

$$\frac{\mathrm{d}\langle f(X)_{i_0}, X^\top W_{*,j_1}\rangle}{\mathrm{d}x_{i_2,j_2}}$$

$$= \langle \frac{\mathrm{d}f(X)_{i_0}}{\mathrm{d}x_{i_2,j_2}}, X^\top W_{*,j_1}\rangle + \langle f(X)_{i_0}, \frac{\mathrm{d}X^\top W_{*,j_1}}{\mathrm{d}x_{i_2,j_2}}\rangle$$

$$= \langle \frac{\mathrm{d}f(X)_{i_0}}{\mathrm{d}x_{i_2,j_2}}, X^\top W_{*,j_1}\rangle + \langle f(X)_{i_0}, e_{i_2} \cdot w_{j_2,j_1}\rangle$$

$$= \langle \frac{\mathrm{d}f(X)_{i_0}}{\mathrm{d}x_{i_2,j_2}}, X^\top W_{*,j_1}\rangle + f(X)_{i_0,i_2} \cdot w_{j_2,j_1}$$

$$= \langle -(\alpha(X)_{i_0})^{-1} \cdot f(X)_{i_0} \cdot u(X)_{i_0,i_0} \cdot w(X)_{i_0,j_2}$$
$$+ f(X)_{i_0} \circ (e_{i_0} \cdot w(X)_{i_0,j_2}), X^\top W_{*,j_1}\rangle + f(X)_{i_0,i_0} \cdot w_{j_2,j_1}$$

$$= \langle -f(X)_{i_0} \cdot f(X)_{i_0,i_0} \cdot w(X)_{i_0,j_2}$$
$$+ f(X)_{i_0} \circ (e_{i_0} \cdot w(X)_{i_0,j_2}), X^\top W_{*,j_1}\rangle + f(X)_{i_0,i_0} \cdot w_{j_2,j_1}$$

$$= - z(X)_{i_0,j_1} \cdot f(X)_{i_0,i_0} \cdot w(X)_{i_0,j_2}$$
$$+ f(X)_{i_0,i_0} \cdot w(X)_{i_0,j_2} \cdot \langle W_{*,j_1}, X_{*,i_0}\rangle$$
$$+ f(X)_{i_0,i_0} \cdot w_{j_2,j_1}$$

where the 1st step is by Fact A.2, the 2nd step uses Lemma B.3, the 3rd step is taking the $i_0$-th entry of $f(X)_{i_0}$, the 4th step uses Lemma A.14, the last step is by the definition of $f(X)_{i_0}$ (see Definition A.6). $\square$

## B.6 Derivative of Scalar Function $f(X)_{i_0,i_0} \cdot h(X)_{j_0,i_0}$

**Lemma B.7.** *If the following holds:*

- *Let $f(X)_{i_0}$ be defined as Definition A.6*

- Let $h(X)_{j_0}$ be defined as Definition A.7

*We have*

- **Part 1** *For $i_0 = i_1 = i_2 \in [n]$, $j_1, j_2 \in [d]$*

$$\frac{\mathrm{d} f(X)_{i_0,i_0} \cdot h(X)_{j_0,i_0}}{\mathrm{d} x_{i_2,j_2}}$$

$$= (-f(X)_{i_0,i_0} \cdot (f(X)_{i_0,i_0} \cdot w(X)_{i_0,j_2} + \langle f(X)_{i_0}, X^\top W_{*,j_2} \rangle)$$
$$+ f(X)_{i_0,i_0} \cdot \langle W_{j_2,*} + W_{*,j_2}, X_{*,i_0} \rangle) \cdot h(X)_{j_0,i_0} + f(X)_{i_0,i_0} \cdot v_{j_2,j_0}$$

- **Part 2** *For $i_0 = i_1 \neq i_2 \in [n]$, $j_1, j_2 \in [d]$*

$$\frac{\mathrm{d} f(X)_{i_0,i_0} \cdot h(X)_{j_0,i_0}}{\mathrm{d} x_{i_2,j_2}} = -f(X)_{i_0,i_0} \cdot f(X)_{i_0,i_2} \cdot w(X)_{i_0,j_2} \cdot h(X)_{j_0,i_0}$$

*Proof.* **Proof of Part 1**

$$\frac{\mathrm{d} f(X)_{i_0,i_0} \cdot h(X)_{j_0,i_0}}{\mathrm{d} x_{i_2,j_2}}$$

$$= \frac{\mathrm{d} f(X)_{i_0,i_0}}{\mathrm{d} x_{i_2,j_2}} \cdot h(X)_{j_0,i_0} + f(X)_{i_0,i_0} \cdot \frac{\mathrm{d} h(X)_{j_0,i_0}}{\mathrm{d} x_{i_2,j_2}}$$

$$= \frac{\mathrm{d} f(X)_{i_0,i_0}}{\mathrm{d} x_{i_2,j_2}} \cdot h(X)_{j_0,i_0} + f(X)_{i_0,i_0} \cdot v_{j_2,j_0}$$

$$= (-(\alpha(X)_{i_0})^{-1} \cdot f(X)_{i_0,i_0} \cdot (u(X)_{i_0,i_0} \cdot w(X)_{i_0,j_2} + \langle u(X)_{i_0}, X^\top W_{*,j_2} \rangle)$$
$$+ f(X)_{i_0,i_0} \cdot \langle W_{j_2,*} + W_{*,j_2}, X_{*,i_0} \rangle) \cdot h(X)_{j_0,i_0} + f(X)_{i_0,i_0} \cdot v_{j_2,j_0}$$

$$= (-f(X)_{i_0,i_0} \cdot (f(X)_{i_0,i_0} \cdot w(X)_{i_0,j_2} + \langle f(X)_{i_0}, X^\top W_{*,j_2} \rangle)$$
$$+ f(X)_{i_0,i_0} \cdot \langle W_{j_2,*} + W_{*,j_2}, X_{*,i_0} \rangle) \cdot h(X)_{j_0,i_0} + f(X)_{i_0,i_0} \cdot v_{j_2,j_0}$$

where the fist step is by Fact A.2, the 2nd step calls Lemma B.5, the 3rd step uses Lemma B.4, the last step is by the definition of $f(X)_{i_0}$ (see Definition A.6).

**Proof of Part 2**

$$\frac{\mathrm{d} f(X)_{i_0,i_0} \cdot h(X)_{j_0,i_0}}{\mathrm{d} x_{i_2,j_2}}$$

$$= \frac{\mathrm{d} f(X)_{i_0,i_0}}{\mathrm{d} x_{i_2,j_2}} \cdot h(X)_{j_0,i_0} + f(X)_{i_0,i_0} \cdot \frac{\mathrm{d} h(X)_{j_0,i_0}}{\mathrm{d} x_{i_2,j_2}}$$

$$= -(\alpha(X)_{i_0})^{-1} \cdot f(X)_{i_0,i_0} \cdot u(X)_{i_0,i_2} \cdot w(X)_{i_0,j_2} \cdot h(X)_{j_0,i_0}$$

$$= -f(X)_{i_0,i_0} \cdot f(X)_{i_0,i_2} \cdot w(X)_{i_0,j_2} \cdot h(X)_{j_0,i_0}$$

where the fist step is by Fact A.2, the 2nd step calls Lemma B.5, the 3rd step uses Lemma B.4, the last step is by the definition of $f(X)_{i_0}$ (see Definition A.6). $\square$

### B.7 Derivative of Scalar Function $f(X)_{i_0,i_0} \cdot w(X)_{i_0,j_1}$

**Lemma B.8.** *If the following holds:*

- *Let $f(X)_{i_0}$ be defined as Definition A.6*

*We have*

- **Part 1** *For $i_0 = i_1 = i_2 \in [n]$, $j_1, j_2 \in [d]$*

$$\frac{\mathrm{d} f(X)_{i_0,i_0} \cdot w(X)_{i_0,j_1}}{\mathrm{d} x_{i_2,j_2}}$$

$$= (f(X)_{i_0,i_0} \cdot (f(X)_{i_0,i_0} \cdot w(X)_{i_0,j_2} + \langle f(X)_{i_0}, X^\top W_{*,j_2} \rangle)$$
$$+ f(X)_{i_0,i_0} \cdot \langle W_{j_2,*} + W_{*,j_2}, X_{*,i_0} \rangle) \cdot w(X)_{i_0,j_1} + f(X)_{i_0,i_0} \cdot w_{j_1,j_2}$$

- **Part 2** *For $i_0 = i_1 \neq i_2 \in [n]$, $j_1, j_2 \in [d]$*

$$\frac{\mathrm{d}f(X)_{i_0,i_0} \cdot w(X)_{i_0,j_1}}{\mathrm{d}x_{i_2,j_2}} = -f(X)_{i_0,i_0} \cdot f(X)_{i_0,i_2} \cdot w(X)_{i_0,j_2} \cdot w(X)_{i_0,j_1}$$

*Proof.* **Proof of Part 1**

$$\frac{\mathrm{d}f(X)_{i_0,i_0} \cdot w(X)_{i_0,j_1}}{\mathrm{d}x_{i_2,j_2}}$$

$$= \frac{\mathrm{d}f(X)_{i_0,i_0}}{\mathrm{d}x_{i_2,j_2}} \cdot w(X)_{i_0,j_1} + f(X)_{i_0,i_0} \cdot \frac{\mathrm{d}w(X)_{i_0,j_1}}{\mathrm{d}x_{i_2,j_2}}$$

$$= \frac{\mathrm{d}f(X)_{i_0,i_0}}{\mathrm{d}x_{i_2,j_2}} \cdot w(X)_{i_0,j_1} + f(X)_{i_0,i_0} \cdot w_{j_1,j_2}$$

$$= (-(\alpha(X)_{i_0})^{-1} \cdot f(X)_{i_0,i_0} \cdot (u(X)_{i_0,i_0} \cdot w(X)_{i_0,j_2} + \langle u(X)_{i_0}, X^\top W_{*,j_2}\rangle)$$
$$+ f(X)_{i_0,i_0} \cdot \langle W_{j_2,*} + W_{*,j_2}, X_{*,i_0}\rangle) \cdot w(X)_{i_0,j_1} + f(X)_{i_0,i_0} \cdot w_{j_1,j_2}$$

$$= (-f(X)_{i_0,i_0} \cdot (f(X)_{i_0,i_0} \cdot w(X)_{i_0,j_2} + \langle f(X)_{i_0}, X^\top W_{*,j_2}\rangle)$$
$$+ f(X)_{i_0,i_0} \cdot \langle W_{j_2,*} + W_{*,j_2}, X_{*,i_0}\rangle) \cdot w(X)_{i_0,j_1} + f(X)_{i_0,i_0} \cdot w_{j_1,j_2}$$

where step 1 is by Fact A.2, the 2nd step calls Lemma B.2, the 3rd step uses Lemma B.4, the last step is by the definition of $f(X)_{i_0}$ (see Definition A.6).

**Proof of Part 2**

$$\frac{\mathrm{d}f(X)_{i_0,i_0} \cdot w(X)_{i_0,j_1}}{\mathrm{d}x_{i_2,j_2}}$$

$$= \frac{\mathrm{d}f(X)_{i_0,i_0}}{\mathrm{d}x_{i_2,j_2}} \cdot w(X)_{i_0,j_1} + f(X)_{i_0,i_0} \cdot \frac{\mathrm{d}w(X)_{i_0,j_1}}{\mathrm{d}x_{i_2,j_2}}$$

$$= \frac{\mathrm{d}f(X)_{i_0,i_0}}{\mathrm{d}x_{i_2,j_2}} \cdot w(X)_{i_0,j_1}$$

$$= -(\alpha(X)_{i_0})^{-1} \cdot f(X)_{i_0,i_0} \cdot u(X)_{i_0,i_2} \cdot w(X)_{i_0,j_2} \cdot w(X)_{i_0,j_1}$$

$$= -f(X)_{i_0,i_0} \cdot f(X)_{i_0,i_2} \cdot w(X)_{i_0,j_2} \cdot w(X)_{i_0,j_1}$$

where step 1 is by Fact A.2, the 2nd step calls Lemma B.2, the 3rd step uses Lemma B.4, the last step is by the definition of $f(X)_{i_0}$ (see Definition A.6). □

### B.8 DERIVATIVE OF VECTOR FUNCTION $f(X)_{i_0} \circ (X^\top W_{*,j_1})$

**Lemma B.9.** *If the following holds:*

- *Let $f(X)_{i_0}$ be defined as Definition A.6*

*We have*

- **Part 1** *For $i_0 = i_1 = i_2 \in [n]$, $j_1, j_2 \in [d]$*

$$\frac{\mathrm{d}f(X)_{i_0} \circ (X^\top W_{*,j_1})}{\mathrm{d}x_{i_2,j_2}}$$

$$= (-f(X)_{i_0} \cdot (f(X)_{i_0,i_0} \cdot w(X)_{i_0,j_2} + \langle f(X)_{i_0}, X^\top W_{*,j_2}\rangle)$$
$$+ f(X)_{i_0} \circ (e_{i_0} \cdot w(X)_{i_0,j_2} + X^\top W_{*,j_2})) \circ (X^\top W_{*,j_1}) + f(X)_{i_0} \circ (e_{i_0} \cdot w_{j_2,j_1})$$

- **Part 2** *For $i_0 = i_1 \neq i_2 \in [n]$, $j_1, j_2 \in [d]$*

$$\frac{\mathrm{d}f(X)_{i_0} \circ (X^\top W_{*,j_1})}{\mathrm{d}x_{i_2,j_2}}$$

$$= (-f(X)_{i_0} \cdot f(X)_{i_0,i_2} \cdot w(X)_{i_0,j_2}$$
$$+ f(X)_{i_0} \circ (e_{i_2} \cdot w(X)_{i_0,j_2})) \circ (X^\top W_{*,j_1}) + f(X)_{i_0} \circ (e_{i_2} \cdot w_{j_2,j_1})$$

Table 2: $C_1$ Part 1 Summary

| ID | Term | Symmetric? | Table Name |
|---|---|---|---|
| 1 | $+2s(X)_{i_0,j_0} \cdot f(X)_{i_0,i_0}^2 \cdot w(X)_{i_0,j_1} \cdot w(X)_{i_0,j_2}$ | Yes | N/A |
| 2 | $-f(X)_{i_0,i_0}^2 \cdot h(X)_{j_0,i_0} \cdot w(X)_{i_0,j_2} \cdot w(X)_{i_0,j_1}$ | Yes | N/A |
| 3 | $-f(X)_{i_0,i_0} \cdot \langle f(X)_{i_0} \circ (X^\top W_{*,j_2}), h(X)_{j_0} \rangle \cdot w(X)_{i_0,j_1}$ | No | Table 5: 1 |
| 4 | $-f(X)_{i_0,i_0}^2 \cdot v_{j_2,j_0} \cdot w(X)_{i_0,j_1}$ | No | Table 6: 1 |
| 5 | $-s(X)_{i_0,j_0} \cdot f(X)_{i_0,i_0} \cdot w(X)_{i_0,j_2} \cdot w(X)_{i_0,j_1}$ | Yes | N/A |
| 6 | $-s(X)_{i_0,j_0} \cdot f(X)_{i_0,i_0} \cdot \langle W_{*,j_2}, X_{*,i_0} \rangle \cdot w(X)_{i_0,j_1}$ | No | Table 3: 7 |
| 7 | $-s(X)_{i_0,j_0} \cdot f(X)_{i_0,i_0} \cdot w_{j_1,j_2}$ | No | Table 3: 9 |
| 8 | $2f(X)_{i_0,i_0} \cdot s(X)_{i_0,j_0} \cdot z(X)_{i_0,j_2} \cdot w(X)_{i_0,j_1}$ | No | Table 3: 1 |

*Proof.* **Proof of Part 1**

$$\frac{\mathrm{d}f(X)_{i_0} \circ (X^\top W_{*,j_1})}{\mathrm{d}x_{i_2,j_2}}$$

$$= \frac{\mathrm{d}f(X)_{i_0}}{\mathrm{d}x_{i_2,j_2}} \circ (X^\top W_{*,j_1}) + f(X)_{i_0} \circ \frac{\mathrm{d}X^\top W_{*,j_1}}{\mathrm{d}x_{i_2,j_2}}$$

$$= \frac{\mathrm{d}f(X)_{i_0}}{\mathrm{d}x_{i_2,j_2}} \circ (X^\top W_{*,j_1}) + f(X)_{i_0} \circ (e_{i_0} \cdot w_{j_2,j_1})$$

$$= (-(\alpha(X)_{i_0})^{-1} \cdot f(X)_{i_0} \cdot (u(X)_{i_0,i_0} \cdot w(X)_{i_0,j_2} + \langle u(X)_{i_0}, X^\top W_{*,j_2} \rangle)$$
$$+ f(X)_{i_0} \circ (e_{i_0} \cdot w(X)_{i_0,j_2} + X^\top W_{*,j_2})) \circ (X^\top W_{*,j_1}) + f(X)_{i_0} \circ (e_{i_0} \cdot w_{j_2,j_1})$$

$$= (-f(X)_{i_0} \cdot (f(X)_{i_0,i_0} \cdot w(X)_{i_0,j_2} + \langle f(X)_{i_0}, X^\top W_{*,j_2} \rangle)$$
$$+ f(X)_{i_0} \circ (e_{i_0} \cdot w(X)_{i_0,j_2} + X^\top W_{*,j_2})) \circ (X^\top W_{*,j_1}) + f(X)_{i_0} \circ (e_{i_0} \cdot w_{j_2,j_1})$$

where the 1st step is by Fact A.2, the 2nd step uses Lemma B.3, the 3rd step uses Lemma A.14, the last step is by the definition of $f(X)_{i_0}$ (see Definition A.6).

**Proof of Part 2**

$$\frac{\mathrm{d}f(X)_{i_0} \circ (X^\top W_{*,j_1})}{\mathrm{d}x_{i_2,j_2}}$$

$$= \frac{\mathrm{d}f(X)_{i_0}}{\mathrm{d}x_{i_2,j_2}} \circ (X^\top W_{*,j_1}) + f(X)_{i_0} \circ \frac{\mathrm{d}X^\top W_{*,j_1}}{\mathrm{d}x_{i_2,j_2}}$$

$$= \frac{\mathrm{d}f(X)_{i_0}}{\mathrm{d}x_{i_2,j_2}} \circ (X^\top W_{*,j_1}) + f(X)_{i_0} \circ (e_{i_2} \cdot w_{j_2,j_1})$$

$$= -((\alpha(X)_{i_0})^{-1} \cdot f(X)_{i_0} \cdot u(X)_{i_0,i_2} \cdot w(X)_{i_0,j_2}$$
$$+ f(X)_{i_0} \circ (e_{i_2} \cdot w(X)_{i_0,j_2})) \circ (X^\top W_{*,j_1}) + f(X)_{i_0} \circ (e_{i_2} \cdot w_{j_2,j_1})$$

$$= (-f(X)_{i_0} \cdot f(X)_{i_0,i_2} \cdot w(X)_{i_0,j_2}$$
$$+ f(X)_{i_0} \circ (e_{i_2} \cdot w(X)_{i_0,j_2})) \circ (X^\top W_{*,j_1}) + f(X)_{i_0} \circ (e_{i_2} \cdot w_{j_2,j_1})$$

where the 1st step is by Fact A.2, the 2nd step uses Lemma B.3, the 3rd step uses Lemma A.14, the last step is by the definition of $f(X)_{i_0}$ (see Definition A.6). $\square$

### B.9 DERIVATIVE OF $C_1(X)$

**Lemma B.10.** *If the following holds:*

- *Let $C_1(X) \in \mathbb{R}$ be defined as in Lemma A.16*

- *Let $z(X)_{i_0,j_1} = \langle f(X)_{i_0}, X^\top W_{*,j_1} \rangle$*

- *Let $w(X)_{i_0,j_1} = \langle W_{j_1,*}, X_{*,i_0} \rangle$*

*We have*

- **Part 1** *For $i_0 = i_1 = i_2 \in [n]$, $j_1, j_2 \in [d]$*

$$
\frac{dC_1(X)}{dx_{i_2,j_2}}
$$
$$
\begin{aligned}
= &+ 2s(X)_{i_0,j_0} \cdot f(X)_{i_0,i_0}^2 \cdot w(X)_{i_0,j_2} \cdot w(X)_{i_0,j_1} \\
&+ 2f(X)_{i_0,i_0} \cdot s(X)_{i_0,j_0} \cdot z(X)_{i_0,j_2} \cdot w(X)_{i_0,j_1} \\
&- f(X)_{i_0,i_0}^2 \cdot h(X)_{j_0,i_0} \cdot w(X)_{i_0,j_2} \cdot w(X)_{i_0,j_1} \\
&- f(X)_{i_0,i_0} \cdot \langle f(X)_{i_0} \circ (X^\top W_{*,j_2}), h(X)_{j_0} \rangle \cdot w(X)_{i_0,j_1} \\
&- f(X)_{i_0,i_0}^2 \cdot v_{j_2,j_0} \cdot w(X)_{i_0,j_1} \\
&- s(X)_{i_0,j_0} \cdot f(X)_{i_0,i_0} \cdot w(X)_{i_0,j_2} \cdot w(X)_{i_0,j_1} \\
&- s(X)_{i_0,j_0} \cdot f(X)_{i_0,i_0} \cdot \langle W_{*,j_2}, X_{*,i_0} \rangle \cdot w(X)_{i_0,j_1} \\
&- s(X)_{i_0,j_0} \cdot f(X)_{i_0,i_0} \cdot w_{j_1,j_2}
\end{aligned}
$$

- **Part 2** *For $i_0 = i_1 \neq i_2 \in [n]$, $j_1, j_2 \in [d]$*

$$
\frac{dC_1(X)}{dx_{i_2,j_2}}
$$
$$
\begin{aligned}
= &\ s(X)_{i_0,j_0} \cdot f(X)_{i_0,i_2} \cdot w(X)_{i_0,j_2} \cdot f(X)_{i_0,i_0} \cdot w(X)_{i_0,j_1} \\
&- f(X)_{i_0,i_2} \cdot h(X)_{j_0,i_2} \cdot w(X)_{i_0,j_2} \cdot f(X)_{i_0,i_0} \cdot w(X)_{i_0,j_1} \\
&- f(X)_{i_0,i_2} \cdot v_{j_2,j_0} \cdot f(X)_{i_0,i_0} \cdot w(X)_{i_0,j_1} \\
&+ s(X)_{i_0,j_0} \cdot f(X)_{i_0,i_0} \cdot f(X)_{i_0,i_2} \cdot w(X)_{i_0,j_2} \cdot w(X)_{i_0,j_1}
\end{aligned}
$$

*Proof.* **Proof of Part 1**

$$
\frac{dC_1(X)}{dx_{i_2,j_2}}
$$
$$
= \frac{d - s(X)_{i_0,j_0} \cdot f(X)_{i_0,i_0} \cdot w(X)_{i_0,j_1}}{dx_{i_2,j_2}}
$$
$$
= -\frac{ds(X)_{i_0,j_0}}{dx_{i_2,j_2}} \cdot f(X)_{i_0,i_0} \cdot w(X)_{i_0,j_1}
$$
$$
- s(X)_{i_0,j_0} \cdot \frac{df(X)_{i_0,i_0} \cdot w(X)_{i_0,j_1}}{dx_{i_2,j_2}}
$$
$$
= -\frac{ds(X)_{i_0,j_0}}{dx_{i_2,j_2}} \cdot f(X)_{i_0,i_0} \cdot w(X)_{i_0,j_1}
$$
$$
- s(X)_{i_0,j_0} \cdot ((-(\alpha(X)_{i_0})^{-1} \cdot f(X)_{i_0,i_0} \cdot (u(X)_{i_0,i_0} \cdot w(X)_{i_0,j_2} + \langle u(X)_{i_0}, X^\top W_{*,j_2} \rangle)
$$
$$
+ f(X)_{i_0,i_0} \cdot \langle W_{j_2,*} + W_{*,j_2}, X_{*,i_0} \rangle) \cdot w(X)_{i_0,j_1} + f(X)_{i_0,i_0} \cdot w_{j_1,j_2})
$$
$$
= -(-s(X)_{i_0,j_0} \cdot f(X)_{i_0,i_0} \cdot w(X)_{i_0,j_2} - s(X)_{i_0,j_0} \cdot \langle f(X)_{i_0}, X^\top W_{*,j_2} \rangle
$$
$$
+ f(X)_{i_0,i_0} \cdot h(X)_{j_0,i_0} \cdot w(X)_{i_0,j_2}
$$
$$
+ \langle f(X)_{i_0} \circ (X^\top W_{*,j_2}), h(X)_{j_0} \rangle + f(X)_{i_0,i_2} \cdot v_{j_2,j_0}) \cdot f(X)_{i_0,i_0} \cdot w(X)_{i_0,j_1}
$$
$$
- s(X)_{i_0,j_0} \cdot ((-f(X)_{i_0,i_0} \cdot (f(X)_{i_0,i_0} \cdot w(X)_{i_0,j_2} + \langle f(X)_{i_0}, X^\top W_{*,j_2} \rangle)
$$
$$
+ f(X)_{i_0,i_0} \cdot \langle W_{j_2,*} + W_{*,j_2}, X_{*,i_0} \rangle) \cdot w(X)_{i_0,j_1} + f(X)_{i_0,i_0} \cdot w_{j_1,j_2})
$$
$$
= 2s(X)_{i_0,j_0} \cdot f(X)_{i_0,i_0}^2 \cdot w(X)_{i_0,j_2} \cdot w(X)_{i_0,j_1}
$$
$$
+ 2s(X)_{i_0,j_0} \cdot Z(X)_{i_0,j_2} \cdot f(X)_{i_0,i_0} \cdot w(X)_{i_0,j_1}
$$
$$
- f(X)_{i_0,i_0}^2 \cdot h(X)_{j_0,i_0} \cdot w(X)_{i_0,j_2} \cdot w(X)_{i_0,j_1}
$$
$$
- f(X)_{i_0,i_0} \cdot \langle f(X)_{i_0} \circ (X^\top W_{*,j_2}), h(X)_{j_0} \rangle \cdot w(X)_{i_0,j_1}
$$

Table 3: $C_2$ Part 1 Summary

| ID | Term | Symmetric Terms | Table Name |
|---|---|---|---|
| 1 | $2s(X)_{i_0,j_0} \cdot f(X)_{i_0,i_0} \cdot w(X)_{i_0,j_2} \cdot z(X)_{i_0,j_1}$ | No | Table 2: 9 |
| 2 | $s(X)_{i_0,j_0} \cdot z(X)_{i_0,j_2} \cdot z(X)_{i_0,j_1}$ | Yes | N/A |
| 3 | $-f(X)_{i_0,i_0} \cdot h(X)_{j_0,i_0} \cdot w(X)_{i_0,j_2} \cdot z(X)_{i_0,j_1}$ | No | Table 4: 3 |
| 4 | $-\langle f(X)_{i_0} \circ (X^\top W_{*,j_2}), h(X)_{j_0} \rangle \cdot z(X)_{i_0,j_1}$ | No | Table 5: 2 |
| 5 | $-f(X)_{i_0,i_0} \cdot v_{j_2,j_0} \cdot z(X)_{i_0,j_1}$ | No | Table 6: 2 |
| 6 | $+s(X)_{i_0,j_0} \cdot z(X)_{i_0,j_1} \cdot f(X)_{i_0,i_0} \cdot z(X)_{i_0,j_2}$ | Yes | N/A |
| 7 | $-s(X)_{i_0,j_0} \cdot f(X)_{i_0,i_0} \cdot \langle W_{*,j_1}, X_{*,i_0} \rangle \cdot w(X)_{i_0,j_2}$ | No | Table 2: 6 |
| 8 | $-s(X)_{i_0,j_0} \cdot \langle f(X)_{i_0} \circ (X^\top W_{*,j_2}), X^\top W_{*,j_1} \rangle$ | Yes | N/A |
| 9 | $-s(X)_{i_0,j_0} \cdot f(X)_{i_0,i_0} \cdot w_{j_2,j_1}$ | No | Table 2: 7 |

$$- f(X)_{i_0,i_0}^2 \cdot v_{j_2,j_0} \cdot w(X)_{i_0,j_1}$$
$$- s(X)_{i_0,j_0} \cdot f(X)_{i_0,i_0} \cdot \langle W_{j_2,*} + W_{*,j_2}, X_{*,i_0} \rangle \cdot w(X)_{i_0,j_1}$$
$$- s(X)_{i_0,j_0} \cdot f(X)_{i_0,i_0} \cdot w_{j_1,j_2}$$

where the first step is by definition of $C_1(X)$ (see Lemma A.16), the 2nd step is by Fact A.2, the 3rd step is by Lemma B.8, the 4th step is because Lemma A.16, the 5th step is a rearrangement.

**Proof of Part 2**

$$\frac{\mathrm{d}C_1(X)}{\mathrm{d}x_{i_2,j_2}}$$

$$= \frac{\mathrm{d} - s(X)_{i_0,j_0} \cdot f(X)_{i_0,i_0} \cdot w(X)_{i_0,j_1}}{\mathrm{d}x_{i_2,j_2}}$$

$$= -\frac{\mathrm{d}s(X)_{i_0,j_0}}{\mathrm{d}x_{i_2,j_2}} \cdot f(X)_{i_0,i_0} \cdot w(X)_{i_0,j_1}$$
$$- s(X)_{i_0,j_0} \cdot \frac{\mathrm{d}f(X)_{i_0,i_0} \cdot w(X)_{i_0,j_1}}{\mathrm{d}x_{i_2,j_2}}$$

$$= -\frac{\mathrm{d}s(X)_{i_0,j_0}}{\mathrm{d}x_{i_2,j_2}} \cdot f(X)_{i_0,i_0} \cdot w(X)_{i_0,j_1}$$
$$+ s(X)_{i_0,j_0} \cdot f(X)_{i_0,i_0} \cdot f(X)_{i_0,i_2} \cdot w(X)_{i_0,j_2} \cdot w(X)_{i_0,j_1}$$

$$= -(-s(X)_{i_0,j_0} \cdot f(X)_{i_0,i_2} \cdot w(X)_{i_0,j_2} + f(X)_{i_0,i_2} \cdot h(X)_{j_0,i_2} \cdot w(X)_{i_0,j_2}$$
$$+ f(X)_{i_0,i_2} \cdot v_{j_2,j_0}) \cdot f(X)_{i_0,i_0} \cdot w(X)_{i_0,j_1}$$
$$+ s(X)_{i_0,j_0} \cdot f(X)_{i_0,i_0} \cdot f(X)_{i_0,i_2} \cdot w(X)_{i_0,j_2} \cdot w(X)_{i_0,j_1}$$

$$= s(X)_{i_0,j_0} \cdot f(X)_{i_0,i_2} \cdot w(X)_{i_0,j_2} \cdot f(X)_{i_0,i_0} \cdot w(X)_{i_0,j_1}$$
$$- f(X)_{i_0,i_2} \cdot h(X)_{j_0,i_2} \cdot w(X)_{i_0,j_2} \cdot f(X)_{i_0,i_0} \cdot w(X)_{i_0,j_1}$$
$$- f(X)_{i_0,i_2} \cdot v_{j_2,j_0} \cdot f(X)_{i_0,i_0} \cdot w(X)_{i_0,j_1}$$
$$+ s(X)_{i_0,j_0} \cdot f(X)_{i_0,i_0} \cdot f(X)_{i_0,i_2} \cdot w(X)_{i_0,j_2} \cdot w(X)_{i_0,j_1}$$

where the first step is by definition of $C_1(X)$ (see Lemma A.16), the 2nd step is by Fact A.2, the 3rd step is by Lemma B.8, the 4th step is because Lemma A.16, the 5th step is a rearrangement. □

## B.10 DERIVATIVE OF $C_2(X)$

**Lemma B.11.** *If the following holds:*

- *Let $C_2(X)$ be defined as in Lemma A.16*

- *We define $z(X)_{i_0,j_1} := \langle f(X)_{i_0}, X^\top W_{*,j_1} \rangle$.*

*We have*

- **Part 1** *For $i_0 = i_1 = i_2 \in [n]$, $j_1, j_2 \in [d]$*

$$\frac{\mathrm{d}C_2(X)}{\mathrm{d}x_{i_2,j_2}}$$
$$= + 2s(X)_{i_0,j_0} \cdot f(X)_{i_0,i_0} \cdot w(X)_{i_0,j_2} \cdot z(X)_{i_0,j_1}$$
$$+ s(X)_{i_0,j_0} \cdot z(X)_{i_0,j_2} \cdot z(X)_{i_0,j_1}$$
$$- f(X)_{i_0,i_0} \cdot h(X)_{j_0,i_0} \cdot w(X)_{i_0,j_2} \cdot z(X)_{i_0,j_1}$$
$$- \langle f(X)_{i_0} \circ (X^\top W_{*,j_2}), h(X)_{j_0} \rangle \cdot z(X)_{i_0,j_1}$$
$$- f(X)_{i_0,i_0} \cdot v_{j_2,j_0} \cdot z(X)_{i_0,j_1}$$
$$+ s(X)_{i_0,j_0} \cdot z(X)_{i_0,j_1} \cdot f(X)_{i_0,i_0} \cdot z(X)_{i_0,j_2}$$
$$- s(X)_{i_0,j_0} \cdot f(X)_{i_0,i_0} \cdot \langle W_{*,j_1}, X_{*,i_0} \rangle \cdot w(X)_{i_0,j_2}$$
$$- s(X)_{i_0,j_0} \cdot \langle f(X)_{i_0} \circ (X^\top W_{*,j_2}), X^\top W_{*,j_1} \rangle$$
$$- s(X)_{i_0,j_0} \cdot f(X)_{i_0,i_0} \cdot w_{j_2,j_1}$$

- **Part 2** *For $i_0 = i_1 \neq i_2 \in [n]$, $j_1, j_2 \in [d]$*

$$\frac{\mathrm{d}C_2(X)}{\mathrm{d}x_{i_2,j_2}}$$
$$= + s(X)_{i_0,j_0} \cdot f(X)_{i_0,i_2} \cdot w(X)_{i_0,j_2} \cdot z(X)_{i_0,j_1}$$
$$- f(X)_{i_0,i_2} \cdot h(X)_{j_0,i_2} \cdot w(X)_{i_0,j_2} \cdot z(X)_{i_0,j_1}$$
$$- f(X)_{i_0,i_2} \cdot v_{j_2,j_0} \cdot z(X)_{i_0,j_1}$$
$$+ s(X)_{i_0,j_0} \cdot \langle f(X)_{i_0}, X^\top W_{*,j_1} \rangle \cdot f(X)_{i_0,i_0} \cdot w(X)_{i_0,j_2}$$
$$- s(X)_{i_0,j_0} \cdot f(X)_{i_0,i_0} \cdot \langle W_{*,j_1}, X_{*,i_0} \rangle \cdot w(X)_{i_0,j_2}$$
$$- s(X)_{i_0,j_0} \cdot f(X)_{i_0,i_0} \cdot w_{j_2,j_1}$$

*Proof.* **Proof of Part 1**

$$\frac{\mathrm{d} - C_2(X)}{\mathrm{d}x_{i_2,j_2}}$$
$$= \frac{\mathrm{d}s(X)_{i_0,j_0} \cdot z(X)_{i_0,j_1}}{\mathrm{d}x_{i_2,j_2}}$$
$$= \frac{\mathrm{d}s(X)_{i_0,j_0}}{\mathrm{d}x_{i_2,j_2}} \cdot z(X)_{i_0,j_1} + s(X)_{i_0,j_0} \cdot \frac{\mathrm{d}z(X)_{i_0,j_1}}{\mathrm{d}x_{i_2,j_2}}$$
$$= \frac{\mathrm{d}s(X)_{i_0,j_0}}{\mathrm{d}x_{i_2,j_2}} \cdot z(X)_{i_0,j_1}$$
$$+ s(X)_{i_0,j_0} \cdot (\langle -(\alpha(X)_{i_0})^{-1} \cdot f(X)_{i_0} \cdot (u(X)_{i_0,i_0} \cdot w(X)_{i_0,j_2} + \langle u(X)_{i_0}, X^\top W_{*,j_2} \rangle)$$
$$+ f(X)_{i_0} \circ (e_{i_0} \cdot w(X)_{i_0,j_2} + X^\top W_{*,j_2}), X^\top W_{*,j_1} \rangle + f(X)_{i_0,i_0} \cdot w_{j_2,j_1})$$
$$= (-s(X)_{i_0,j_0} \cdot f(X)_{i_0,i_0} \cdot w(X)_{i_0,j_2} - s(X)_{i_0,j_0} \cdot \langle f(X)_{i_0}, X^\top W_{*,j_2} \rangle$$
$$+ f(X)_{i_0,i_0} \cdot h(X)_{j_0,i_0} \cdot w(X)_{i_0,j_2}$$
$$+ \langle f(X)_{i_0} \circ (X^\top W_{*,j_2}), h(X)_{j_0} \rangle + f(X)_{i_0,i_2} \cdot v_{j_2,j_0}) \cdot z(X)_{i_0,j_1}$$
$$+ s(X)_{i_0,j_0} \cdot (\langle -(\alpha(X)_{i_0})^{-1} \cdot f(X)_{i_0} \cdot (u(X)_{i_0,i_0} \cdot w(X)_{i_0,j_2} + \langle u(X)_{i_0}, X^\top W_{*,j_2} \rangle)$$
$$+ f(X)_{i_0} \circ (e_{i_0} \cdot w(X)_{i_0,j_2} + X^\top W_{*,j_2}), X^\top W_{*,j_1} \rangle + f(X)_{i_0,i_0} \cdot w_{j_2,j_1})$$
$$= - s(X)_{i_0,j_0} \cdot f(X)_{i_0,i_0} \cdot w(X)_{i_0,j_2} \cdot z(X)_{i_0,j_1}$$
$$- s(X)_{i_0,j_0} \cdot z(X)_{i_0,j_2} \cdot z(X)_{i_0,j_1}$$
$$+ f(X)_{i_0,i_0} \cdot h(X)_{j_0,i_0} \cdot w(X)_{i_0,j_2} \cdot z(X)_{i_0,j_1}$$
$$+ \langle f(X)_{i_0} \circ (X^\top W_{*,j_2}), h(X)_{j_0} \rangle \cdot z(X)_{i_0,j_1}$$
$$+ f(X)_{i_0,i_2} \cdot v_{j_2,j_0} \cdot z(X)_{i_0,j_1}$$
$$- s(X)_{i_0,j_0} \cdot \langle f(X)_{i_0}, X^\top W_{*,j_1} \rangle \cdot f(X)_{i_0,i_0} \cdot w(X)_{i_0,j_2}$$

Table 4: $C_3$ Part 1 Summary

| ID | Term | Symmetric Terms | Table Name |
|---|---|---|---|
| 1 | $-f(X)_{i_0,i_0}^2 \cdot h(X)_{j_0,i_0} \cdot w(X)_{i_0,j_2} \cdot w(X)_{i_0,j_1}$ | Yes | N/A |
| 2 | $f(X)_{i_0,i_0} \cdot w(X)_{i_0,j_2} \cdot h(X)_{j_0,i_0} \cdot w(X)_{i_0,j_1}$ | Yes | N/A |
| 3 | $-f(X)_{i_0,i_0} \cdot z(X)_{i_0,j_2} \cdot h(X)_{j_0,i_0} \cdot w(X)_{i_0,j_1}$ | No | Table 3: 3 |
| 4 | $f(X)_{i_0,i_0} \cdot \langle W_{*,j_2}, X_{*,i_0} \rangle \cdot h(X)_{j_0,i_0} \cdot w(X)_{i_0,j_1}$ | No | Table 5: 3 |
| 5 | $f(X)_{i_0,i_0} \cdot v_{j_2,j_0} \cdot w(X)_{i_0,j_1}$ | No | Table 6: 3 |
| 6 | $f(X)_{i_0,i_0} \cdot h(X)_{i_0,i_0} \cdot w_{j_1,j_2}$ | No | Table 5: 5 |

$$- s(X)_{i_0,j_0} \cdot \langle f(X)_{i_0}, X^\top W_{*,j_1} \rangle \cdot f(X)_{i_0,i_0} \cdot \langle f(X)_{i_0}, X^\top W_{*,j_2} \rangle$$
$$+ s(X)_{i_0,j_0} \cdot f(X)_{i_0,i_0} \cdot \langle W_{*,j_1}, X_{*,i_0} \rangle \cdot w(X)_{i_0,j_2}$$
$$+ s(X)_{i_0,j_0} \cdot \langle f(X)_{i_0} \circ (X^\top W_{*,j_2}), X^\top W_{*,j_1} \rangle$$
$$+ s(X)_{i_0,j_0} \cdot f(X)_{i_0,i_0} \cdot w_{j_2,j_1}$$

where the first step is by definition of $C_2(X)$ (see Lemma A.16), the 2nd step is by Fact A.2, the 3rd step is by Lemma B.6, the 4th step is because Lemma A.16, the 5th step is a rearrangement.

**Proof of Part 2**

$$\frac{\mathrm{d} - C_2(X)}{\mathrm{d}x_{i_2,j_2}}$$
$$= \frac{\mathrm{d}s(X)_{i_0,j_0} \cdot \langle f(X)_{i_0}, X^\top W_{*,j_1} \rangle}{\mathrm{d}x_{i_2,j_2}}$$
$$= \frac{\mathrm{d}s(X)_{i_0,j_0}}{\mathrm{d}x_{i_2,j_2}} \cdot z(X)_{i_0,j_1} + s(X)_{i_0,j_0} \cdot \frac{\mathrm{d}\langle f(X)_{i_0}, X^\top W_{*,j_1} \rangle}{\mathrm{d}x_{i_2,j_2}}$$
$$= \frac{\mathrm{d}s(X)_{i_0,j_0}}{\mathrm{d}x_{i_2,j_2}} \cdot z(X)_{i_0,j_1}$$
$$\quad + s(X)_{i_0,j_0} \cdot ((\langle -(\alpha(X)_{i_0})^{-1} \cdot f(X)_{i_0} \cdot u(X)_{i_0,i_0} \cdot w(X)_{i_0,j_2}$$
$$\quad + f(X)_{i_0} \circ (e_{i_0} \cdot w(X)_{i_0,j_2}), X^\top W_{*,j_1} \rangle + f(X)_{i_0,i_0} \cdot w_{j_2,j_1})$$
$$= (-s(X)_{i_0,j_0} \cdot f(X)_{i_0,i_2} \cdot w(X)_{i_0,j_2} + f(X)_{i_0,i_2} \cdot h(X)_{j_0,i_2} \cdot w(X)_{i_0,j_2}$$
$$\quad + f(X)_{i_0,i_2} \cdot v_{j_2,j_0}) \cdot z(X)_{i_0,j_1}$$
$$\quad + s(X)_{i_0,j_0} \cdot ((\langle -(\alpha(X)_{i_0})^{-1} \cdot f(X)_{i_0} \cdot u(X)_{i_0,i_0} \cdot w(X)_{i_0,j_2}$$
$$\quad + f(X)_{i_0} \circ (e_{i_0} \cdot w(X)_{i_0,j_2}), X^\top W_{*,j_1} \rangle + f(X)_{i_0,i_0} \cdot w_{j_2,j_1})$$
$$= -s(X)_{i_0,j_0} \cdot f(X)_{i_0,i_2} \cdot w(X)_{i_0,j_2} \cdot z(X)_{i_0,j_1}$$
$$\quad + f(X)_{i_0,i_2} \cdot h(X)_{j_0,i_2} \cdot w(X)_{i_0,j_2} \cdot z(X)_{i_0,j_1}$$
$$\quad + f(X)_{i_0,i_2} \cdot v_{j_2,j_0} \cdot z(X)_{i_0,j_1}$$
$$\quad - s(X)_{i_0,j_0} \cdot \langle f(X)_{i_0}, X^\top W_{*,j_1} \rangle \cdot f(X)_{i_0,i_0} \cdot w(X)_{i_0,j_2}$$
$$\quad + s(X)_{i_0,j_0} \cdot f(X)_{i_0,i_0} \cdot \langle W_{*,j_1}, X_{*,i_0} \rangle \cdot w(X)_{i_0,j_2}$$
$$\quad + s(X)_{i_0,j_0} \cdot f(X)_{i_0,i_0} \cdot w_{j_2,j_1}$$

where the first step is by definition of $C_2(X)$ (see Lemma A.16), the 2nd step is by Fact A.2, the 3rd step is by Lemma B.6, the 4th step is because Lemma A.16, the 5th step is a rearrangement. $\square$

### B.11 DERIVATIVE OF $C_3(X)$

**Lemma B.12.** *If the following holds:*

- *Let $C_3(X)$ be defined as in Lemma A.16*

*We have*

- **Part 1** *For $i_0 = i_1 = i_2 \in [n]$, $j_1, j_2 \in [d]$*

$$\frac{\mathrm{d}C_3(X)}{\mathrm{d}x_{i_2,j_2}}$$

$$\begin{aligned}
= &- f(X)_{i_0,i_0}^2 \cdot h(X)_{j_0,i_0} \cdot w(X)_{i_0,j_2} \cdot w(X)_{i_0,j_1} \\
&- f(X)_{i_0,i_0} \cdot z(X)_{i_0,j_2} \cdot h(X)_{j_0,i_0} \cdot w(X)_{i_0,j_1} \\
&+ f(X)_{i_0,i_0} \cdot w(X)_{i_0,j_2} \cdot h(X)_{j_0,i_0} \cdot w(X)_{i_0,j_1} \\
&+ f(X)_{i_0,i_0} \cdot \langle W_{*,j_2}, X_{*,i_0}\rangle \cdot h(X)_{j_0,i_0} \cdot w(X)_{i_0,j_1} \\
&+ f(X)_{i_0,i_0} \cdot v_{j_2,j_0} \cdot w(X)_{i_0,j_1} \\
&+ f(X)_{i_0,i_0} \cdot h(X)_{i_0,i_0} \cdot w_{j_1,j_2}
\end{aligned}$$

- **Part 2** *For $i_0 = i_1 \neq i_2 \in [n]$, $j_1, j_2 \in [d]$*

$$\frac{\mathrm{d}C_3(X)}{\mathrm{d}x_{i_2,j_2}}$$

$$= - f(X)_{i_0,i_0} \cdot f(X)_{i_0,i_2} \cdot w(X)_{i_0,j_2} \cdot h(X)_{j_0,i_0} \cdot w(X)_{i_0,j_1}$$

*Proof.* **Proof of Part 1**

$$\frac{\mathrm{d}C_3(X)}{\mathrm{d}x_{i_2,j_2}}$$

$$= \frac{\mathrm{d}f(X)_{i_0,i_0} \cdot h(X)_{i_0,i_0} \cdot w(X)_{i_0,j_1}}{\mathrm{d}x_{i_2,j_2}}$$

$$= \frac{\mathrm{d}f(X)_{i_0,i_0} \cdot h(X)_{i_0,i_0}}{\mathrm{d}x_{i_2,j_2}} \cdot w(X)_{i_0,j_1} + f(X)_{i_0,i_0} \cdot h(X)_{i_0,i_0} \cdot \frac{\mathrm{d}w(X)_{i_0,j_1}}{\mathrm{d}x_{i_2,j_2}}$$

$$= \frac{\mathrm{d}f(X)_{i_0,i_0} \cdot h(X)_{i_0,i_0}}{\mathrm{d}x_{i_2,j_2}} \cdot w(X)_{i_0,j_1} + f(X)_{i_0,i_0} \cdot h(X)_{i_0,i_0} \cdot w_{j_1,j_2}$$

$$\begin{aligned}
= &((-f(X)_{i_0,i_0} \cdot (f(X)_{i_0,i_0} \cdot w(X)_{i_0,j_2} + \langle f(X)_{i_0}, X^\top W_{*,j_2}\rangle) \\
&+ f(X)_{i_0,i_0} \cdot \langle W_{j_2,*} + W_{*,j_2}, X_{*,i_0}\rangle) \cdot h(X)_{j_0,i_0} + f(X)_{i_0,i_0} \cdot v_{j_2,j_0}) \cdot w(X)_{i_0,j_1} \\
&+ f(X)_{i_0,i_0} \cdot h(X)_{i_0,i_0} \cdot w_{j_1,j_2}
\end{aligned}$$

$$\begin{aligned}
= &- f(X)_{i_0,i_0}^2 \cdot h(X)_{j_0,i_0} \cdot w(X)_{i_0,j_2} \cdot w(X)_{i_0,j_1} \\
&- f(X)_{i_0,i_0} \cdot Z(X)_{i_0,j_2} \cdot h(X)_{j_0,i_0} \cdot w(X)_{i_0,j_1} \\
&+ f(X)_{i_0,i_0} \cdot \langle W_{j_2,*} + W_{*,j_2}, X_{*,i_0}\rangle \cdot h(X)_{j_0,i_0} \cdot w(X)_{i_0,j_1} \\
&+ f(X)_{i_0,i_0} \cdot v_{j_2,j_0} \cdot w(X)_{i_0,j_1} \\
&+ f(X)_{i_0,i_0} \cdot h(X)_{i_0,i_0} \cdot w_{j_1,j_2}
\end{aligned}$$

where the first step is by definition of $C_3(X)$ (see Lemma A.16), the 2nd step is by Fact A.2, the 3rd step is by Lemma B.2, the 4th step is because Lemma B.7, the 5th step is a rearrangement.

**Proof of Part 2**

$$\frac{\mathrm{d}C_3(X)}{\mathrm{d}x_{i_2,j_2}}$$

$$= \frac{\mathrm{d}f(X)_{i_0,i_0} \cdot h(X)_{i_0,i_0} \cdot w(X)_{i_0,j_1}}{\mathrm{d}x_{i_2,j_2}}$$

$$= \frac{\mathrm{d}f(X)_{i_0,i_0} \cdot h(X)_{i_0,i_0}}{\mathrm{d}x_{i_2,j_2}} \cdot w(X)_{i_0,j_1} + f(X)_{i_0,i_0} \cdot h(X)_{i_0,i_0} \cdot \frac{\mathrm{d}w(X)_{i_0,j_1}}{\mathrm{d}x_{i_2,j_2}}$$

$$= \frac{\mathrm{d}f(X)_{i_0,i_0} \cdot h(X)_{i_0,i_0}}{\mathrm{d}x_{i_2,j_2}} \cdot w(X)_{i_0,j_1}$$

$$= - f(X)_{i_0,i_0} \cdot f(X)_{i_0,i_2} \cdot w(X)_{i_0,j_2} \cdot h(X)_{j_0,i_0} \cdot w(X)_{i_0,j_1}$$

where the first step is by definition of $C_3(X)$ (see Lemma A.16), the 2nd step is by Fact A.2, the 3rd step is by Lemma B.2, the 4th step is because Lemma B.7, the 5th step is a rearrangement.

$\square$

Table 5: $C_4$ Part 1 Summary

| ID | Term | Symmetric? | Table Name |
|----|------|------------|------------|
| 1 | $-\langle f(X)_{i_0} \circ (X^\top W_{*,j_1}), h(X)_{j_0}\rangle \cdot f(X)_{i_0,i_0} \cdot w(X)_{i_0,j_2}$ | No | Table 2: 3 |
| 2 | $-\langle f(X)_{i_0} \circ (X^\top W_{*,j_1}), h(X)_{j_0}\rangle \cdot Z(X)_{i_0,j_2}$ | No | Table 3: 4 |
| 3 | $f(X)_{i_0,i_0} \cdot h(X)_{j_0,i_0} \cdot \langle W_{*,j_1}, X_{*,i_0}\rangle \cdot w(X)_{i_0,j_2}$ | No | Table 4: 4 |
| 4 | $\langle f(X)_{i_0} \circ (X^\top W_{*,j_2}) \circ (X^\top W_{*,j_1}), h(X)_{j_0}\rangle$ | Yes | N/A |
| 5 | $f(X)_{i_0,i_0} \cdot h(X)_{j_0,i_0} \cdot w_{j_2,j_1}$ | No | Table 4: 6 |
| 6 | $f(X)_{i_0,i_0} \cdot \langle W_{*,j_1}, X_{*,i_0}\rangle \cdot v_{j_2,j_0}$ | No | Table 6:4 |

## B.12 DERIVATIVE OF $C_4(X)$

**Lemma B.13.** *If the following holds:*

- *Let $C_4(X)$ be defined as in Lemma A.16*

*We have*

- **Part 1** *For $i_0 = i_1 = i_2 \in [n]$, $j_1, j_2 \in [d]$*

$$\frac{\mathrm{d}C_4(X)}{\mathrm{d}x_{i_2,j_2}}$$
$$= -\langle f(X)_{i_0} \circ (X^\top W_{*,j_1}), h(X)_{j_0}\rangle \cdot f(X)_{i_0,i_0} \cdot w(X)_{i_0,j_2}$$
$$-\langle f(X)_{i_0} \circ (X^\top W_{*,j_1}), h(X)_{j_0}\rangle \cdot Z(X)_{i_0,j_2}$$
$$+ f(X)_{i_0,i_0} \cdot h(X)_{j_0,i_0} \cdot \langle W_{*,j_1}, X_{*,i_0}\rangle \cdot w(X)_{i_0,j_2}$$
$$+ \langle f(X)_{i_0} \circ (X^\top W_{*,j_2}) \circ (X^\top W_{*,j_1}), h(X)_{j_0}\rangle$$
$$+ f(X)_{i_0,i_0} \cdot h(X)_{j_0,i_0} \cdot w_{j_2,j_1}$$
$$+ f(X)_{i_0,i_0} \cdot \langle W_{*,j_1}, X_{*,i_0}\rangle \cdot v_{j_2,j_0}$$

- **Part 2** *For $i_0 = i_1 \neq i_2 \in [n]$, $j_1, j_2 \in [d]$*

$$\frac{\mathrm{d}C_4(X)}{\mathrm{d}x_{i_2,j_2}}$$
$$= -\langle f(X)_{i_0} \circ (X^\top W_{*,j_1}), h(X)_{j_0}\rangle \cdot f(X)_{i_0,i_2} \cdot w(X)_{i_0,j_2}$$
$$+ f(X)_{i_0,i_2} \cdot h(X)_{j_0,i_2} \cdot \langle W_{*,j_1}, X_{*,i_2}\rangle \cdot w(X)_{i_0,j_2}$$
$$+ f(X)_{i_0,i_2} \cdot h(X)_{j_0,i_2} \cdot w_{j_2,j_1}$$
$$+ f(X)_{i_0,i_2} \cdot \langle W_{*,j_1}, X_{*,i_2}\rangle \cdot v_{j_2,j_0}$$

*Proof.* **Proof of Part 1**

$$\frac{\mathrm{d}C_4(X)}{\mathrm{d}x_{i_2,j_2}}$$
$$= \frac{\mathrm{d}\langle f(X)_{i_0} \circ (X^\top W_{*,j_1}), h(X)_{j_0}\rangle}{\mathrm{d}x_{i_2,j_2}}$$
$$= \langle \frac{\mathrm{d}f(X)_{i_0} \circ (X^\top W_{*,j_1})}{\mathrm{d}x_{i_2,j_2}}, h(X)_{j_0}\rangle + \langle f(X)_{i_0} \circ (X^\top W_{*,j_1}), \frac{\mathrm{d}h(X)_{j_0}}{\mathrm{d}x_{i_2,j_2}}\rangle$$
$$= \langle \frac{\mathrm{d}f(X)_{i_0} \circ (X^\top W_{*,j_1})}{\mathrm{d}x_{i_2,j_2}}, h(X)_{j_0}\rangle + \langle f(X)_{i_0} \circ (X^\top W_{*,j_1}), e_{i_2} \cdot v_{j_2,j_0}\rangle$$
$$= \langle (-f(X)_{i_0} \cdot (f(X)_{i_0,i_0} \cdot w(X)_{i_0,j_2} + \langle f(X)_{i_0}, X^\top W_{*,j_2}\rangle)$$
$$+ f(X)_{i_0} \circ (e_{i_0} \cdot w(X)_{i_0,j_2} + X^\top W_{*,j_2})) \circ (X^\top W_{*,j_1}) + f(X)_{i_0} \circ (e_{i_0} \cdot w_{j_2,j_1}), h(X)_{j_0}\rangle$$

Table 6: $C_5$ Part 1 Summary

| Term | Symmetric Terms | Table Name |
|---|---|---|
| $-f(X)^2_{i_0,i_0} \cdot w(X)_{i_0,j_2} \cdot v_{j_1,j_0}$ | No | $C_1(X) : 4$ |
| $-f(X)_{i_0,i_0} \cdot z(X)_{i_0,j_2} \cdot v_{j_1,j_0}$ | No | Table 3: 5 |
| $f(X)_{i_0,i_0} \cdot w(X)_{i_0,j_2} \cdot v_{j_1,j_0}$ | No | Table 4:5 |
| $f(X)_{i_0,i_0} \cdot \langle W_{*,j_2}, X_{*,i_0} \rangle \cdot v_{j_1,j_0}$ | No | Table 5: 6 |

$$+ \langle f(X)_{i_0} \circ (X^\top W_{*,j_1}), e_{i_0} \cdot v_{j_2,j_0} \rangle$$
$$= - \langle f(X)_{i_0} \circ (X^\top W_{*,j_1}), h(X)_{j_0} \rangle \cdot f(X)_{i_0,i_0} \cdot w(X)_{i_0,j_2}$$
$$- \langle f(X)_{i_0} \circ (X^\top W_{*,j_1}), h(X)_{j_0} \rangle \cdot \langle f(X)_{i_0}, X^\top W_{*,j_2} \rangle$$
$$+ f(X)_{i_0,i_0} \cdot h(X)_{j_0,i_0} \cdot \langle W_{*,j_1}, X_{*,i_0} \rangle \cdot w(X)_{i_0,j_2}$$
$$+ \langle f(X)_{i_0} \circ (X^\top W_{*,j_2}) \circ (X^\top W_{*,j_1}), h(X)_{j_0} \rangle$$
$$+ f(X)_{i_0,i_0} \cdot h(X)_{j_0,i_0} \cdot w_{j_2,j_1}$$
$$+ f(X)_{i_0,i_0} \cdot \langle W_{*,j_1}, X_{*,i_0} \rangle \cdot v_{j_2,j_0}$$

where the first step is by definition of $C_4(X)$ (see Lemma A.16), the 2nd step is by Fact A.2, the 3rd step is by Lemma A.15, the 4th step is because Lemma B.9, the 5th step is a rearrangement.

**Proof of Part 2**

$$\frac{\mathrm{d}C_4(X)}{\mathrm{d}x_{i_2,j_2}}$$
$$= \frac{\mathrm{d}\langle f(X)_{i_0} \circ (X^\top W_{*,j_1}), h(X)_{j_0} \rangle}{\mathrm{d}x_{i_2,j_2}}$$
$$= \langle \frac{\mathrm{d}f(X)_{i_0} \circ (X^\top W_{*,j_1})}{\mathrm{d}x_{i_2,j_2}}, h(X)_{j_0} \rangle + \langle f(X)_{i_0} \circ (X^\top W_{*,j_1}), \frac{\mathrm{d}h(X)_{j_0}}{\mathrm{d}x_{i_2,j_2}} \rangle$$
$$= \langle \frac{\mathrm{d}f(X)_{i_0} \circ (X^\top W_{*,j_1})}{\mathrm{d}x_{i_2,j_2}}, h(X)_{j_0} \rangle + \langle f(X)_{i_0} \circ (X^\top W_{*,j_1}), e_{i_2} \cdot v_{j_2,j_0} \rangle$$
$$= \langle -(f(X)_{i_0} \cdot f(X)_{i_0,i_2} \cdot w(X)_{i_0,j_2}$$
$$\quad + f(X)_{i_0} \circ (e_{i_2} \cdot w(X)_{i_0,j_2})) \circ (X^\top W_{*,j_1}) + f(X)_{i_0} \circ (e_{i_2} \cdot w_{j_2,j_1}), h(X)_{j_0} \rangle$$
$$\quad + \langle f(X)_{i_0} \circ (X^\top W_{*,j_1}), e_{i_2} \cdot v_{j_2,j_0} \rangle$$
$$= - \langle f(X)_{i_0} \circ (X^\top W_{*,j_1}), h(X)_{j_0} \rangle \cdot f(X)_{i_0,i_2} \cdot w(X)_{i_0,j_2}$$
$$\quad + f(X)_{i_0,i_2} \cdot h(X)_{j_0,i_2} \cdot \langle W_{*,j_1}, X_{*,i_2} \rangle \cdot w(X)_{i_0,j_2}$$
$$\quad + f(X)_{i_0,i_2} \cdot h(X)_{j_0,i_2} \cdot w_{j_2,j_1}$$
$$\quad + f(X)_{i_0,i_2} \cdot \langle W_{*,j_1}, X_{*,i_2} \rangle \cdot v_{j_2,j_0}$$

where the first step is by definition of $C_4(X)$ (see Lemma A.16), the 2nd step is by Fact A.2, the 3rd step is by Lemma A.15, the 4th step is because Lemma B.9, the 5th step is a rearrangement.

$\square$

## B.13 DERIVATIVE OF $C_5(X)$

**Lemma B.14.** *If the following holds:*

- *Let $C_5(X)$ be defined as in Lemma A.16*

*We have*

- **Part 1** *For $i_0 = i_1 = i_2 \in [n]$, $j_1, j_2 \in [d]$*

$$\frac{\mathrm{d}C_5(X)}{\mathrm{d}x_{i_2,j_2}} = - f(X)^2_{i_0,i_0} \cdot w(X)_{i_0,j_2} \cdot v_{j_1,j_0}$$

$$- f(X)_{i_0,i_0} \cdot z(X)_{i_0,j_2} \cdot v_{j_1,j_0}$$
$$+ f(X)_{i_0,i_0} \cdot w(X)_{i_0,j_2} \cdot v_{j_1,j_0}$$
$$+ f(X)_{i_0,i_0} \cdot \langle W_{*,j_2}, X_{*,i_0} \rangle \cdot v_{j_1,j_0}$$

- **Part 2** *For $i_0 = i_1 \neq i_2 \in [n]$, $j_1, j_2 \in [d]$*

$$\frac{\mathrm{d}C_5(X)}{\mathrm{d}x_{i_2,j_2}} = - f(X)_{i_0,i_0} \cdot f(X)_{i_0,i_2} \cdot w(X)_{i_0,j_2} \cdot v_{j_1,j_0}$$

*Proof.* **Proof of Part 1**

$$\frac{\mathrm{d}C_5(X)}{\mathrm{d}x_{i_2,j_2}}$$
$$= \frac{\mathrm{d}f(X)_{i_0,i_0} \cdot v_{j_1,j_0}}{\mathrm{d}x_{i_2,j_2}}$$
$$= \frac{\mathrm{d}f(X)_{i_0,i_0}}{\mathrm{d}x_{i_2,j_2}} \cdot v_{j_1,j_0}$$
$$= (-f(X)_{i_0,i_0} \cdot (f(X)_{i_0,i_0} \cdot w(X)_{i_0,j_2} + \langle f(X)_{i_0}, X^\top W_{*,j_2} \rangle)$$
$$+ f(X)_{i_0,i_0} \cdot \langle W_{j_2,*} + W_{*,j_2}, X_{*,i_0} \rangle) \cdot v_{j_1,j_0}$$
$$= - f(X)_{i_0,i_0}^2 \cdot w(X)_{i_0,j_2} \cdot v_{j_1,j_0}$$
$$- f(X)_{i_0,i_0} \cdot \langle f(X)_{i_0}, X^\top W_{*,j_2} \rangle \cdot v_{j_1,j_0}$$
$$+ f(X)_{i_0,i_0} \cdot \langle W_{j_2,*} + W_{*,j_2}, X_{*,i_0} \rangle \cdot v_{j_1,j_0}$$

where the first step is by definition of $C_5(X)$ (see Lemma A.16), the 2nd step is by Fact A.2, the 3rd step is by Lemma B.4, the 4th step is a rearrangement.

**Proof of Part 2**

$$\frac{\mathrm{d}C_5(X)}{\mathrm{d}x_{i_2,j_2}}$$
$$= \frac{\mathrm{d}f(X)_{i_0,i_0} \cdot v_{j_1,j_0}}{\mathrm{d}x_{i_2,j_2}}$$
$$= \frac{\mathrm{d}f(X)_{i_0,i_0}}{\mathrm{d}x_{i_2,j_2}} \cdot v_{j_1,j_0}$$
$$= - f(X)_{i_0,i_0} \cdot f(X)_{i_0,i_2} \cdot w(X)_{i_0,j_2} \cdot v_{j_1,j_0}$$

where the first step is by definition of $C_5(X)$ (see Lemma A.16), the 2nd step is by Fact A.2, the 3rd step is by Lemma B.4. $\qquad\square$

### B.14 DERIVATIVE OF $\frac{c(X)_{i_0,j_0}}{\mathrm{d}x_{i_1,j_1}}$

**Lemma B.15.** *If the following holds:*

- *Let $c(X)_{i_0,j_0}$ be defined as in Definition A.8*

*We have*

- **Part 1** *For $i_0 = i_1 = i_2 \in [n]$, $j_1, j_2 \in [d]$*

$$\frac{\mathrm{d}c(X)_{i_0,j_0}}{\mathrm{d}x_{i_1,j_1} x_{i_2,j_2}} = \sum_{i=1}^{21} D_i(X)$$

*where we have following definitions*

$$D_1(X) := 2s(X)_{i_0,j_0} \cdot f(X)_{i_0,i_0}^2 \cdot w(X)_{i_0,j_2} \cdot w(X)_{i_0,j_1}$$
$$D_2(X) := 2f(X)_{i_0,i_0} \cdot s(X)_{i_0,j_0} \cdot z(X)_{i_0,j_2} \cdot w(X)_{i_0,j_1}$$

$$+ 2f(X)_{i_0,i_0} \cdot s(X)_{i_0,j_0} \cdot z(X)_{i_0,j_1} \cdot w(X)_{i_0,j_2}$$

$$D_3(X) := -f(X)_{i_0,i_0}^2 \cdot h(X)_{j_0,i_0} \cdot w(X)_{i_0,j_2} \cdot w(X)_{i_0,j_1}$$

$$D_4(X) := -f(X)_{i_0,i_0} \cdot \langle f(X)_{i_0} \circ (X^\top W_{*,j_2}), h(X)_{j_0} \rangle \cdot w(X)_{i_0,j_1}$$
$$- f(X)_{i_0,i_0} \cdot \langle f(X)_{i_0} \circ (X^\top W_{*,j_1}), h(X)_{j_0} \rangle \cdot w(X)_{i_0,j_2}$$

$$D_5(X) := -f(X)_{i_0,i_0}^2 \cdot v_{j_2,j_0} \cdot w(X)_{i_0,j_1} - f(X)_{i_0,i_0}^2 \cdot v_{j_1,j_0} \cdot w(X)_{i_0,j_2}$$

$$D_6(X) := -s(X)_{i_0,j_0} \cdot f(X)_{i_0,i_0} \cdot w(X)_{i_0,j_2} \cdot w(X)_{i_0,j_1}$$

$$D_7(X) := -s(X)_{i_0,j_0} \cdot f(X)_{i_0,i_0} \cdot \langle W_{*,j_2}, X_{*,i_0} \rangle \cdot w(X)_{i_0,j_1}$$
$$- s(X)_{i_0,j_0} \cdot f(X)_{i_0,i_0} \cdot \langle W_{*,j_1}, X_{*,i_0} \rangle \cdot w(X)_{i_0,j_2}$$

$$D_8(X) := -s(X)_{i_0,j_0} \cdot f(X)_{i_0,i_0} \cdot w_{j_1,j_2} - s(X)_{i_0,j_0} \cdot f(X)_{i_0,i_0} \cdot w_{j_2,j_1}$$

$$D_9(X) := s(X)_{i_0,j_0} \cdot z(X)_{i_0,j_2} \cdot z(X)_{i_0,j_1}$$

$$D_{10}(X) := -f(X)_{i_0,i_0} \cdot h(X)_{j_0,i_0} \cdot w(X)_{i_0,j_2} \cdot z(X)_{i_0,j_1}$$
$$- f(X)_{i_0,i_0} \cdot h(X)_{j_0,i_0} \cdot w(X)_{i_0,j_1} \cdot z(X)_{i_0,j_2}$$

$$D_{11}(X) := -\langle f(X)_{i_0} \circ (X^\top W_{*,j_2}), h(X)_{j_0} \rangle \cdot z(X)_{i_0,j_1}$$
$$- \langle f(X)_{i_0} \circ (X^\top W_{*,j_1}), h(X)_{j_0} \rangle \cdot z(X)_{i_0,j_2}$$

$$D_{12}(X) := -f(X)_{i_0,i_0} \cdot v_{j_2,j_0} \cdot z(X)_{i_0,j_1} - f(X)_{i_0,i_0} \cdot v_{j_1,j_0} \cdot z(X)_{i_0,j_2}$$

$$D_{13}(X) := s(X)_{i_0,j_0} \cdot z(X)_{i_0,j_1} \cdot f(X)_{i_0,i_0} \cdot z(X)_{i_0,j_2}$$

$$D_{14}(X) := -s(X)_{i_0,j_0} \cdot \langle f(X)_{i_0} \circ (X^\top W_{*,j_2}), X^\top W_{*,j_1} \rangle$$

$$D_{15}(X) := -f(X)_{i_0,i_0}^2 \cdot h(X)_{j_0,i_0} \cdot w(X)_{i_0,j_2} \cdot w(X)_{i_0,j_1}$$

$$D_{16}(X) := f(X)_{i_0,i_0} \cdot w(X)_{i_0,j_2} \cdot h(X)_{j_0,i_0} \cdot w(X)_{i_0,j_1}$$

$$D_{17}(X) := f(X)_{i_0,i_0} \cdot \langle W_{*,j_2}, X_{*,i_0} \rangle \cdot h(X)_{j_0,i_0} \cdot w(X)_{i_0,j_1}$$
$$+ f(X)_{i_0,i_0} \cdot \langle W_{*,j_1}, X_{*,i_0} \rangle \cdot h(X)_{j_0,i_0} \cdot w(X)_{i_0,j_2}$$

$$D_{18}(X) := f(X)_{i_0,i_0} \cdot v_{j_2,j_0} \cdot w(X)_{i_0,j_1} + f(X)_{i_0,i_0} \cdot v_{j_1,j_0} \cdot w(X)_{i_0,j_2}$$

$$D_{19}(X) := f(X)_{i_0,i_0} \cdot h(X)_{i_0,i_0} \cdot w_{j_1,j_2} + f(X)_{i_0,i_0} \cdot h(X)_{i_0,i_0} \cdot w_{j_2,j_1}$$

$$D_{20}(X) := \langle f(X)_{i_0} \circ (X^\top W_{*,j_2}) \circ (X^\top W_{*,j_1}), h(X)_{j_0} \rangle$$

$$D_{21}(X) := f(X)_{i_0,i_0} \cdot \langle W_{*,j_2}, X_{*,i_0} \rangle \cdot v_{j_1,j_0} + f(X)_{i_0,i_0} \cdot \langle W_{*,j_1}, X_{*,i_0} \rangle \cdot v_{j_2,j_0}$$

- **Part 2** *For $i_0 = i_1 \neq i_2 \in [n]$, $j_1, j_2 \in [d]$*

$$\frac{\mathrm{d}c(X)_{i_0,j_0}}{\mathrm{d}x_{i_1,j_1} x_{i_2,j_2}} = \sum_{i=1}^{15} E_i(X)$$

*where we have following definitions*

$$E_1(X) := 2s(X)_{i_0,j_0} \cdot f(X)_{i_0,i_2} \cdot w(X)_{i_0,j_2} \cdot f(X)_{i_0,i_0} \cdot w(X)_{i_0,j_1}$$

$$E_2(X) := -2f(X)_{i_0,i_2} \cdot h(X)_{j_0,i_2} \cdot w(X)_{i_0,j_2} \cdot f(X)_{i_0,i_0} \cdot w(X)_{i_0,j_1}$$

$$E_3(X) := -f(X)_{i_0,i_2} \cdot v_{j_2,j_0} \cdot f(X)_{i_0,i_0} \cdot w(X)_{i_0,j_1}$$

$$E_4(X) := s(X)_{i_0,j_0} \cdot f(X)_{i_0,i_2} \cdot w(X)_{i_0,j_2} \cdot z(X)_{i_0,j_1}$$

$$E_5(X) := -f(X)_{i_0,i_2} \cdot h(X)_{j_0,i_2} \cdot w(X)_{i_0,j_2} \cdot z(X)_{i_0,j_1}$$

$$E_6(X) := -f(X)_{i_0,i_2} \cdot v_{j_2,j_0} \cdot z(X)_{i_0,j_1}$$

$$E_7(X) := s(X)_{i_0,j_0} \cdot \langle f(X)_{i_0}, X^\top W_{*,j_1} \rangle \cdot f(X)_{i_0,i_0} \cdot w(X)_{i_0,j_2}$$

$$E_8(X) := -s(X)_{i_0,j_0} \cdot f(X)_{i_0,i_0} \cdot \langle W_{*,j_1}, X_{*,i_0} \rangle \cdot w(X)_{i_0,j_2}$$

$$E_9(X) := -s(X)_{i_0,j_0} \cdot f(X)_{i_0,i_0} \cdot w_{j_2,j_1}$$

$$E_{10}(X) := -f(X)_{i_0,i_0} \cdot f(X)_{i_0,i_2} \cdot w(X)_{i_0,j_2} \cdot h(X)_{j_0,i_0} \cdot w(X)_{i_0,j_1}$$

$$E_{11}(X) := -\langle f(X)_{i_0} \circ (X^\top W_{*,j_1}), h(X)_{j_0} \rangle \cdot f(X)_{i_0,i_2} \cdot w(X)_{i_0,j_2}$$

$$E_{12}(X) := f(X)_{i_0,i_2} \cdot h(X)_{j_0,i_2} \cdot \langle W_{*,j_1}, X_{*,i_2} \rangle \cdot w(X)_{i_0,j_2}$$

$$E_{13}(X) := f(X)_{i_0,i_2} \cdot h(X)_{j_0,i_2} \cdot w_{j_2,j_1}$$

$$E_{14}(X) := f(X)_{i_0,i_2} \cdot \langle W_{*,j_1}, X_{*,i_2} \rangle \cdot v_{j_2,j_0}$$

$$E_{15}(X) := -f(X)_{i_0,i_0} \cdot f(X)_{i_0,i_2} \cdot w(X)_{i_0,j_2} \cdot v_{j_1,j_0}$$

*Proof.* The proof is a combination of derivatives of $C_i(X)$ in this section.

Notice that the symmetricity for **Part 1** is verified by tables in this section. $\qquad\square$

## C  HESSIAN CASE 2: $i_0 \neq i_1$

In this section, we focus on the second case of Hessian. In Sections C.1, C.2, C.3, C.4 and C.5, we calculated derivative of some important terms. In Sections C.6, C.7 and C.8 we calculate derivative of $C_6$, $C_7$ and $C_8$ respectively. And in Section C.9 we calculate the derivative of $\frac{\mathrm{d}c(X)_{i_0,j_1}}{\mathrm{d}x_{i_1,j_1}}$.

### C.1  DERIVATIVE OF SCALAR FUNCTION $f(X)_{i_0,i_1}$

**Lemma C.1.** *If the following holds:*

- *Let $f(X)_{i_0}$ be defined as Definition A.6*

- *For $i_0 \neq i_2 \in [n]$, $j_1, j_2 \in [d]$*

*We have*

- **Part 1.** *For $i_0 \neq i_2, i_1 = i_2 \in [n]$, $j_1, j_2 \in [d]$*

$$\frac{\mathrm{d}f(X)_{i_0,i_1}}{\mathrm{d}x_{i_2,j_2}} = -f(X)_{i_0,i_1} \cdot f(X)_{i_0,i_2} \cdot w(X)_{i_0,j_2}$$
$$+ f(X)_{i_0,i_1} \cdot w(X)_{i_0,j_2}$$

- **Part 2.** *For $i_0 \neq i_2, i_1 \neq i_2 \in [n]$, $j_1, j_2 \in [d]$*

$$\frac{\mathrm{d}f(X)_{i_0,i_1}}{\mathrm{d}x_{i_2,j_2}} = -f(X)_{i_0,i_1} \cdot f(X)_{i_0,i_2} \cdot w(X)_{i_0,j_2}$$

*Proof.* **Proof of Part 1**

$$\frac{\mathrm{d}f(X)_{i_0,i_1}}{\mathrm{d}x_{i_2,j_2}} = (-(\alpha(X)_{i_0})^{-1} \cdot f(X)_{i_0} \cdot u(X)_{i_0,i_2} \cdot \langle W_{j_2,*}, X_{*,i_0} \rangle$$
$$+ f(X)_{i_0} \circ (e_{i_1} \cdot \langle W_{j_2,*}, X_{*,i_0} \rangle))_{i_1}$$
$$= -(\alpha(X)_{i_0})^{-1} \cdot f(X)_{i_0,i_1} \cdot u(X)_{i_0,i_2} \cdot \langle W_{j_2,*}, X_{*,i_0} \rangle$$
$$+ f(X)_{i_0,i_1} \cdot \langle W_{j_2,*}, X_{*,i_0} \rangle$$
$$= -f(X)_{i_0,i_1} \cdot f(X)_{i_0,i_2} \cdot w(X)_{i_0,j_2}$$
$$+ f(X)_{i_0,i_1} \cdot w(X)_{i_0,j_2}$$

where the first step follows from Part 1 of Lemma A.14, the second step follows from simple algebra, the first step follows from Definition A.6.

**Proof of Part 2**

$$\frac{\mathrm{d}f(X)_{i_0,i_1}}{\mathrm{d}x_{i_2,j_2}} = (-(\alpha(X)_{i_0})^{-1} \cdot f(X)_{i_0} \cdot u(X)_{i_0,i_2} \cdot \langle W_{j_2,*}, X_{*,i_0} \rangle$$
$$+ f(X)_{i_0} \circ (e_{i_2} \cdot \langle W_{j_2,*}, X_{*,i_0} \rangle))_{i_1}$$
$$= -(\alpha(X)_{i_0})^{-1} \cdot f(X)_{i_0,i_1} \cdot u(X)_{i_0,i_2} \cdot \langle W_{j_2,*}, X_{*,i_0} \rangle$$
$$= -f(X)_{i_0,i_1} \cdot f(X)_{i_0,i_2} \cdot w(X)_{i_0,j_2}$$

where the first step follows from Part 1 of Lemma A.14, the second step follows from simple algebra, the first step follows from Definition A.6. $\qquad\square$

## C.2 Derivative of scalar function $h(X)_{j_0,i_1}$

**Lemma C.2.** *If the following holds:*

- *Let $h(X)_{j_0}$ be defined as Definition A.7*

- *For $i_0 \neq i_2 \in [n]$, $j_1, j_2 \in [d]$*

*We have*

- **Part 1.** *For $i_0 \neq i_2, i_1 = i_2 \in [n]$, $j_1, j_2 \in [d]$*

$$\frac{\mathrm{d}h(X)_{j_0,i_1}}{\mathrm{d}x_{i_2,j_2}} = v_{j_2,j_0}$$

- **Part 2.** *For $i_0 \neq i_2, i_1 \neq i_2 \in [n]$, $j_1, j_2 \in [d]$*

$$\frac{\mathrm{d}h(X)_{j_0,i_1}}{\mathrm{d}x_{i_2,j_2}} = 0$$

*Proof.* **Proof of Part 1.**

$$\frac{\mathrm{d}h(X)_{j_0,i_1}}{\mathrm{d}x_{i_2,j_2}} = (e_{i_2} \cdot v_{j_2,j_0})_{i_1}$$
$$= v_{j_2,j_0}$$

where the first step follows from Lemma A.7, the second step follows from $i_1 = i_2$.

**Proof of Part 1.**

$$\frac{\mathrm{d}h(X)_{j_0,i_1}}{\mathrm{d}x_{i_2,j_2}} = (e_{i_2} \cdot v_{j_2,j_0})_{i_1}$$
$$= 0$$

where the first step follows from Lemma A.7, the second step follows from $i_1 \neq i_2$. $\qquad \square$

## C.3 Derivative of scalar function $\langle f(X)_{i_0}, h(X)_{j_0} \rangle$

**Lemma C.3.** *If the following holds:*

- *Let $f(X)_{i_0}$ be defined as Definition A.6*

- *Let $h(X)_{j_0}$ be defined as Definition A.7*

- *For $i_0 \neq i_2 \in [n]$, $j_1, j_2 \in [d]$*

*We have*

$$\frac{\mathrm{d}\langle f(X)_{i_0}, h(X)_{j_0} \rangle}{\mathrm{d}x_{i_2,j_2}} = \langle -f(X)_{i_0} \cdot f(X)_{i_0,i_2} \cdot \langle W_{j_2,*}, X_{*,i_0} \rangle$$
$$+ f(X)_{i_0} \circ (e_{i_2} \cdot \langle W_{j_2,*}, X_{*,i_0} \rangle), h(X)_{j_0} \rangle + f(X)_{i_0,i_2} \cdot v_{j_2,j_0}$$

*Proof.*

$$\frac{\mathrm{d}\langle f(X)_{i_0}, h(X)_{j_0} \rangle}{\mathrm{d}x_{i_2,j_2}} = \langle \frac{\mathrm{d}f(X)_{i_0}}{\mathrm{d}x_{i_2,j_2}}, h(X)_{j_0} \rangle + \langle f(X)_{i_0}, \frac{\mathrm{d}h(X)_{j_0}}{\mathrm{d}x_{i_2,j_2}} \rangle$$
$$= \langle -(\alpha(X)_{i_0})^{-1} \cdot f(X)_{i_0} \cdot u(X)_{i_0,i_2} \cdot \langle W_{j_2,*}, X_{*,i_0} \rangle$$
$$+ f(X)_{i_0} \circ (e_{i_2} \cdot \langle W_{j_2,*}, X_{*,i_0} \rangle), h(X)_{j_0} \rangle + \langle f(X)_{i_0}, \frac{\mathrm{d}h(X)_{j_0}}{\mathrm{d}x_{i_2,j_2}} \rangle$$
$$= \langle -f(X)_{i_0} \cdot f(X)_{i_0,i_2} \cdot \langle W_{j_2,*}, X_{*,i_0} \rangle$$

$$+ f(X)_{i_0} \circ (e_{i_2} \cdot \langle W_{j_2,*}, X_{*,i_0} \rangle)), h(X)_{j_0} \rangle + \langle f(X)_{i_0}, \frac{\mathrm{d}h(X)_{j_0}}{\mathrm{d}x_{i_2,j_2}} \rangle$$

$$= \langle -f(X)_{i_0} \cdot f(X)_{i_0,i_2} \cdot \langle W_{j_2,*}, X_{*,i_0} \rangle$$
$$+ f(X)_{i_0} \circ (e_{i_2} \cdot \langle W_{j_2,*}, X_{*,i_0} \rangle)), h(X)_{j_0} \rangle + \langle f(X)_{i_0}, e_{i_2} \cdot v_{j_2,j_0} \rangle$$
$$= \langle -f(X)_{i_0} \cdot f(X)_{i_0,i_2} \cdot \langle W_{j_2,*}, X_{*,i_0} \rangle$$
$$+ f(X)_{i_0} \circ (e_{i_2} \cdot \langle W_{j_2,*}, X_{*,i_0} \rangle)), h(X)_{j_0} \rangle + f(X)_{i_0,i_2} \cdot v_{j_2,j_0}$$

where the first step follows from simple differential rule, the second step follows from Lemma A.14, the third step follows from simple algebra and Definition A.6, the fourth step follows from Lemma A.15, the last step follows from simple algebra. $\square$

### C.4 Derivative of scalar function $f(X)_{i_0,i_1} \cdot \langle W_{j_1,*}, X_{*,i_0} \rangle$

**Lemma C.4.** *If the following holds:*

- *Let $f(X)_{i_0}$ be defined as Definition A.6*

- *For $i_0 \neq i_2 \in [n]$, $j_1, j_2 \in [d]$*

*We have*

- **Part 1.** *For $i_0 \neq i_2, i_1 = i_2 \in [n]$, $j_1, j_2 \in [d]$*

$$\frac{\mathrm{d}f(X)_{i_0,i_1} \cdot \langle W_{j_1,*}, X_{*,i_0} \rangle}{\mathrm{d}x_{i_2,j_2}}$$
$$= (-f(X)_{i_0,i_2} \cdot f(X)_{i_0,i_1} + f(X)_{i_0,i_1}) \cdot \langle W_{j_2,*}, X_{*,i_0} \rangle \cdot \langle W_{j_1,*}, X_{*,i_0} \rangle$$

- **Part 2.** *For $i_0 \neq i_2, i_1 \neq i_2 \in [n]$, $j_1, j_2 \in [d]$*

$$\frac{\mathrm{d}f(X)_{i_0,i_1} \cdot \langle W_{j_1,*}, X_{*,i_0} \rangle}{\mathrm{d}x_{i_2,j_2}}$$
$$= - f(X)_{i_0,i_2} \cdot f(X)_{i_0,i_1} \cdot \langle W_{j_2,*}, X_{*,i_0} \rangle \cdot \langle W_{j_1,*}, X_{*,i_0} \rangle$$

*Proof.* **Proof of Part 1**

$$\frac{\mathrm{d}f(X)_{i_0,i_1} \cdot \langle W_{j_1,*}, X_{*,i_0} \rangle}{\mathrm{d}x_{i_2,j_2}}$$
$$= \frac{\mathrm{d}f(X)_{i_0,i_1}}{\mathrm{d}x_{i_2,j_2}} \cdot \langle W_{j_1,*}, X_{*,i_0} \rangle + \frac{\mathrm{d}\langle W_{j_1,*}, X_{*,i_0} \rangle}{\mathrm{d}x_{i_2,j_2}} \cdot f(X)_{i_0,i_1}$$
$$= (-f(X)_{i_0,i_2} f(X)_{i_0,i_1} + f(X)_{i_0,i_1}) \cdot \langle W_{j_2,*}, X_{*,i_0} \rangle \cdot \langle W_{j_1,*}, X_{*,i_0} \rangle$$
$$+ \frac{\mathrm{d}\langle W_{j_1,*}, X_{*,i_0} \rangle}{\mathrm{d}x_{i_2,j_2}} \cdot f(X)_{i_0,i_1}$$
$$= (-f(X)_{i_0,i_2} f(X)_{i_0,i_1} + f(X)_{i_0,i_1}) \cdot \langle W_{j_2,*}, X_{*,i_0} \rangle \cdot \langle W_{j_1,*}, X_{*,i_0} \rangle + \mathbf{0}_d \cdot f(X)_{i_0,i_1}$$
$$= (-f(X)_{i_0,i_2} f(X)_{i_0,i_1} + f(X)_{i_0,i_1}) \cdot \langle W_{j_2,*}, X_{*,i_0} \rangle \cdot \langle W_{j_1,*}, X_{*,i_0} \rangle$$

where the first step follows from simple differential rule, the second step follows from Lemma C.1, the third step follows from $i_0 \neq i_2$, the last step follows from simple algebra.

**Proof of Part 2**

$$\frac{\mathrm{d}f(X)_{i_0,i_1} \cdot \langle W_{j_1,*}, X_{*,i_0} \rangle}{\mathrm{d}x_{i_2,j_2}}$$
$$= \frac{\mathrm{d}f(X)_{i_0,i_1}}{\mathrm{d}x_{i_2,j_2}} \cdot \langle W_{j_1,*}, X_{*,i_0} \rangle + \frac{\mathrm{d}\langle W_{j_1,*}, X_{*,i_0} \rangle}{\mathrm{d}x_{i_2,j_2}} \cdot f(X)_{i_0,i_1}$$
$$= (-f(X)_{i_0,i_2} f(X)_{i_0,i_1} + f(X)_{i_0,i_1}) \cdot \langle W_{j_2,*}, X_{*,i_0} \rangle \cdot \langle W_{j_1,*}, X_{*,i_0} \rangle$$
$$+ \frac{\mathrm{d}\langle W_{j_1,*}, X_{*,i_0} \rangle}{\mathrm{d}x_{i_2,j_2}} \cdot f(X)_{i_0,i_1}$$

$$= (-f(X)_{i_0,i_2} f(X)_{i_0,i_1} + f(X)_{i_0,i_1}) \cdot \langle W_{j_2,*}, X_{*,i_0} \rangle \cdot \langle W_{j_1,*}, X_{*,i_0} \rangle + \mathbf{0}_d \cdot f(X)_{i_0,i_1}$$
$$= -f(X)_{i_0,i_2} \cdot f(X)_{i_0,i_1} \cdot \langle W_{j_2,*}, X_{*,i_0} \rangle \cdot \langle W_{j_1,*}, X_{*,i_0} \rangle$$

where the first step follows from simple differential rule, the second step follows from Lemma C.1, the third step follows from $i_0 \neq i_2$, the last step follows from simple algebra. $\square$

## C.5 DERIVATIVE OF SCALAR FUNCTION $f(X)_{i_0,i_1} \cdot h(X)_{j_0,i_1}$

**Lemma C.5.** *If the following holds:*

- *Let $f(X)_{i_0}$ be defined as Definition A.6*

- *Let $h(X)_{j_0}$ be defined as Definition A.7*

*We have*

- **Part 1** *For $i_0 \neq i_2, i_1 = i_2 \in [n]$, $j_1, j_2 \in [d]$*

$$\frac{\mathrm{d} f(X)_{i_0,i_1} \cdot h(X)_{j_0,i_1}}{\mathrm{d} x_{i_2,j_2}}$$
$$= (-f(X)_{i_0,i_2} \cdot f(X)_{i_0,i_1} + f(X)_{i_0,i_1}) \cdot \langle W_{j_2,*}, X_{*,i_0} \rangle \cdot h(X)_{j_0,i_1}$$
$$+ v_{j_2,j_0} \cdot f(X)_{i_0,i_1}$$

- **Part 2** *For $i_0 \neq i_2, i_1 \neq i_2 \in [n]$, $j_1, j_2 \in [d]$*

$$\frac{\mathrm{d} f(X)_{i_0,i_0} \cdot h(X)_{j_0,i_0}}{\mathrm{d} x_{i_2,j_2}}$$
$$= -f(X)_{i_0,i_2} \cdot f(X)_{i_0,i_1} \cdot \langle W_{j_2,*}, X_{*,i_0} \rangle \cdot h(X)_{j_0,i_1}$$

*Proof.* **Proof of Part 1.**

$$\frac{\mathrm{d} f(X)_{i_0,i_1} \cdot h(X)_{j_0,i_1}}{\mathrm{d} x_{i_2,j_2}} = \frac{\mathrm{d} f(X)_{i_0,i_1}}{\mathrm{d} x_{i_2,j_2}} \cdot h(X)_{j_0,i_1} + \frac{\mathrm{d} h(X)_{j_0,i_1}}{\mathrm{d} x_{i_2,j_2}} \cdot f(X)_{i_0,i_1}$$
$$= (-f(X)_{i_0,i_2} f(X)_{i_0,i_1} + f(X)_{i_0,i_1}) \cdot \langle W_{j_2,*}, X_{*,i_0} \rangle \cdot h(X)_{j_0,i_1}$$
$$+ \frac{\mathrm{d} h(X)_{j_0,i_1}}{\mathrm{d} x_{i_2,j_2}} \cdot f(X)_{i_0,i_1}$$
$$= (-f(X)_{i_0,i_2} \cdot f(X)_{i_0,i_1} + f(X)_{i_0,i_1}) \cdot \langle W_{j_2,*}, X_{*,i_0} \rangle \cdot h(X)_{j_0,i_1}$$
$$+ v_{j_2,j_0} \cdot f(X)_{i_0,i_1}$$

where the first step follows from simple differential rule, the second step follows from Lemma C.1, the third step follows from Part 1 of Lemma C.2.

**Proof of Part 2.**

$$\frac{\mathrm{d} f(X)_{i_0,i_1} \cdot h(X)_{j_0,i_1}}{\mathrm{d} x_{i_2,j_2}} = \frac{\mathrm{d} f(X)_{i_0,i_1}}{\mathrm{d} x_{i_2,j_2}} \cdot h(X)_{j_0,i_1} + \frac{\mathrm{d} h(X)_{j_0,i_1}}{\mathrm{d} x_{i_2,j_2}} \cdot f(X)_{i_0,i_1}$$
$$= -f(X)_{i_0,i_2} \cdot f(X)_{i_0,i_1} \cdot \langle W_{j_2,*}, X_{*,i_0} \rangle \cdot h(X)_{j_0,i_1}$$
$$+ \frac{\mathrm{d} h(X)_{j_0,i_1}}{\mathrm{d} x_{i_2,j_2}} \cdot f(X)_{i_0,i_1}$$
$$= -f(X)_{i_0,i_2} \cdot f(X)_{i_0,i_1} \cdot \langle W_{j_2,*}, X_{*,i_0} \rangle \cdot h(X)_{j_0,i_1}$$

where the first step follows from simple differential rule, the second step follows from Lemma C.1, the third step follows from Part 2 of Lemma C.2. $\square$

## C.6 DERIVATIVE OF $C_6(X)$

**Lemma C.6.** *If the following holds:*

- *Let $C_6(X) \in \mathbb{R}$ be defined as in Lemma A.16*

- *For $i_0 \neq i_2 \in [n]$, $j_1, j_2 \in [d]$*

*We have*

- **Part 1** *For $i_0 \neq i_2, i_1 = i_2 \in [n]$, $j_1, j_2 \in [d]$*

$$\frac{\mathrm{d}C_6(X)}{\mathrm{d}x_{i_2,j_2}}$$
$$= -(\langle -f(X)_{i_0} \cdot f(X)_{i_0,i_2} \cdot \langle W_{j_2,*}, X_{*,i_0} \rangle$$
$$+ f(X)_{i_0} \circ (e_{i_1} \cdot \langle W_{j_2,*}, X_{*,i_0} \rangle), h(X)_{j_0} \rangle + f(X)_{i_0,i_2} \cdot v_{j_2,j_0}) \cdot f(X)_{i_0,i_1} \cdot \langle W_{j_1,*}, X_{*,i_0} \rangle$$
$$+ (-\langle f(X)_{i_0}, h(X)_{j_0} \rangle) \cdot (-f(X)_{i_0,i_2} f(X)_{i_0,i_1} + f(X)_{i_0,i_1}) \cdot \langle W_{j_2,*}, X_{*,i_0} \rangle \cdot \langle W_{j_1,*}, X_{*,i_0} \rangle$$

- **Part 2** *For $i_0 \neq i_2, i_1 \neq i_2 \in [n]$, $j_1, j_2 \in [d]$*

$$\frac{\mathrm{d}C_6(X)}{\mathrm{d}x_{i_2,j_2}}$$
$$= -(\langle -f(X)_{i_0} \cdot f(X)_{i_0,i_2} \cdot \langle W_{j_2,*}, X_{*,i_0} \rangle$$
$$+ f(X)_{i_0} \circ (e_{i_2} \cdot \langle W_{j_2,*}, X_{*,i_0} \rangle), h(X)_{j_0} \rangle + f(X)_{i_0,i_2} \cdot v_{j_2,j_0}) \cdot f(X)_{i_0,i_1} \cdot \langle W_{j_1,*}, X_{*,i_0} \rangle$$
$$+ \langle f(X)_{i_0}, h(X)_{j_0} \rangle \cdot f(X)_{i_0,i_2} \cdot f(X)_{i_0,i_1} \cdot \langle W_{j_2,*}, X_{*,i_0} \rangle \cdot \langle W_{j_1,*}, X_{*,i_0} \rangle$$

*Proof.* **Proof of Part 1**

$$\frac{\mathrm{d}C_6(X)}{\mathrm{d}x_{i_2,j_2}}$$
$$= \frac{\mathrm{d}}{\mathrm{d}x_{i_2,j_2}}(-\langle f(X)_{i_0}, h(X)_{j_0} \rangle \cdot f(X)_{i_0,i_1} \cdot \langle W_{j_1,*}, X_{*,i_0} \rangle)$$
$$= \frac{\mathrm{d}}{\mathrm{d}x_{i_2,j_2}}(-\langle f(X)_{i_0}, h(X)_{j_0} \rangle) \cdot f(X)_{i_0,i_1} \cdot \langle W_{j_1,*}, X_{*,i_0} \rangle$$

$$+ (-\langle f(X)_{i_0}, h(X)_{j_0} \rangle) \cdot \frac{\mathrm{d}}{\mathrm{d}x_{i_2,j_2}}(f(X)_{i_0,i_1} \cdot \langle W_{j_1,*}, X_{*,i_0} \rangle)$$
$$= \frac{\mathrm{d}}{\mathrm{d}x_{i_2,j_2}}(-\langle f(X)_{i_0}, h(X)_{j_0} \rangle) \cdot f(X)_{i_0,i_1} \cdot \langle W_{j_1,*}, X_{*,i_0} \rangle$$
$$+ (-\langle f(X)_{i_0}, h(X)_{j_0} \rangle) \cdot (-f(X)_{i_0,i_2} f(X)_{i_0,i_1} + f(X)_{i_0,i_1}) \cdot \langle W_{j_2,*}, X_{*,i_0} \rangle \cdot \langle W_{j_1,*}, X_{*,i_0} \rangle$$
$$= -(\langle -f(X)_{i_0} \cdot f(X)_{i_0,i_2} \cdot \langle W_{j_2,*}, X_{*,i_0} \rangle$$
$$+ f(X)_{i_0} \circ (e_{i_1} \cdot \langle W_{j_2,*}, X_{*,i_0} \rangle), h(X)_{j_0} \rangle + f(X)_{i_0,i_2} \cdot v_{j_2,j_0}) \cdot f(X)_{i_0,i_1} \cdot \langle W_{j_1,*}, X_{*,i_0} \rangle$$
$$+ (-\langle f(X)_{i_0}, h(X)_{j_0} \rangle) \cdot (-f(X)_{i_0,i_2} f(X)_{i_0,i_1} + f(X)_{i_0,i_1}) \cdot \langle W_{j_2,*}, X_{*,i_0} \rangle \cdot \langle W_{j_1,*}, X_{*,i_0} \rangle$$

where the first step follows from Lemma A.16, the second step follows from simple differential rule, the third step follows from Lemma C.4, last step follows from Lemma C.3.

**Proof of Part 2**

$$\frac{\mathrm{d}C_6(X)}{\mathrm{d}x_{i_2,j_2}}$$
$$= \frac{\mathrm{d}}{\mathrm{d}x_{i_2,j_2}}(-\langle f(X)_{i_0}, h(X)_{j_0} \rangle \cdot f(X)_{i_0,i_1} \cdot \langle W_{j_1,*}, X_{*,i_0} \rangle)$$
$$= \frac{\mathrm{d}}{\mathrm{d}x_{i_2,j_2}}(-\langle f(X)_{i_0}, h(X)_{j_0} \rangle) \cdot f(X)_{i_0,i_1} \cdot \langle W_{j_1,*}, X_{*,i_0} \rangle$$

$$+ (-\langle f(X)_{i_0}, h(X)_{j_0} \rangle) \cdot \frac{\mathrm{d}}{\mathrm{d}x_{i_2,j_2}}(f(X)_{i_0,i_1} \cdot \langle W_{j_1,*}, X_{*,i_0} \rangle)$$
$$= \frac{\mathrm{d}}{\mathrm{d}x_{i_2,j_2}}(-\langle f(X)_{i_0}, h(X)_{j_0} \rangle) \cdot f(X)_{i_0,i_1} \cdot \langle W_{j_1,*}, X_{*,i_0} \rangle$$

$$+ \langle f(X)_{i_0}, h(X)_{j_0} \rangle) \cdot f(X)_{i_0,i_2} \cdot f(X)_{i_0,i_1} \cdot \langle W_{j_2,*}, X_{*,i_0} \rangle \cdot \langle W_{j_1,*}, X_{*,i_0} \rangle$$

$$= - ((\langle -f(X)_{i_0} \cdot f(X)_{i_0,i_2} \cdot \langle W_{j_2,*}, X_{*,i_0} \rangle$$
$$+ f(X)_{i_0} \circ (e_{i_2} \cdot \langle W_{j_2,*}, X_{*,i_0} \rangle), h(X)_{j_0} \rangle + f(X)_{i_0,i_2} \cdot v_{j_2,j_0}) \cdot f(X)_{i_0,i_1} \cdot \langle W_{j_1,*}, X_{*,i_0} \rangle$$
$$+ \langle f(X)_{i_0}, h(X)_{j_0} \rangle \cdot f(X)_{i_0,i_2} \cdot f(X)_{i_0,i_1} \cdot \langle W_{j_2,*}, X_{*,i_0} \rangle \cdot \langle W_{j_1,*}, X_{*,i_0} \rangle$$

where the first step follows from Lemma A.16, the second step follows from simple differential rule, the third step follows from Lemma C.4, last step follows from Lemma C.3. $\qquad\square$

## C.7    DERIVATIVE OF $C_7(X)$

**Lemma C.7.** *If the following holds:*

- *Let $C_7(X) \in \mathbb{R}$ be defined as in Lemma A.16*

*We have*

- **Part 1.** *For $i_0 \neq i_2, i_1 = i_2 \in [n], j_1, j_2 \in [d]$*

$$\frac{\mathrm{d}C_7(X)}{\mathrm{d}x_{i_2,j_2}}$$
$$= (-f(X)_{i_0,i_2} + 1) \cdot f(X)_{i_0,i_1} \cdot \langle W_{j_2,*}, X_{*,i_0} \rangle \cdot h(X)_{j_0,i_1} \cdot \langle W_{j_1,*}, X_{*,i_0} \rangle$$
$$+ v_{j_2,j_0} \cdot f(X)_{i_0,i_1} \cdot \langle W_{j_1,*}, X_{*,i_0} \rangle$$

- **Part 2.** *For $i_0 \neq i_2, i_1 \neq i_2 \in [n], j_1, j_2 \in [d]$*

$$\frac{\mathrm{d}C_7(X)}{\mathrm{d}x_{i_2,j_2}}$$
$$= - f(X)_{i_0,i_2} \cdot f(X)_{i_0,i_1} \cdot \langle W_{j_2,*}, X_{*,i_0} \rangle \cdot h(X)_{j_0,i_1} \cdot \langle W_{j_1,*}, X_{*,i_0} \rangle$$

*Proof.* **Proof of Part 1.**

$$\frac{\mathrm{d}C_7(X)}{\mathrm{d}x_{i_2,j_2}}$$
$$= \frac{\mathrm{d}}{\mathrm{d}x_{i_2,j_2}} (f(X)_{i_0,i_1} \cdot h(X)_{j_0,i_1} \cdot \langle W_{j_1,*}, X_{*,i_0} \rangle)$$
$$= \frac{\mathrm{d}}{\mathrm{d}x_{i_2,j_2}} (f(X)_{i_0,i_1} \cdot h(X)_{j_0,i_1}) \cdot \langle W_{j_1,*}, X_{*,i_0} \rangle + f(X)_{i_0,i_1} \cdot h(X)_{j_0,i_1} \cdot \frac{\mathrm{d}}{\mathrm{d}x_{i_2,j_2}} (\langle W_{j_1,*}, X_{*,i_0} \rangle)$$
$$= (-f(X)_{i_0,i_2} + 1) \cdot f(X)_{i_0,i_1} \cdot \langle W_{j_2,*}, X_{*,i_0} \rangle \cdot h(X)_{j_0,i_1} \cdot \langle W_{j_1,*}, X_{*,i_0} \rangle$$
$$+ v_{j_2,j_0} \cdot f(X)_{i_0,i_1} \cdot \langle W_{j_1,*}, X_{*,i_0} \rangle$$
$$+ f(X)_{i_0,i_1} \cdot h(X)_{j_0,i_1} \cdot \frac{\mathrm{d}}{\mathrm{d}x_{i_2,j_2}} (\langle W_{j_1,*}, X_{*,i_0} \rangle)$$
$$= (-f(X)_{i_0,i_2} + 1) \cdot f(X)_{i_0,i_1} \cdot \langle W_{j_2,*}, X_{*,i_0} \rangle \cdot h(X)_{j_0,i_1} \cdot \langle W_{j_1,*}, X_{*,i_0} \rangle$$
$$+ v_{j_2,j_0} \cdot f(X)_{i_0,i_1} \cdot \langle W_{j_1,*}, X_{*,i_0} \rangle$$

where the first step follows from Lemma A.16, the second step follows from differential rule, the third step follows from Part 1 of Lemma C.3, the fourth step follows from $i_0 \neq i_2$.

**Proof of Part 2.**

$$\frac{\mathrm{d}C_7(X)}{\mathrm{d}x_{i_2,j_2}}$$
$$= \frac{\mathrm{d}}{\mathrm{d}x_{i_2,j_2}} (f(X)_{i_0,i_1} \cdot h(X)_{j_0,i_1} \cdot \langle W_{j_1,*}, X_{*,i_0} \rangle)$$
$$= \frac{\mathrm{d}}{\mathrm{d}x_{i_2,j_2}} (f(X)_{i_0,i_1} \cdot h(X)_{j_0,i_1}) \cdot \langle W_{j_1,*}, X_{*,i_0} \rangle + f(X)_{i_0,i_1} \cdot h(X)_{j_0,i_1} \cdot \frac{\mathrm{d}}{\mathrm{d}x_{i_2,j_2}} (\langle W_{j_1,*}, X_{*,i_0} \rangle)$$

$$= - f(X)_{i_0,i_2} f(X)_{i_0,i_1} \cdot \langle W_{j_2,*}, X_{*,i_0} \rangle \cdot h(X)_{j_0,i_1} \cdot \langle W_{j_1,*}, X_{*,i_0} \rangle$$

$$+ f(X)_{i_0,i_1} \cdot h(X)_{j_0,i_1} \cdot \frac{\mathrm{d}}{\mathrm{d}x_{i_2,j_2}} (\langle W_{j_1,*}, X_{*,i_0} \rangle)$$

$$= - f(X)_{i_0,i_2} f(X)_{i_0,i_1} \cdot \langle W_{j_2,*}, X_{*,i_0} \rangle \cdot h(X)_{j_0,i_1} \cdot \langle W_{j_1,*}, X_{*,i_0} \rangle$$

$$+ f(X)_{i_0,i_1} \cdot h(X)_{j_0,i_1} \cdot \mathbf{0}_d$$

$$= - f(X)_{i_0,i_2} \cdot f(X)_{i_0,i_1} \cdot \langle W_{j_2,*}, X_{*,i_0} \rangle \cdot h(X)_{j_0,i_1} \cdot \langle W_{j_1,*}, X_{*,i_0} \rangle$$

where the first step follows from Lemma A.16, the second step follows from differential rule, the third step follows from Part 2 of Lemma C.3, the fourth step follows from $i_0 \neq i_2$, the last step follows from simple algebra. $\qquad\square$

## C.8 DERIVATIVE OF $C_8(X)$

**Lemma C.8.** *If the following holds:*

- *Let $C_8(X) \in \mathbb{R}$ be defined as in Lemma A.16*

- *For $i_0 \neq i_2 \in [n]$, $j_1, j_2 \in [d]$*

*We have*

- **Part 1.** *For $i_0 \neq i_2, i_1 = i_2 \in [n]$, $j_1, j_2 \in [d]$*

$$\frac{\mathrm{d}C_8(X)}{\mathrm{d}x_{i_2,j_2}}$$
$$= (-f(X)_{i_0,i_2} f(X)_{i_0,i_1} + f(X)_{i_0,i_1}) \cdot \langle W_{j_2,*}, X_{*,i_0} \rangle \cdot v_{j_1,j_0}$$

- **Part 2.** *For $i_0 \neq i_2, i_1 \neq i_2 \in [n]$, $j_1, j_2 \in [d]$*

$$\frac{\mathrm{d}C_8(X)}{\mathrm{d}x_{i_2,j_2}}$$
$$= - f(X)_{i_0,i_2} \cdot f(X)_{i_0,i_1} \cdot \langle W_{j_2,*}, X_{*,i_0} \rangle \cdot v_{j_1,j_0}$$

*Proof.* **Proof of Part 1**

$$\frac{\mathrm{d}C_8(X)}{\mathrm{d}x_{i_2,j_2}} = \frac{\mathrm{d}}{\mathrm{d}x_{i_2,j_2}} f(X)_{i_0,i_1} \cdot v_{j_1,j_0}$$
$$= (-f(X)_{i_0,i_2} f(X)_{i_0,i_1} + f(X)_{i_0,i_1}) \cdot \langle W_{j_2,*}, X_{*,i_0} \rangle \cdot v_{j_1,j_0}$$

where the first step follows from Lemma A.16, the second step follows from differential rule and Lemma C.1.

**Proof of Part 2**

$$\frac{\mathrm{d}C_8(X)}{\mathrm{d}x_{i_2,j_2}} = \frac{\mathrm{d}}{\mathrm{d}x_{i_2,j_2}} f(X)_{i_0,i_1} \cdot v_{j_1,j_0}$$
$$= - f(X)_{i_0,i_2} \cdot f(X)_{i_0,i_1} \cdot \langle W_{j_2,*}, X_{*,i_0} \rangle \cdot v_{j_1,j_0}$$

where the first step follows from Lemma A.16, the second step follows from differential rule and Lemma C.1. $\qquad\square$

## C.9 DERIVATIVE OF $\frac{\mathrm{d}c(X)_{i_0,j_1}}{\mathrm{d}x_{i_1,j_1}}$

**Lemma C.9.** *If the following holds:*

- *Let $c(X)_{i_0,j_1} \in \mathbb{R}$ be defined as in Lemma A.16 and Definition A.8*

*We have*

- **Part 1** *For $i_0 \neq i_2, i_1 = i_2 \in [n]$, $j_1, j_2 \in [d]$*

$$\frac{\mathrm{d}c(X)}{\mathrm{d}x_{i_1,j_1}, \mathrm{d}x_{i_2,j_2}} = \sum_{i=1}^{6} F_i(X)$$

*where we have following definitions*

$$F_1(X) = 2s(X)_{i_0,j_0} \cdot f(X)_{i_0,i_1}^2 \cdot w(X)_{i_0,j_2} \cdot w(X)_{i_0,j_1}$$
$$F_2(X) = -f(X)_{i_0,i_1}^2 \cdot h(X)_{j_0,i_1} \cdot w(X)_{i_0,j_2} \cdot w(X)_{i_0,j_1}$$
$$F_3(X) = -f(X)_{i_0,i_1}^2 \cdot v_{j_2,j_0} \cdot w(X)_{i_0,j_1} - f(X)_{i_0,i_1}^2 \cdot v_{j_1,j_0} \cdot w(X)_{i_0,j_2}$$
$$F_4(X) = -s(X)_{i_0,j_0} \cdot f(X)_{i_0,i_1} \cdot w(X)_{i_0,j_1} \cdot w(X)_{i_0,j_2}$$
$$F_5(X) = f(X)_{i_0,i_1} \cdot w(X)_{i_0,j_1} \cdot w(X)_{i_0,j_2} \cdot h(X)_{j_0,i_1}$$
$$F_6(X) = v_{j_2,j_0} \cdot f(X)_{i_0,i_1} \cdot w(X)_{i_0,j_1} + v_{j_1,j_0} \cdot f(X)_{i_0,i_1} \cdot w(X)_{i_0,j_2}$$

- **Part 2** *For $i_0 \neq i_2, i_1 \neq i_2 \in [n]$, $j_1, j_2 \in [d]$*

$$\frac{\mathrm{d}c(X)}{\mathrm{d}x_{i_1,j_1}, \mathrm{d}x_{i_2,j_2}} = \sum_{i=1}^{3} G_i(X)$$

*where we have following definitions*

$$G_1(X) = 2s(X)_{i_0,j_0} \cdot f(X)_{i_0,i_1} \cdot f(X)_{i_0,i_2} \cdot w(X)_{i_0,j_2} \cdot w(X)_{i_0,j_1}$$
$$G_2(X) = -f(X)_{i_0,i_1} \cdot f(X)_{i_0,i_2} \cdot w(X)_{i_0,j_2} \cdot w(X)_{i_0,j_1} \cdot (h(X)_{j_0,i_2} + h(X)_{j_0,i_1})$$
$$G_3(X) = -f(X)_{i_0,i_1} \cdot f(X)_{i_0,i_2} \cdot (v_{j_2,j_0} \cdot w(X)_{i_0,j_1} + v_{j_1,j_0} \cdot w(X)_{i_0,j_2})$$

*Proof.* **Proof of Part 1.**

$$\frac{\mathrm{d}c(X)_{i_0,j_0}}{\mathrm{d}x_{i_1,j_1}, \mathrm{d}x_{i_2,j_2}}$$
$$= \frac{\mathrm{d}C_6}{\mathrm{d}x_{i_2,j_2}} + \frac{\mathrm{d}C_7}{\mathrm{d}x_{i_2,j_2}} + \frac{\mathrm{d}C_8}{\mathrm{d}x_{i_2,j_2}}$$
$$= -(\langle -f(X)_{i_0} \cdot f(X)_{i_0,i_1} \cdot \langle W_{j_2,*}, X_{*,i_0} \rangle + f(X)_{i_0} \circ (e_{i_1} \cdot \langle W_{j_2,*}, X_{*,i_0} \rangle), h(X)_{j_0} \rangle$$
$$+ f(X)_{i_0,i_1} \cdot v_{j_2,j_0}) \cdot f(X)_{i_0,i_1} \cdot \langle W_{j_1,*}, X_{*,i_0} \rangle$$
$$+ (-\langle f(X)_{i_0}, h(X)_{j_0} \rangle) \cdot (-f(X)_{i_0,i_1}^2 + f(X)_{i_0,i_1}) \cdot \langle W_{j_2,*}, X_{*,i_0} \rangle \cdot \langle W_{j_1,*}, X_{*,i_0} \rangle$$
$$(-f(X)_{i_0,i_2} + 1) \cdot f(X)_{i_0,i_1} \cdot \langle W_{j_2,*}, X_{*,i_0} \rangle \cdot h(X)_{j_0,i_1} \cdot \langle W_{j_1,*}, X_{*,i_0} \rangle$$
$$+ v_{j_2,j_0} \cdot f(X)_{i_0,i_1} \cdot \langle W_{j_1,*}, X_{*,i_0} \rangle$$
$$+ (-f(X)_{i_0,i_1}^2 + f(X)_{i_0,i_1}) \cdot \langle W_{j_2,*}, X_{*,i_0} \rangle \cdot v_{j_1,j_0}$$
$$= 2s(X)_{i_0,j_0} \cdot f(X)_{i_0,i_1}^2 \cdot w(X)_{i_0,j_2} \cdot w(X)_{i_0,j_1}$$
$$- 2f(X)_{i_0,i_1}^2 \cdot h(X)_{j_0,i_1} \cdot w(X)_{i_0,j_2} \cdot w(X)_{i_0,j_1}$$
$$- f(X)_{i_0,i_1}^2 \cdot v_{j_2,j_0} \cdot w(X)_{i_0,j_1} - f(X)_{i_0,i_1}^2 \cdot v_{j_1,j_0} \cdot w(X)_{i_0,j_2}$$
$$- s(X)_{i_0,j_0} \cdot f(X)_{i_0,i_1} \cdot w(X)_{i_0,j_1} \cdot w(X)_{i_0,j_2}$$
$$+ f(X)_{i_0,i_1} \cdot w(X)_{i_0,j_1} \cdot w(X)_{i_0,j_2} \cdot h(X)_{j_0,i_1}$$
$$+ v_{j_2,j_0} \cdot f(X)_{i_0,i_1} \cdot w(X)_{i_0,j_1} + v_{j_1,j_0} \cdot f(X)_{i_0,i_1} \cdot w(X)_{i_0,j_2}$$

where the first step follows from Lemma A.16, the second step follows from previous results in this section, the last step is a rearrangement.

**Proof of Part 2.**

$$\frac{\mathrm{d}c(X)_{i_0,j_0}}{\mathrm{d}x_{i_1,j_1}, \mathrm{d}x_{i_2,j_2}}$$
$$= \frac{\mathrm{d}C_6}{\mathrm{d}x_{i_2,j_2}} + \frac{\mathrm{d}C_7}{\mathrm{d}x_{i_2,j_2}} + \frac{\mathrm{d}C_8}{\mathrm{d}x_{i_2,j_2}}$$

$$
\begin{aligned}
&= - ((\langle -f(X)_{i_0} \cdot f(X)_{i_0,i_2} \cdot \langle W_{j_2,*}, X_{*,i_0} \rangle \\
&\quad + f(X)_{i_0} \circ (e_{i_2} \cdot \langle W_{j_2,*}, X_{*,i_0} \rangle)), h(X)_{j_0} \rangle + f(X)_{i_0,i_2} \cdot v_{j_2,j_0}) \cdot f(X)_{i_0,i_1} \cdot \langle W_{j_1,*}, X_{*,i_0} \rangle \\
&\quad + \langle f(X)_{i_0}, h(X)_{j_0} \rangle \cdot f(X)_{i_0,i_2} \cdot f(X)_{i_0,i_1} \cdot \langle W_{j_2,*}, X_{*,i_0} \rangle \cdot \langle W_{j_1,*}, X_{*,i_0} \rangle \\
&\quad - f(X)_{i_0,i_2} \cdot f(X)_{i_0,i_1} \cdot \langle W_{j_2,*}, X_{*,i_0} \rangle \cdot h(X)_{j_0,i_1} \cdot \langle W_{j_1,*}, X_{*,i_0} \rangle \\
&\quad - f(X)_{i_0,i_2} \cdot f(X)_{i_0,i_1} \cdot \langle W_{j_2,*}, X_{*,i_0} \rangle \cdot v_{j_1,j_0} \\
&= 2s(X)_{i_0,j_0} \cdot f(X)_{i_0,i_1} \cdot f(X)_{i_0,i_2} \cdot w(X)_{i_0,j_2} \cdot w(X)_{i_0,j_1} \\
&\quad - f(X)_{i_0,i_1} \cdot f(X)_{i_0,i_2} \cdot h(X)_{j_0,i_2} \cdot w(X)_{i_0,j_2} \cdot w(X)_{i_0,j_1} \\
&\quad - f(X)_{i_0,i_1} \cdot f(X)_{i_0,i_2} \cdot w(X)_{i_0,j_1} \cdot w(X)_{i_0,j_2} \cdot h(X)_{j_0,i_1} \\
&\quad - f(X)_{i_0,i_1} \cdot f(X)_{i_0,i_2} \cdot v_{j_2,j_0} \cdot w(X)_{i_0,j_1} - f(X)_{i_0,i_1} \cdot f(X)_{i_0,i_2} \cdot v_{j_1,j_0} \cdot w(X)_{i_0,j_2}
\end{aligned}
$$

where the first step follows from Lemma A.16, the second step follows from Lemma C.6, the third step follows from Part 2 of Lemma C.7, the last step follows from Lemma C.8.

Notice that, by our construction, **Part 1** should be symmetric w.r.t. $j_1, j_2$, **Part 2** should be symmetric w.r.t. $i_1, i_2$, which are all satisfied. □

## D HESSIAN REFORMULATION

In this section, we provide a reformulation of Hessian formula, which simplifies our calculation and analysis. In Section D.1 we show the way we split the Hessian. In Section D.2 we show the decomposition when $i_0 = i_1 = i_2$.

### D.1 HESSIAN SPLIT

**Definition D.1** (Hessian of functions of matrix). *We define the Hessian of $c(X)_{i_0,j_0}$ by considering its Hessian with respect to $x = \mathrm{vec}(X)$. This means that, $\nabla^2 c(X)_{i_0,j_0}$ is a $nd \times nd$ matrix with its $(i_1 \cdot j_1, i_2 \cdot j_2)$-th entry being*

$$
\frac{\mathrm{d}c(X)_{i_0,j_0}}{\mathrm{d}x_{i_1,j_2} x_{i_2,j_2}}
$$

**Definition D.2** (Hessian split). *We split the hessian of $c(X)_{i_0,j_0}$ into following cases*

- *Part 1: $i_0 = i_1 = i_2$ : $H_1^{(i_1,i_2)}$*

- *Part 2: $i_0 = i_1, i_0 \neq i_2$ : $H_2^{(i_1,i_2)}$*

- *Part 3: $i_0 \neq i_1, i_0 = i_2$ : $H_3^{(i_1,i_2)}$*

- *Part 4: $i_0 \neq i_1, i_0 \neq i_2, i_1 = i_2$: $H_4^{(i_1,i_2)}$*

- *Part 5: $i_0 \neq i_1, i_0 \neq i_2, i_1 \neq i_2$: $H_5^{(i_1,i_2)}$*

*In above, $H_i^{(i_1,i_2)}$ is a $d \times d$ matrix with its $j_1, j_2$-th entry being*

$$
\frac{\mathrm{d}c(X)_{i_0,j_0}}{\mathrm{d}x_{i_1,j_2} x_{i_2,j_2}}
$$

Utilizing above definitions, we split the Hessian to a $n \times n$ partition with its $i_1, i_2$-th component being $H_i(i_1, i_2)$ based on above definition.

**Definition D.3.** *We define $\nabla^2 c(X)_{i_0,j_0}$ to be as following*

$$
\begin{bmatrix}
H_4^{(1,1)} & H_5^{(1,2)} & H_5^{(1,3)} & \cdots & H_5^{(1,i_0-1)} & H_3^{(1,i_0)} & H_5^{(1,i_0+1)} & \cdots & H_5^{(1,n)} \\
H_5^{(2,1)} & H_4^{(2,2)} & H_5^{(2,3)} & \cdots & H_5^{(2,i_0-1)} & H_3^{(2,i_0)} & H_5^{(2,i_0+1)} & \cdots & H_5^{(2,n)} \\
H_5^{(3,1)} & H_5^{(3,2)} & H_4^{(3,3)} & \cdots & H_5^{(3,i_0-1)} & H_3^{(3,i_0)} & H_5^{(3,i_0+1)} & \cdots & H_5^{(3,n)} \\
\vdots & \vdots & \vdots & \ddots & \vdots & \vdots & \vdots & \ddots & \vdots \\
H_2^{(i_0,1)} & H_2^{(i_0,2)} & H_2^{(i_0,3)} & \cdots & H_2^{(i_0,i_0-1)} & H_1^{(i_0,i_0)} & H_2^{(i_0,i_0+1)} & \cdots & H_2^{(i_0,n)} \\
H_5^{(i_0+1,1)} & H_5^{(i_0+1,2)} & H_5^{(i_0+1,3)} & \cdots & H_5^{(i_0+1,i_0-1)} & H_3^{(i_0+1,i_0)} & H_4^{(i_0+1,i_0+1)} & \cdots & H_5^{(i_0+1,n)} \\
\vdots & \vdots & \vdots & \ddots & \vdots & \vdots & \vdots & \ddots & \vdots \\
H_5^{(n,1)} & H_5^{(n,2)} & H_5^{(n,3)} & \cdots & H_5^{(n,i_0-1)} & H_3^{(n,i_0)} & H_5^{(n,i_0+1)} & \cdots & H_4^{(n,n)}
\end{bmatrix}
$$

## D.2 Decomposition Hessian : Part 1

**Lemma D.4** (Helpful lemma). *Under following conditions*

- *Let $z(X)_{i_0} := W^\top X \cdot f(X)_{i_0}$*

- *Let $w(X)_{i_0,*} := W X_{*,i_0}$*

*we have*

- *Part 1: $w(X)_{i_0,j_1} = e_{j_1}^\top \cdot w(X)_{i_0,*}$*

- *Part 2: $z(X)_{i_0,j_1} = e_{j_1}^\top \cdot z(X)_{i_0}$*

*Proof.* **Proof of Part 1**

$$
\begin{aligned}
w(X)_{i_0,j_1} &= \langle W_{j_1,*}, X_{*,i_0} \rangle \\
&= W_{j_1,*}^\top X_{*,i_0} \\
&= e_{j_1}^\top \cdot W X_{*,i_0} \\
&= e_{j_1}^\top \cdot w(X)_{i_0,*}
\end{aligned}
$$

where the first step is by the definition of $w(X)_{i_0,j_1}$ the 2nd and 3rd step are from linear algebra facts, the 4th step is by the definition of $w(X)_{i_0,*}$.

**Proof of Part 2**

$$
\begin{aligned}
z(X)_{i_0,j_1} &= \langle f(X)i_0, X^\top W_{*,j_1} \rangle \\
&= (X^\top W_{*,j_1})^\top f(X)_{i_0} \\
&= W_{*,j_1}^\top X \cdot f(X)_{i_0} \\
&= e_{j_1}^\top \cdot W^\top X \cdot f(X)_{i_0} \\
&= e_{j_1}^\top \cdot z(X)_{i_0}
\end{aligned}
$$

where the first step is by the definition of $w(X)_{i_0,j_1}$ the 2nd, 3rd, and the 4th step are from linear algebra facts, the 5th step is by the definition of $w(X)_{i_0,*}$. $\square$

**Lemma D.5.** *Under following conditions*

- *Let $D_i(X)$ be defined as Lemma B.15*

- *Let $z(X)_{i_0} := W^\top X \cdot f(X)_{i_0}$*

- *Let $w(X)_{i_0,*} := W X_{*,i_0}$*

*we have*

$$
D_1(X) = e_{j_1}^\top \cdot w(X)_{i_0,*} \cdot 2s(X)_{i_0,j_0} \cdot f(X)_{i_0,i_0}^2 \cdot w(X)_{i_0,*}^\top \cdot e_{j_2}
$$

$$D_2(X) = e_{j_1}^\top \cdot (w(X)_{i_0,*} \cdot 2f(X)_{i_0,i_0} \cdot s(X)_{i_0,j_0} \cdot z(X)_{i_0}^\top$$
$$+ z(X)_{i_0} \cdot 2f(X)_{i_0,i_0} \cdot s(X)_{i_0,j_0} \cdot w(X)_{i_0,*}^\top) \cdot e_{j_2}$$

$$D_3(X) = -e_{j_1}^\top \cdot w(X)_{i_0,*} \cdot f(X)_{i_0,i_0}^2 \cdot h(X)_{j_0,i_0} \cdot w(X)_{i_0,*}^\top \cdot e_{j_2}$$

$$D_4(X) = -e_{j_1}^\top \cdot W^\top \cdot f(X)_{i_0,i_0} \cdot X \cdot \mathrm{diag}(f(X)_{i_0}) \cdot h(X)_{j_0} \cdot w(X)_{i_0,*}^\top \cdot e_{j_2}$$
$$- e_{j_1}^\top \cdot w(X)_{i_0,*} \cdot f(X)_{i_0,i_0} \cdot h(X)_{j_0}^\top \cdot \mathrm{diag}(f(X)_{i_0}) \cdot X^\top \cdot W \cdot e_{j_2}$$

$$D_5(X) = -e_{j_1}^\top \cdot (w(X)_{i_0,*} \cdot f(X)_{i_0,i_0}^2 \cdot V_{*,j_0}^\top + V_{*,j_0} \cdot f(X)_{i_0,i_0}^2 \cdot w(X)_{i_0,*}^\top) \cdot e_{j_2}$$

$$D_6(X) = -e_{j_1}^\top \cdot w(X)_{i_0,*} \cdot s(X)_{i_0,j_0} \cdot f(X)_{i_0,i_0} \cdot w(X)_{i_0,*}^\top \cdot e_{j_2}$$

$$D_7(X) = -e_{j_1}^\top \cdot w(X)_{i_0,*} \cdot s(X)_{i_0,j_0} \cdot f(X)_{i_0,i_0} \cdot X_{*,i_0}^\top \cdot W \cdot e_{j_2}$$
$$- e_{j_1}^\top \cdot W^\top \cdot X_{*,i_0} \cdot s(X)_{i_0,j_0} \cdot f(X)_{i_0,i_0} \cdot w(X)_{i_0,*}^\top \cdot e_{j_2}$$

$$D_8(X) = e_{j_1}^\top \cdot s(X)_{i_0,j_0} \cdot f(X)_{i_0,i_0} \cdot (W^\top - W) \cdot e_{j_2}$$

$$D_9(X) = e_{j_1}^\top \cdot z(X)_{i_0} \cdot s(X)_{i_0,j_0} \cdot z(X)_{i_0}^\top \cdot e_{j_2}$$

$$D_{10}(X) = -e_{j_1}^\top \cdot (z(X)_{i_0} \cdot f(X)_{i_0,i_0} \cdot h(X)_{j_0,i_0} \cdot w(X)_{i_0,*}^\top$$
$$+ w(X)_{i_0,*} \cdot f(X)_{i_0,i_0} \cdot h(X)_{j_0,i_0} \cdot z(X)_{i_0}^\top) \cdot e_{j_2}$$

$$D_{11}(X) = -e_{j_1}^\top \cdot (z(X)_{i_0} \cdot h(X)_{j_0}^\top \cdot \mathrm{diag}(f(X)_{i_0}) \cdot X^\top \cdot W$$
$$+ W^\top \cdot X \cdot \mathrm{diag}(f(X)_{i_0}) \cdot h(X)_{j_0} \cdot z(X)_{i_0}^\top) \cdot e_{j_2}$$

$$D_{12}(X) = -e_{j_1}^\top \cdot (z(X)_{i_0} \cdot f(X)_{i_0,i_0} \cdot V_{*,j_0}^\top + V_{*,j_0} \cdot f(X)_{i_0,i_0} \cdot z(X)_{i_0}^\top) \cdot e_{j_2}$$

$$D_{13}(X) = e_{j_1}^\top \cdot z(X)_{i_0} \cdot s(X)_{i_0,j_0} \cdot f(X)_{i_0,i_0} \cdot z(X)_{i_0}^\top \cdot e_{j_2}$$

$$D_{14}(X) = -e_{j_1}^\top \cdot W^\top \cdot X \cdot s(X)_{i_0,j_0} \cdot \mathrm{diag}(f(X)_{i_0}) \cdot X^\top \cdot W \cdot e_{j_2}$$

$$D_{15}(X) = -e_{j_1}^\top \cdot w(X)_{i_0,*} \cdot f(X)_{i_0,i_0}^2 \cdot h(X)_{j_0,i_0} \cdot w(X)_{i_0,*}^\top \cdot e_{j_2}$$

$$D_{16}(X) = e_{j_1}^\top \cdot w(X)_{i_0,*} \cdot f(X)_{i_0,i_0} \cdot h(X)_{j_0,i_0} \cdot w(X)_{i_0,*}^\top \cdot e_{j_2}$$

$$D_{17}(X) = e_{j_1}^\top \cdot (w(X)_{i_0,*} \cdot f(X)_{i_0,i_0} \cdot X_{*,i_0}^\top \cdot h(X)_{j_0,i_0} \cdot W$$
$$+ W^\top \cdot X_{*,i_0} \cdot f(X)_{i_0,i_0} \cdot h(X)_{j_0,i_0} \cdot w(X)_{i_0}) \cdot e_{j_2}$$

$$D_{18}(X) = e_{j_1}^\top \cdot (w(X)_{i_0,*} f(X)_{i_0,i_0} \cdot V_{j_2,*}^\top + V_{j_1,*}^\top \cdot f(X)_{i_0,i_0} \cdot w(X)_{i_0,*}^\top) \cdot e_{j_2}$$

$$D_{19}(X) = e_{j_1}^\top \cdot f(X)_{i_0,i_0} \cdot h(X)_{i_0,i_0} \cdot (W + W^\top) \cdot e_{j_2}$$

$$D_{20}(X) := e_{j_1}^\top \cdot W^\top \cdot X \cdot \mathrm{diag}(f(X)_{i_0}) \cdot \mathrm{diag}(h(X)_{j_0}) \cdot X^\top \cdot W \cdot e_{j_2}$$

$$D_{21}(X) := e_{j_1}^\top \cdot (W^\top \cdot X_{*,i_0} \cdot f(X)_{i_0,i_0} \cdot V_{*,j_0}^\top + V_{*,j_0} \cdot f(X)_{i_0,i_0} \cdot X_{*,i_0}^\top \cdot W) \cdot e_{j_2}$$

*Proof.* This lemma is followed by Lemma D.4 and linear algebra facts. □

Based on above auxiliary lemma, we have following definition.

**Definition D.6.** *Under following conditions*

- *Let $z(X)_{i_0} := W^\top X \cdot f(X)_{i_0}$*

- *Let $w(X)_{i_0,*} := W X_{*,i_0}$*

*We present the **Case 1** component of Hessian $c(X)_{i_0,j_0}$ to be*

$$H_1^{(i_0,i_0)}(X) := B(X)$$

*where we have*

$$B(X) := \sum_{i=1}^{21} B_i(X)$$
$$B_1(X) := w(X)_{i_0,*} \cdot 2s(X)_{i_0,j_0} \cdot f(X)_{i_0,i_0}^2 \cdot w(X)_{i_0,*}^\top$$

$$B_2(X) := w(X)_{i_0,*} \cdot 2f(X)_{i_0,i_0} \cdot s(X)_{i_0,j_0} \cdot z(X)_{i_0}^\top$$
$$+ z(X)_{i_0} \cdot 2f(X)_{i_0,i_0} \cdot s(X)_{i_0,j_0} \cdot w(X)_{i_0,*}^\top$$
$$B_3(X) := -w(X)_{i_0,*} \cdot f(X)_{i_0,i_0}^2 \cdot h(X)_{j_0,i_0} \cdot w(X)_{i_0,*}^\top$$
$$B_4(X) := -W^\top \cdot f(X)_{i_0,i_0} \cdot X \cdot \mathrm{diag}(f(X)_{i_0}) \cdot h(X)_{j_0} \cdot w(X)_{i_0,*}^\top$$
$$- w(X)_{i_0,*} \cdot f(X)_{i_0,i_0} \cdot h(X)_{j_0}^\top \cdot \mathrm{diag}(f(X)_{i_0}) \cdot X^\top \cdot W$$
$$B_5(X) := -w(X)_{i_0,*} \cdot f(X)_{i_0,i_0}^2 \cdot V_{*,j_0}^\top - V_{*,j_0} \cdot f(X)_{i_0,i_0}^2 \cdot w(X)_{i_0,*}^\top$$
$$B_6(X) := -w(X)_{i_0,*} \cdot s(X)_{i_0,j_0} \cdot f(X)_{i_0,i_0} \cdot w(X)_{i_0,*}^\top$$
$$B_7(X) := -w(X)_{i_0,*} \cdot s(X)_{i_0,j_0} \cdot f(X)_{i_0,i_0} \cdot X_{*,i_0}^\top \cdot W$$
$$- W^\top \cdot X_{*,i_0} \cdot s(X)_{i_0,j_0} \cdot f(X)_{i_0,i_0} \cdot w(X)_{i_0,*}^\top$$
$$B_8(X) := s(X)_{i_0,j_0} \cdot f(X)_{i_0,i_0} \cdot (W^\top - W)$$
$$B_9(X) := z(X)_{i_0} \cdot s(X)_{i_0,j_0} \cdot z(X)_{i_0}^\top$$
$$B_{10}(X) := -z(X)_{i_0} \cdot f(X)_{i_0,i_0} \cdot h(X)_{j_0,i_0} \cdot w(X)_{i_0,*}^\top$$
$$- w(X)_{i_0,*} \cdot f(X)_{i_0,i_0} \cdot h(X)_{j_0,i_0} \cdot z(X)_{i_0}^\top$$
$$B_{11}(X) := -z(X)_{i_0} \cdot (h(X)_{j_0}^\top \cdot \mathrm{diag}(f(X)_{i_0}) \cdot X^\top \cdot W$$
$$- W^\top \cdot X \cdot \mathrm{diag}(f(X)_{i_0}) \cdot h(X)_{j_0} \cdot z(X)_{i_0}^\top$$
$$B_{12}(X) := -z(X)_{i_0} \cdot f(X)_{i_0,i_0} \cdot V_{*,j_0}^\top + V_{*,j_0} \cdot f(X)_{i_0,i_0} \cdot z(X)_{i_0}^\top$$
$$B_{13}(X) := z(X)_{i_0} \cdot s(X)_{i_0,j_0} \cdot f(X)_{i_0,i_0} \cdot z(X)_{i_0}^\top$$
$$B_{14}(X) := -W^\top \cdot X \cdot s(X)_{i_0,j_0} \cdot \mathrm{diag}(f(X)_{i_0}) \cdot X^\top \cdot W$$
$$B_{15}(X) := -w(X)_{i_0,*} \cdot f(X)_{i_0,i_0}^2 \cdot h(X)_{j_0,i_0} \cdot w(X)_{i_0,*}^\top$$
$$B_{16}(X) := w(X)_{i_0,*} \cdot f(X)_{i_0,i_0} \cdot h(X)_{j_0,i_0} \cdot w(X)_{i_0,*}^\top$$
$$B_{17}(X) := w(X)_{i_0,*} \cdot f(X)_{i_0,i_0} \cdot X_{*,i_0}^\top \cdot h(X)_{j_0,i_0} \cdot W$$
$$+ W^\top \cdot X_{*,i_0} \cdot f(X)_{i_0,i_0} \cdot h(X)_{j_0,i_0} \cdot w(X)_{i_0}$$
$$B_{18}(X) := w(X)_{i_0,*} \cdot f(X)_{i_0,i_0} \cdot V_{j_2,*}^\top + V_{j_1,*}^\top \cdot f(X)_{i_0,i_0} \cdot w(X)_{i_0,*}^\top$$
$$B_{19}(X) := f(X)_{i_0,i_0} \cdot h(X)_{i_0,i_0} \cdot (W + W^\top)$$
$$B_{20}(X) := W^\top \cdot X \cdot \mathrm{diag}(f(X)_{i_0}) \cdot \mathrm{diag}(h(X)_{j_0}) \cdot X^\top$$
$$B_{21}(X) := W^\top \cdot X_{*,i_0} \cdot f(X)_{i_0,i_0} \cdot V_{*,j_0}^\top + V_{*,j_0} \cdot f(X)_{i_0,i_0} \cdot X_{*,i_0}^\top \cdot W$$

## D.3 DECOMPOSITION HESSIAN: PART 2 AND PART 3

**Lemma D.7.** *Under following conditions*

- *Let $E_i(X)$ be defined as Lemma B.15*
- *Let $z(X)_{i_0} := W^\top X \cdot f(X)_{i_0}$*
- *Let $w(X)_{i_0,*} := W X_{*,i_0}$*

*we have*

$$E_1(X) = e_{j_1}^\top \cdot w(X)_{i_0,*} \cdot 2s(X)_{i_0,j_0} \cdot f(X)_{i_0,i_2} \cdot f(X)_{i_0,i_0} \cdot w(X)_{i_0,*}^\top \cdot e_{j_2}$$
$$E_2(X) = -e_{j_1}^\top \cdot w(X)_{i_0,*} \cdot 2f(X)_{i_0,i_2} \cdot h(X)_{j_0,i_2} \cdot f(X)_{i_0,i_0} \cdot w(X)_{i_0,*}^\top \cdot e_{j_2}$$
$$E_3(X) = -e_{j_1}^\top \cdot w(X)_{i_0,*} \cdot f(X)_{i_0,i_2} \cdot f(X)_{i_0,i_0} \cdot V_{*,j_0}^\top \cdot e_{j_2}$$
$$E_4(X) = e_{j_1}^\top \cdot z(X)_{i_0} \cdot s(X)_{i_0,j_0} \cdot f(X)_{i_0,i_2} \cdot w(X)_{i_0,*}^\top \cdot e_{j_2}$$
$$E_5(X) = -e_{j_1}^\top \cdot z(X)_{i_0} \cdot f(X)_{i_0,i_2} \cdot h(X)_{j_0,i_2} \cdot w(X)_{i_0,*}^\top \cdot e_{j_2}$$
$$E_6(X) = -e_{j_1}^\top \cdot z(X)_{i_0} \cdot f(X)_{i_0,i_2} \cdot V_{*,j_0}^\top \cdot e_{j_2}$$

$$E_7(X) = e_{j_1}^\top \cdot z(X)_{i_0} \cdot s(X)_{i_0,j_0} \cdot f(X)_{i_0,i_0} \cdot w(X)_{i_0,*}^\top \cdot e_{j_2}$$

$$E_8(X) = -e_{j_1}^\top \cdot w(X)_{i_0,*} \cdot s(X)_{i_0,j_0} \cdot f(X)_{i_0,i_0} \cdot w(X)_{i_0,*}^\top \cdot e_{j_2}$$

$$E_9(X) = -e_{j_1}^\top \cdot W^\top \cdot s(X)_{i_0,j_0} \cdot f(X)_{i_0,i_0} \cdot e_{j_2}$$

$$E_{10}(X) = -e_{j_1}^\top \cdot w(X)_{i_0,*} \cdot f(X)_{i_0,i_0} \cdot f(X)_{i_0,i_2} \cdot h(X)_{j_0,i_0} \cdot w(X)_{i_0,*}^\top \cdot e_{j_2}$$

$$E_{11}(X) = -e_{j_1}^\top \cdot W^\top \cdot X \cdot \mathrm{diag}(f(X)_{i_0}) \cdot h(X)_{j_0} \cdot f(X)_{i_0,i_2} \cdot w(X)_{i_0,*}^\top \cdot e_{j_2}$$

$$E_{12}(X) = e_{j_1}^\top \cdot W^\top \cdot X_{*,i_2} \cdot f(X)_{i_0,i_2} \cdot h(X)_{j_0,i_2} \cdot w(X)_{i_0,*}^\top \cdot e_{j_2}$$

$$E_{13}(X) = e_{j_1}^\top \cdot W^\top f(X)_{i_0,i_2} \cdot h(X)_{j_0,i_2} \cdot e_{j_2}$$

$$E_{14}(X) = e_{j_1}^\top \cdot W^\top \cdot X_{*,i_2} \cdot f(X)_{i_0,i_2} \cdot V_{*,j_0}^\top \cdot e_{j_2}$$

$$E_{15}(X) = -e_{j_1}^\top \cdot V_{*,j_0} \cdot f(X)_{i_0,i_0} \cdot f(X)_{i_0,i_2} \cdot w(X)_{i_0,*}^\top \cdot e_{j_2}$$

*Proof.* This lemma is followed by Lemma D.4 and linear algebra facts. □

Based on above auxiliary lemma, we have following definition.

**Definition D.8.** *Under following conditions*

- *Let $z(X)_{i_0} := W^\top X \cdot f(X)_{i_0}$*

- *Let $w(X)_{i_0,*} := W X_{*,i_0}$*

*We present the* **Case 2** *component of Hessian $c(X)_{i_0,j_0}$ to be*

$$H_2^{(i_0,i_2)}(X) := J(X)$$

*where we have*

$$J(X) := \sum_{i=1}^{15} J_i(X)$$

$$J_1(X) := w(X)_{i_0,*} \cdot 2s(X)_{i_0,j_0} \cdot f(X)_{i_0,i_2} \cdot f(X)_{i_0,i_0} \cdot w(X)_{i_0,*}^\top$$

$$J_2(X) := -w(X)_{i_0,*} \cdot 2f(X)_{i_0,i_2} \cdot h(X)_{j_0,i_2} \cdot f(X)_{i_0,i_0} \cdot w(X)_{i_0,*}^\top$$

$$J_3(X) := -w(X)_{i_0,*} \cdot f(X)_{i_0,i_2} \cdot f(X)_{i_0,i_0} \cdot V_{*,j_0}^\top$$

$$J_4(X) := z(X)_{i_0} \cdot s(X)_{i_0,j_0} \cdot f(X)_{i_0,i_2} \cdot w(X)_{i_0,*}^\top$$

$$J_5(X) := -z(X)_{i_0} \cdot f(X)_{i_0,i_2} \cdot h(X)_{j_0,i_2} \cdot w(X)_{i_0,*}^\top$$

$$J_6(X) := -z(X)_{i_0} \cdot f(X)_{i_0,i_2} \cdot V_{*,j_0}^\top$$

$$J_7(X) := z(X)_{i_0} \cdot s(X)_{i_0,j_0} \cdot f(X)_{i_0,i_0} \cdot w(X)_{i_0,*}^\top$$

$$J_8(X) := -w(X)_{i_0,*} \cdot s(X)_{i_0,j_0} \cdot f(X)_{i_0,i_0} \cdot w(X)_{i_0,*}^\top$$

$$J_9(X) := -W^\top \cdot s(X)_{i_0,j_0} \cdot f(X)_{i_0,i_0}$$

$$J_{10}(X) := -w(X)_{i_0,*} \cdot f(X)_{i_0,i_0} \cdot f(X)_{i_0,i_2} \cdot h(X)_{j_0,i_0} \cdot w(X)_{i_0,*}^\top$$

$$J_{11}(X) := -W^\top \cdot X \cdot \mathrm{diag}(f(X)_{i_0}) \cdot h(X)_{j_0} \cdot f(X)_{i_0,i_2} \cdot w(X)_{i_0,*}^\top$$

$$J_{12}(X) := W^\top \cdot X_{*,i_2} \cdot f(X)_{i_0,i_2} \cdot h(X)_{j_0,i_2} \cdot w(X)_{i_0,*}^\top$$

$$J_{13}(X) := W^\top f(X)_{i_0,i_2} \cdot h(X)_{j_0,i_2}$$

$$J_{14}(X) := W^\top \cdot X_{*,i_2} \cdot f(X)_{i_0,i_2} \cdot V_{*,j_0}^\top$$

$$J_{15}(X) := -V_{*,j_0} \cdot f(X)_{i_0,i_0} \cdot f(X)_{i_0,i_2} \cdot w(X)_{i_0,*}^\top$$

Next, we define the third case by the symmetricity of Hessian.

**Definition D.9.** *We present the* **Case 3** *component of Hessian $c(X)_{i_0,j_0}$ to be*

$$H_3^{(i,i_0)}(X) := H_2^{(i_0,i)}(X)$$

### D.4 DECOMPOSITION HESSIAN : PART 4

**Lemma D.10.** *Under following conditions*

- *Let $F_i(X)$ be defined as Lemma C.9*

- *Let $z(X)_{i_0} := W^\top X \cdot f(X)_{i_0}$*

- *Let $w(X)_{i_0,*} := W X_{*,i_0}$*

*we have*

$$F_1(X) = e_{j_1}^\top \cdot w(X)_{i_0,*} \cdot 2s(X)_{i_0,j_0} \cdot f(X)_{i_0,i_1}^2 \cdot w(X)_{i_0,*}^\top \cdot e_{j_2}$$

$$F_2(X) = -e_{j_1}^\top \cdot w(X)_{i_0,*} \cdot f(X)_{i_0,i_1}^2 \cdot h(X)_{j_0,i_1} \cdot w(X)_{i_0,*}^\top \cdot e_{j_2}$$

$$F_3(X) = -e_{j_1}^\top \cdot (w(X)_{i_0,*} \cdot f(X)_{i_0,i_1}^2 \cdot V_{*,j_0}^\top + V_{*,j_0} \cdot f(X)_{i_0,i_1}^2 \cdot w(X)_{i_0,*}^\top) \cdot e_{j_2}$$

$$F_4(X) = -e_{j_1}^\top \cdot w(X)_{i_0,*} \cdot s(X)_{i_0,j_0} \cdot f(X)_{i_0,i_1} \cdot w(X)_{i_0,*}^\top \cdot e_{j_2}$$

$$F_5(X) = e_{j_1}^\top \cdot w(X)_{i_0,*} \cdot f(X)_{i_0,i_1} \cdot h(X)_{j_0,i_1} \cdot w(X)_{i_0,*}^\top \cdot e_{j_2}$$

$$F_6(X) = e_{j_1}^\top \cdot (w(X)_{i_0,*} \cdot f(X)_{i_0,i_1} \cdot V_{*,j_0}^\top + V_{*,j_0} \cdot f(X)_{i_0,i_1} \cdot w(X)_{i_0,*}^\top) \cdot e_{j_2}$$

*Proof.* This lemma is followed by Lemma D.4 and linear algebra facts. □

Based on above auxiliary lemma, we have following definition.

**Definition D.11.** *Under following conditions*

- *Let $z(X)_{i_0} := W^\top X \cdot f(X)_{i_0}$*

- *Let $w(X)_{i_0,*} := W X_{*,i_0}$*

*We present the* **Case 4** *component of Hessian $c(X)_{i_0,j_0}$ to be*

$$H_4^{(i_1,i_1)}(X) := K(X)$$

*where we have*

$$K(X) := \sum_{i=1}^6 K_i(X)$$

$$K_1(X) := w(X)_{i_0,*} \cdot 2s(X)_{i_0,j_0} \cdot f(X)_{i_0,i_1}^2 \cdot w(X)_{i_0,*}^\top$$

$$K_2(X) := -w(X)_{i_0,*} \cdot f(X)_{i_0,i_1}^2 \cdot h(X)_{j_0,i_1} \cdot w(X)_{i_0,*}^\top$$

$$K_3(X) := -w(X)_{i_0,*} \cdot f(X)_{i_0,i_1}^2 \cdot V_{*,j_0}^\top - V_{*,j_0} \cdot f(X)_{i_0,i_1}^2 \cdot w(X)_{i_0,*}^\top$$

$$K_4(X) := -w(X)_{i_0,*} \cdot s(X)_{i_0,j_0} \cdot f(X)_{i_0,i_1} \cdot w(X)_{i_0,*}^\top$$

$$K_5(X) := w(X)_{i_0,*} \cdot f(X)_{i_0,i_1} \cdot h(X)_{j_0,i_1} \cdot w(X)_{i_0,*}^\top$$

$$K_6(X) := w(X)_{i_0,*} \cdot f(X)_{i_0,i_1} \cdot V_{*,j_0}^\top + V_{*,j_0} \cdot f(X)_{i_0,i_1} \cdot w(X)_{i_0,*}^\top$$

### D.5 DECOMPOSITION HESSIAN : PART 5

**Lemma D.12.** *Under following conditions*

- *Let $G_i(X)$ be defined as Lemma C.9*

- *Let $z(X)_{i_0} := W^\top X \cdot f(X)_{i_0}$*

- *Let $w(X)_{i_0,*} := W X_{*,i_0}$*

*we have*

$$G_1(X) = e_{j_1}^\top \cdot w(X)_{i_0,*} \cdot 2s(X)_{i_0,j_0} \cdot f(X)_{i_0,i_1} \cdot f(X)_{i_0,i_2} \cdot w(X)_{i_0,*}^\top \cdot e_{j_2}$$

$$G_2(X) = -e_{j_1}^\top \cdot w(X)_{i_0,*} \cdot f(X)_{i_0,i_1} \cdot f(X)_{i_0,i_2} \cdot (h(X)_{j_0,i_2} + h(X)_{j_0,i_1}) \cdot w(X)_{i_0,*}^\top \cdot e_{j_2}$$

$$G_3(X) = -e_{j_1}^\top \cdot f(X)_{i_0,i_1} \cdot f(X)_{i_0,i_2} \cdot (w(X)_{i_0,*} \cdot V_{*,j_0}^\top + V_{*,j_0} \cdot w(X)_{*,j_2}^\top) \cdot e_{j_2}$$

*Proof.* This lemma is followed by Lemma D.4 and linear algebra facts. □

Based on above auxiliary lemma, we have following definition.

**Definition D.13.** *Under following conditions*

- *Let $z(X)_{i_0} := W^\top X \cdot f(X)_{i_0}$*

- *Let $w(X)_{i_0,*} := W X_{*,i_0}$*

*We present the* **Case 5** *component of Hessian $c(X)_{i_0,j_0}$ to be*

$$H_5^{(i_1,i_2)}(X) := N(X)$$

*where we have*

$$N(X) := \sum_{i=1}^{3} N_i(X)$$

$$N_1(X) := w(X)_{i_0,*} \cdot 2s(X)_{i_0,j_0} \cdot f(X)_{i_0,i_1} \cdot f(X)_{i_0,i_2} \cdot w(X)_{i_0,*}^\top$$

$$N_2(X) := -w(X)_{i_0,*} \cdot f(X)_{i_0,i_1} \cdot f(X)_{i_0,i_2} \cdot (h(X)_{j_0,i_2} + h(X)_{j_0,i_1}) \cdot w(X)_{i_0,*}^\top$$

$$N_3(X) := -f(X)_{i_0,i_1} \cdot f(X)_{i_0,i_2} \cdot (w(X)_{i_0,*} \cdot V_{*,j_0}^\top + V_{*,j_0} \cdot w(X)_{*,j_2}^\top)$$

# E  HESSIAN OF LOSS FUNCTION

In this section, we provide the Hessian of our loss function.

**Lemma E.1** (A single entry). *Under following conditions*

- *Let $L(X)$ be defined as Definition A.9*

*we have*

$$\frac{\mathrm{d}L(X)}{\mathrm{d}x_{i_1,j_1}x_{i_2,j_2}} = \sum_{i_0=1}^{n} \sum_{j_0=1}^{d} \frac{\mathrm{d}c(X)_{i_0,j_0}}{\mathrm{d}x_{i_1,j_1}} \cdot \frac{\mathrm{d}c(X)_{i_0,j_0}}{\mathrm{d}x_{i_1,j_2}} + c(X)_{i_0,j_0} \cdot \frac{\mathrm{d}c(X)_{i_0,j_0}}{\mathrm{d}x_{i_1,j_1}x_{i_2,j_2}}$$

*Proof.* **Proof of Part 1:** $i_1 = i_2$

$$\frac{\mathrm{d}L(X)}{\mathrm{d}x_{i_1,j_1}x_{i_2,j_2}} = \frac{\mathrm{d}}{\mathrm{d}x_{i_2,j_2}}(\sum_{i_0=1}^{n} \sum_{j_0=1}^{d} c(X)_{i_0,j_0} \cdot \frac{\mathrm{d}c(X)_{i_0,j_0}}{\mathrm{d}x_{i_1,j_1}})$$

$$= \sum_{i_0=1}^{n} \sum_{j_0=1}^{d} \frac{\mathrm{d}c(X)_{i_0,j_0}}{\mathrm{d}x_{i_1,j_1}} \cdot \frac{\mathrm{d}c(X)_{i_0,j_0}}{\mathrm{d}x_{i_2,j_2}} + c(X)_{i_0,j_0} \cdot \frac{\mathrm{d}c(X)_{i_0,j_0}}{\mathrm{d}x_{i_1,j_1}x_{i_2,j_2}}$$

where the first step is given by chain rule, and the 2nd step are given by product rule. □

**Lemma E.2** (Matrix Representation of Hessian). *Under following conditions*

- *Let $c(X)_{i_0,j_0}$ be defined as Definition A.8*

- *Let $L(X)$ be defined as Definition A.9*

*we have*

$$\nabla^2 L(X) = \sum_{i_0=1}^{n} \sum_{j_0=1}^{d} \nabla c(X)_{i_0,j_0} \cdot \nabla c(X)_{i_0,j_0}^\top + c(X)_{i_0,j_0} \cdot \nabla^2 c(X)_{i_0,j_0}$$

*Proof.* This is directly given by the single-entry representation in Lemma E.1. □

## F BOUNDS FOR BASIC FUNCTIONS

In this section, we prove the upper bound for each function, with following assumption about the domain of parameters. In Section F.1 we bound the basic terms. In Section F.2 we bound the gradient of $f(X)_{i_0}$. In Section F.3 we bound the gradient of $c(X)_{i_0,j_0}$

**Assumption F.1** (Bounded parameters, formal version of Assumption 4.1). *Let $W, V, X, B$ be defined as in Section A.2,*

- *Let $R$ be some fixed constant satisfies $R > 1$*

- *We have $\|W\| \leq R$, $\|V\| \leq R$, $\|X\| \leq R$ where $\|\cdot\|$ is the matrix spectral norm*

- *We have $b_{i,j} \leq R^2$*

### F.1 BOUNDS FOR BASIC FUNCTIONS

**Lemma F.2.** *Under Assumption F.1, for all $i_0 \in [n], j_0 \in [d]$, we have following bounds:*

- *Part 1*
$$\|f(X)_{i_0}\|_2 \leq 1$$

- *Part 2*
$$\|h(X)_{i_0}\|_2 \leq R^2$$

- *Part 3*
$$|c(X)_{i_0,j_0}| \leq 2R^2$$

- *Part 4*
$$\|x^\top W_{*,j_0}\|_2 \leq R^2$$

- *Part 5*
$$|w(X)_{i_0,j_0}| \leq R^2$$

- *Part 6*
$$|z(X)_{i_0,j_0}| \leq R^2$$

- *Part 7*
$$|s(X)_{i_0,j_0}| \leq R^2$$

*Proof.* **Proof of Part 1**

The proof is similar to Deng et al. (2023d), and hence is omitted here.

**Proof of Part 2**
$$\|h(X)_{j_0}\|_2 = \|X^\top V_{*,j_0}\|_2$$
$$\leq \|V\| \cdot \|X\|$$
$$\leq R^2$$

where the first step is by Definition A.7, the 2nd step is by basic algebra, the 3rd follows by Assumption F.1.

**Proof of Part 3**
$$|c(X)_{i_0,j_0}| = |\langle f(X)_{i_0}, h(X)_{j_0} \rangle - b_{i_0,j_0}|$$
$$\leq |\langle f(X)_{i_0}, h(X)_{j_0} \rangle| + |b_{i_0,j_0}|$$

$$\leq \|f(X)_{i_0}\|_2 \cdot \|h(X)_{j_0}\|_2 + |b_{i_0,j_0}|$$
$$\leq 2R^2$$

where the first step is by Definition A.8, the 2nd step uses triangle inequality, the 3rd step uses Cauchy-Schwartz inequality, the 4th step is by Assumption F.1 and **Part 2**.

**Proof of Part 4**

$$\|x^\top W_{*,j_0}\|_2 \leq \|x\| \cdot \|W\|$$
$$\leq R^2$$

where the first step is by basic algebra, the second is by Assumption F.1.

**Proof of Part 5**

$$|w(X)_{i_0,j_0}| = |\langle W_{j_0,*}, X_{*,i_0}|$$
$$\leq \|W_{j_0,*}\|_2 \cdot \|X_{*,i_0}\|_2$$
$$\leq R^2$$

where the first step is by the definition of $w(X)_{i_0,j_0}$, the 2nd step is Cauchy-Schwartz inequality, the 3rd step is by Assumption F.1.

**Proof of Part 6**

$$|z(X)_{i_0,j_0}| = |\langle f(X)_{i_0}, X^\top W_{*,j_0}\rangle|$$
$$\leq \|f(X)_{i_0}\|_2 \cdot \|X\| \cdot \|W_{*,j_0}\|$$
$$\leq R^2$$

where the first step is by the definition of $z(X)_{i_0,j_0}$, the 2nd step is Cauchy-Schwartz inequality, the 3rd step is by Assumption F.1.

**Proof of Part 7**

$$|s(X)_{i_0,j_0}| = |\langle f(X)_{i_0}, h(X)_{j_0}\rangle|$$
$$\leq \|f(X)_{i_0}\|_2 \cdot \|h(X)_{j_0}\|_2$$
$$\leq R^2$$

where the first step is by the definition of $s(X)_{i_0,j_0}$, the 2nd step is Cauchy-Schwartz inequality, the 3rd step is by **Part 1** and **Part 2**. □

### F.2  BOUNDS FOR GRADIENT OF $f(X)_{i_0}$

**Lemma F.3.** *Under following conditions*

- *Let $f(X)_{i_0}$ be defined as Definition A.6*

- *Assumption F.1 holds*

- *We use $\nabla f(X)_{i_0}$ to define a matrix that its $(j_0, i_1 \cdot j_1)$-th entry is*

$$\frac{\mathrm{d}f(X)_{i_0,j_0}}{\mathrm{d}x_{i_1,j_1}}$$

  *i.e., its $(i_1 \cdot j_1)$-th column is*

$$\frac{\mathrm{d}f(X)_{i_0}}{\mathrm{d}x_{i_1,j_1}}$$

*Then we have:*

- *Part 1: for all $i_0, i_1 \in [n], j_1 \in [d]$,*

$$\|\frac{\mathrm{d}f(X)_{i_0}}{\mathrm{d}x_{i_1,j_1}}\|_2 \leq 4R^2$$

- *Part 2:*

$$\|\nabla f(X)_{i_0}\|_F \le 4\sqrt{nd}R^2$$

*Proof.* **Proof of Part 1**

$$
\begin{aligned}
|\frac{\mathrm{d}f(X)_{i_0}}{\mathrm{d}x_{i_1,j_1}}| &= |-f(X)_{i_0} \cdot (f(X)_{i_0,i_0} \cdot \langle W_{j_1,*}, X_{*,i_0} \rangle + \langle f(X)_{i_0}, X^\top W_{*,j_1} \rangle) \\
&\quad + f(X)_{i_0} \circ (e_{i_0} \cdot \langle W_{j_1,*}, X_{*,i_0} \rangle + X^\top W_{*,j_1})| \\
&\le \|f(X)_{i_0}\|_2^2 \cdot |\langle W_{j_1,*}, X_{*,i_0} \rangle| + \|f(X)_{i_0}\|_2^2 \cdot \|X^\top W_{*,j_1}\| \\
&\quad + \|f(X)_{i_0}\|_2 \cdot |\langle W_{j_1,*}, X_{*,i_0} \rangle| + \|f(X)_{i_0}\|_2 \cdot \|X^\top W_{*,j_1})\|_2 \\
&\le 4R^2
\end{aligned}
$$

where the 1st step is by Lemma A.14, the 2nd step is by Fact A.1, the 3rd step is by Lemma F.2.

**Proof of Part 2**

$$
\begin{aligned}
\|\nabla f(X)_{i_0}\|_F &= (\sum_{i_1=1}^{n} \sum_{j_1=1}^{d} \|\frac{\mathrm{d}f(X)_{i_0}}{\mathrm{d}x_{i_1,j_1}}\|_2^2)^{\frac{1}{2}} \\
&\le (\sum_{i_1=1}^{n} \sum_{j_1=1}^{d} 16R^4)^{\frac{1}{2}} \\
&= 4\sqrt{nd}R^2
\end{aligned}
$$

where the first step is by the definition of $\nabla f(X)_{i_0}$, the 2nd step is by **Part 1**. $\qquad\square$

### F.3 BOUNDS FOR GRADIENT OF $c(X)_{i_0,j_0}$

**Lemma F.4.** *Under following conditions*

- *Let $c(X)_{i_0,j_0}$ be defined as Definition A.8*

- *Assumption F.1 holds*

- *We use $\nabla c(X)_{i_0,j_0}$ to denote the Hessian of $c(X)_{i_0,j_0}$ w.r.t. $\mathrm{vec}(X)$*

*Then we have:*

- *Part 1: for all $i_0, i_1 \in [n], j_1 \in [d]$,*

$$|\frac{c(X)_{i_0,j_0}}{\mathrm{d}x_{i_1,j_1}}|_2 \le 5R^4$$

- *Part 2:*

$$\|\nabla c(X)_{i_0,j_0}\|_2 \le 5\sqrt{nd}R^4$$

*Proof.* **Proof of part 1**

$$
\begin{aligned}
|\frac{\mathrm{d}c(X)_{i_0,j_0}}{\mathrm{d}x_{i_1,j_1}}| &= |C_1(X) + C_2(X) + C_3(X) + C_4(X) + C_5(X)| \\
&\le |C_1(X)| + |C_2(X)| + |C_3(X)| + |C_4(X)| + |C_5(X)| \\
&\le \|f(X)_{i_0}\|_2^2 \cdot \|h(X)_{j_0}\|_2 \cdot |w(X)_{i_0,j_0}| + \|f(X)_{i_0}\|_2 \cdot \|h(X)_{j_0}\|_2 \cdot |z(X)_{i_0,j_1}| \\
&\quad + \|f(X)_{i_0}\|_2 \cdot \|h(X)_{j_0}\|_2 \cdot |w(X)_{i_0,j_0}| \\
&\quad + \|f(X)_{i_0}\|_2 \cdot \|X\| \cdot \|W_{*,j_1}\|_2 \cdot \|h(X)_{j_0}\|_2 + \|f(X)_{i_0}\|_2 \cdot \|V\| \\
&\le R^4 + R^4 + R^4 + R^4 + R^2 \\
&\le 5R^4
\end{aligned}
$$

where the first step is by Lemma A.16, the 2nd step is by triangle inequality, the 3rd step is by Fact A.1, the 4th step is by Lemma F.2, the 5th step holds by $R > 1$.

**Proof of Part 2**

$$\|\nabla c(X)_{i_0,j_0}\|_2 = (\sum_{i_1=1}^{n} \sum_{j_1=1}^{d} \|\frac{\mathrm{d}c(X)_{i_0,j_0}}{\mathrm{d}x_{i_1,j_1}}\|_2^2)^{\frac{1}{2}}$$

$$\leq (\sum_{i_1=1}^{n} \sum_{j_1=1}^{d} 25R^8)^{\frac{1}{2}}$$

$$= 5\sqrt{nd}R^4$$

where the first step is by the definition of $\nabla f(X)_{i_0}$, the 2nd step is by **Part 1**.

$\square$

### F.4 BOUNDS FOR HESSIAN OF $c(X)_{i_0,j_0}$

**Lemma F.5.** *Under following conditions*

- *Let $c(X)_{i_0,j_0}$ be defined as Definition A.8*

- *Assumption F.1 (Bounded parameter) holds*

- *Let $B_i(X)$ be defined as in Definition D.6*

*we have*

- *Part 1: For all $i_0 = i_1 = i_2 \in [n]$, we have*

$$\|H_1(X)^{(i_0,i_0)}\| \leq 23R^6 + R^5 + 12R^3$$

- *Part 2: For all $i_0 = i_1 \neq i_2 \in [n]$, we have*

$$\|H_2(X)^{(i_0,i_2)}\| \leq 11R^6 + 6R^3$$

- *Part 3: For all $i_0 = i_2 \neq i_1 \in [n]$, we have*

$$\|H_3(X)^{(i_1,i_0)}\| \leq 11R^6 + 6R^3$$

- *Part 4: For all $i_0 \neq i_1 = i_2 \in [n]$, we have*

$$\|H_4(X)^{(i_1,i_1)}\| \leq 5R^6 + 4R^3$$

- *Part 5: For all $i_0 \neq i_1, i_0 \neq i_2, i_1 \neq i_2 \in [n]$, we have*

$$\|H_5(X)^{(i_1,i_2)}\| \leq 4R^6 + 2R^3$$

*Proof.* The proof is similar to Lemma F.4 and hence omit. $\square$

## G LIPSCHITZ OF HESSIAN

In Section G.1 we provide tools and facts. In Sections G.2, G.3, G.4, G.7, G.6, G.7 and G.8 we provide proof of lipschitz property of several important terms. And finally in Section G.9 we provide the proof for Lipschitz property of gradient of $L(X)$. In Section G.10 we provide proof for Lipschitz property of Hessian of $L(X)$.

## G.1 FACTS AND TOOLS

In this section, we introduce 2 tools for effectively calculate the Lipschitz for Hessian.

**Fact G.1** (Mean value theorem for vector function, Fact 34 in Deng et al. (2023d)). *Under following conditions,*

- *Let $x, y \in C \subset \mathbb{R}^n$ where $C$ is an open convex domain*

- *Let $g(x) : C \to \mathbb{R}^n$ be a differentiable vector function on $C$*

- *Let $\|g'(a)\|_F \leq M$ for all $a \in C$, where $g'(a)$ denotes a matrix which its $(i, j)$-th term is $\frac{dg(a)_j}{da_i}$*

*then we have*

$$\|g(y) - g(x)\|_2 \leq M\|y - x\|_2$$

**Fact G.2** (Lipschitz for product of functions). *Under following conditions*

- *Let $\{f_i(x)\}_{i=1}^n$ be a sequence of function with same domain and range*

- *For each $i \in [n]$ we have*

  - *$f_i(x)$ is bounded: $\forall x, \|f_i(x)\| \leq M_i$ with $M_i \geq 1$*
  - *$f_i(x)$ is Lipschitz continuous: $\forall x, y, \|f_i(x) - f_i(y)\| \leq L_i\|x - y\|$*

*Then we have*

$$\|\prod_{i=1}^n f_i(x) - \prod_{i=1}^n f_i(y)\| \leq 2^{n-1} \cdot \max_{i \in [n]}\{L_i\} \cdot (\prod_{i=1}^n M_i) \cdot \|x - y\|$$

*Proof.* We prove it by mathematical induction. The case that $i = 1$ obviously.

Now assume the case holds for $i = k$. Consider $i = k + 1$, we have.

$$\|\prod_{i=1}^{k+1} f_i(x) - \prod_{i=1}^{k+1} f_i(y)\|$$

$$\leq \|\prod_{i=1}^{k+1} f_i(x) - f_{k+1}(x) \cdot \prod_{i=1}^k f_i(y)\| + \|f_{k+1}(x) \cdot \prod_{i=1}^k f_i(y) - \prod_{i=1}^{k+1} f_i(y)\|$$

$$\leq \|f_{k+1}(x)\| \cdot \|\prod_{i=1}^k f_i(x) - \prod_{i=1}^k f_i(y)\| + \|f_{k+1}(x) - f_{k+1}(y)\| \cdot \|\prod_{i=1}^k f_i(y) - \prod_{i=1}^k f_i(y)\|$$

$$\leq M_{k+1} \cdot \|\prod_{i=1}^k f_i(x) - \prod_{i=1}^k f_i(y)\| + (\prod_{i=1}^k M_i) \cdot \|f_{k+1}(x) - f_{k+1}(y)\|$$

$$\leq 2^{k-1}(\prod_{i=1}^{k+1} M_i) \cdot \max_{i \in [k]}\{L_i\}\|x - y\| + (\prod_{i=1}^k M_i) \cdot \|f_{k+1}(x) - f_{k+1}(y)\|$$

$$\leq 2^{k-1}(\prod_{i=1}^{k+1} M_i) \cdot \max_{i \in [k]}\{L_i\}\|x - y\| + (\prod_{i=1}^k M_i) \cdot L_{k+1}\|x - y\|$$

$$\leq 2^{k-1}(\prod_{i=1}^{k+1} M_i) \cdot \max_{i \in [k]}\{L_i\}\|x - y\| + (\prod_{i=1}^{k+1} M_i) \cdot L_{k+1}\|x - y\|$$

$$\leq 2^k(\prod_{i=1}^{k+1} M_i) \cdot \max_{i \in [k+1]}\{L_i\}\|x - y\|$$

where the first step is by triangle inequality, the 2nd step is by property of norm, the 3rd step is by upper bound of functions, the 4th step is by induction hypothesis, the 5th step is by Lipschitz of $f_{k+1}(x)$, the 6th step is by $M_{k+1} \geq 1$, the 7th step is a rearrangement.

Since the claim holds for $i = k + 1$, we prove the desired result. $\qquad\square$

## G.2 LIPSCHITZ FOR $f(X)_{i_0}$

**Definition G.3** (Notation of norm). *For writing efficiency, we use $\|X - Y\|$ to denote $\|\operatorname{vec}(X) - \operatorname{vec}(Y)\|_2$, which is equivalent to $\|X - Y\|_F$.*

**Lemma G.4.** *Under following conditions*

- *Assumption F.1 holds*

- *Let $f(X)_{i_0}$ be defined as Definition A.6*

*For $X, Y \in \mathbb{R}^{d \times n}$, we have*

$$\|f(X)_{i_0} - f(Y)_{i_0}\|_2 \leq 4\sqrt{nd}R^2 \cdot \|X - Y\|$$

*Proof.*

$$\begin{aligned}
\|f(X)_{i_0} - f(Y)_{i_0}\|_2 &\leq \|\nabla f(X)_{i_0}\|_F \cdot \|X - Y\| \\
&\leq 4\sqrt{nd}R^2 \cdot \|X - Y\|
\end{aligned}$$

where the first step is given by Mean Value Theorem (Lemma G.1) and the 2nd step is due to upper bound for gradient of $f(X)_{i_0}$ (Lemma F.3). $\qquad\square$

## G.3 LIPSCHITZ FOR $c(X)_{i_0,j_0}$

**Lemma G.5.** *Under following conditions*

- *Assumption F.1 holds*

- *Let $c(X)_{i_0,j_0}$ be defined as Definition A.8*

*For $X, Y \in \mathbb{R}^{d \times n}$, we have*

$$|c(X)_{i_0,j_0} - c(Y)_{i_0,j_0}| \leq 5\sqrt{nd}R^4 \cdot \|X - Y\|$$

*Proof.*

$$\begin{aligned}
|c(X)_{i_0,j_0} - c(Y)_{i_0,j_0}| &\leq \|\nabla c(X)_{i_0,j_0}\|_2 \cdot \|X - Y\| \\
&\leq 5\sqrt{nd}R^4 \cdot \|X - Y\|
\end{aligned}$$

where the first step is given by Mean Value Theorem (Lemma G.1) and the 2nd step is due to upper bound for gradient of $c(X)_{i_0,j_0}$ (Lemma F.4). $\qquad\square$

## G.4 LIPSCHITZ FOR $h(X)_{j_0}$

**Lemma G.6.** *Under following conditions*

- *Assumption F.1 holds*

- *Let $h(X)_{j_0}$ be defined as Definition A.7*

*For $X, Y \in \mathbb{R}^{d \times n}$, we have*

$$\|h(X)_{j_0} - h(Y)_{j_0}\|_2 \leq R\|X - Y\|$$

*Proof.*

$$\|h(X)_{j_0} - h(Y)_{j_0}\| = \|V_{*,j_0}\|_2 \cdot \|X - Y\|$$
$$\leq R \cdot \|X - Y\|$$

where the first step is from the definition of $h(X)_{j_0}$ (see Definition A.7), the 2nd step is by Assumption F.1. □

### G.5 LIPSCHITZ FOR $w(X)_{i_0,j_0}$

**Lemma G.7.** *Under following conditions*

- *Assumption F.1 holds*

*For $X, Y \in \mathbb{R}^{d \times n}$, we have*

$$|w(X)_{i_0,j_0} - w(Y)_{i_0,j_0}| \leq R\|X - Y\|$$

*Proof.*

$$|w(X)_{i_0,j_0} - w(Y)_{i_0,j_0}| = |\langle W_{j_0,*}, X_{*,i_0} - Y_{*,i_0}\rangle|$$
$$\leq \|W_{j_0,*}\|_2 \cdot \|X - Y\|$$
$$\leq R \cdot \|X - Y\|$$

where the first step is from the definition of $w(X)_{i_0,j_0}$, the 2nd step is by Fact A.1, the 3rd step holds since Assumption F.1. □

### G.6 LIPSCHITZ FOR $z(X)_{i_0,j_0}$

**Lemma G.8.** *Under following conditions*

- *Assumption F.1 holds*

*For $X, Y \in \mathbb{R}^{d \times n}$, we have*

$$|z(X)_{i_0,j_0} - z(Y)_{i_0,j_0}| \leq 5\sqrt{nd}R^4 \cdot \|X - Y\|$$

*Proof.*

$$|z(X)_{i_0,j_0} - z(Y)_{i_0,j_0}| = |\langle f(X)_{i_0}, X^\top W_{*,j_0}\rangle - \langle f(Y)_{i_0}, Y^\top W_{*,j_0}\rangle|$$
$$\leq |\langle f(X)_{i_0}, X^\top W_{*,j_0}\rangle - \langle f(X)_{i_0}, Y^\top W_{*,j_0}\rangle|$$
$$\quad + |\langle f(X)_{i_0}, Y^\top W_{*,j_0}\rangle - \langle f(Y)_{i_0}, Y^\top W_{*,j_0}\rangle|$$
$$\leq \|f(X)_{i_0}\|_2 \cdot \|X - Y\| \cdot \|W_{*,j_0}\|_2 + \|f(X)_{i_0} - f(Y)_{i_0}\| \cdot \|Y\| \cdot \|W_{*,j_0}\|$$
$$\leq R \cdot \|X - Y\| + R^2\|f(X)_{i_0} - f(Y)_{i_0}\|$$
$$\leq 5\sqrt{nd}R^4 \cdot \|X - Y\|$$

where the first step is from the definition of $w(X)_{i_0,j_0}$, the 2nd step is by Fact A.1, the 3rd step holds since Assumption F.1, the 4th step uses Lemma G.4. □

### G.7 LIPSCHITZ FOR FIRST ORDER DERIVATIVE OF $c(X)_{i_0,j_0}$

**Lemma G.9.** *Under following conditions*

- *Assumption F.1 holds*

- *Let $c(X)_{i_0,j_0}$ be defined as Definition A.8*

*For $X, Y \in \mathbb{R}^{d \times n}$, we have*

$$|\frac{c(X)_{i_0,j_0}}{\mathrm{d}x_{i_1,j_1}} - \frac{c(Y)_{i_0,j_0}}{\mathrm{d}y_{i_1,j_1}}| \leq O(\sqrt{nd}R^6) \cdot \|X - Y\|$$

*Proof.* Recall $C_i(X)$ defined in Lemma A.16. The Lipschitz constant of $\frac{c(X)_{i_0,j_0}}{\mathrm{d}x_{i_1,j_1}}$ is bounded the summation of that of $C_i(X)$. We only present the proof for Lipschitz for $C_1(X)$ here.

Notice that

$$C_1(X) := -s(X)_{i_0,j_0} \cdot f(X)_{i_0,i_0} \cdot w(X)_{i_0,j_1}$$

By upper bound and lipschitz constant for basic functions, we have

- $|s(X)_{i_0,j_0}| \leq R^2$

- $|f(X)_{i_0,i_0}| \leq 1$

- $|w(X)_{i_0,j_1}| \leq R^2$

- $\max_{f \in \{s(X)_{i_0,j_0}, f(X)_{i_0,i_0}, w(X)_{i_0,j_1}\}} \{\mathrm{Lipschitz}(f)\} = 4\sqrt{nd}R^2$

- $n = 3$

By Fact G.2.

$$
\begin{aligned}
|C_1(X) - C_1(Y)| &\leq 2^{n-1} \cdot \max_{i \in [n]} \{L_i\} \cdot (\prod_{i=1}^{n} M_i) \cdot \|X - Y\| \\
&= 4 \cdot 4\sqrt{nd}R^2 \cdot R^4 \cdot \|X - Y\| \\
&= 16\sqrt{nd}R^6 \cdot \|X - Y\|
\end{aligned}
$$

$\square$

### G.8 Lipschitz for second order derivative of $c(X)_{i_0,j_0}$

**Lemma G.10.** *Under following conditions*

- *Assumption F.1 holds*

- *Let $c(X)_{i_0,j_0}$ be defined as Definition A.8*

*For $X, Y \in \mathbb{R}^{d \times n}$, we have*

$$|\frac{c(X)_{i_0,j_0}}{\mathrm{d}x_{i_1,j_1} x_{i_2,j_2}} - \frac{c(Y)_{i_0,j_0}}{\mathrm{d}y_{i_1,j_1} y_{i_2,j_2}}| \leq O(\sqrt{nd}R^8) \cdot \|X - Y\|$$

*Proof.* The proof is similar to Lemma G.9 and hence omit. Notice that the upper bound for $\frac{c(X)_{i_0,j_0}}{\mathrm{d}x_{i_1,j_1} x_{i_2,j_2}}$ is given by Lemma F.5. $\square$

### G.9 Lipschitz for gradient of $L(X)$

**Lemma G.11.** *Under following conditions*

- *Assumption F.1 holds*

- *Let $c(X)_{i_0,j_0}$ be defined as Definition A.8*

*For $X, Y \in \mathbb{R}^{d \times n}$, we have*

$$\|\nabla^2 L(X) - \nabla^2 L(Y)\| \leq O(n^{1.5}d^{1.5}R^{10}) \cdot \|X - Y\|$$

*Proof.* We have calculated the gradient of $L(X)$ in Lemma A.17:

$$\frac{\mathrm{d}L(X)}{\mathrm{d}x_{i_1,j_1}} = \sum_{i_0=1}^{n} \sum_{j_0=1}^{d} c(X)_{i_0,j_0} \cdot \frac{\mathrm{d}c(X)_{i_0,j_0}}{\mathrm{d}x_{i_1,j_1}}$$

We can use the proof in Lemma G.9 to generate a Lipschitz bound for he gradient of $L(X)$. Notice that the Lipschitz of $c(X)_{i_0,j_0}$ is given in Lemma G.5 and the Lipschitz of $\frac{\mathrm{d}c(X)_{i_0,j_0}}{\mathrm{d}x_{i_1,j_1}}$ is given in Lemma G.9. $\qquad\square$

## G.10 LIPSCHITZ FOR HESSIAN OF $L(X)$

**Lemma G.12.** *Under following conditions*

- *Assumption F.1 holds*

- *Let $c(X)_{i_0,j_0}$ be defined as Definition A.8*

*For $X, Y \in \mathbb{R}^{d \times n}$, we have*

$$\|\nabla^2 L(X) - \nabla^2 L(Y)\| \le O(n^{3.5} d^{3.5} R^{10}) \cdot \|X - Y\|$$

*Proof.* Recall that

$$\frac{\mathrm{d}L(X)}{\mathrm{d}x_{i_1,j_1} x_{i_2,j_2}} = \sum_{i_0=1}^{n} \sum_{j_0=1}^{d} \frac{\mathrm{d}c(X)_{i_0,j_0}}{\mathrm{d}x_{i_1,j_1}} \cdot \frac{\mathrm{d}c(X)_{i_0,j_0}}{\mathrm{d}x_{i_2,j_2}} + c(X)_{i_0,j_0} \cdot \frac{\mathrm{d}c(X)_{i_0,j_0}}{\mathrm{d}x_{i_1,j_1} x_{i_2,j_2}}$$

$$= \sum_{i_0=1}^{n} \sum_{j_0=1}^{d} U_1(X) + U_2(X)$$

For the first item $U_1(X)$, we have

$$|U_1(X) - U_1(Y)| = |\frac{\mathrm{d}c(X)_{i_0,j_0}}{\mathrm{d}x_{i_1,j_1}} \cdot \frac{\mathrm{d}c(X)_{i_0,j_0}}{\mathrm{d}x_{i_1,j_2}} - \frac{\mathrm{d}c(Y)_{i_0,j_0}}{\mathrm{d}x_{i_1,j_1}} \cdot \frac{\mathrm{d}c(Y)_{i_0,j_0}}{\mathrm{d}y_{i_1,j_2}}|$$

$$\le |\frac{\mathrm{d}c(X)_{i_0,j_0}}{\mathrm{d}x_{i_1,j_1}}| \cdot |\frac{\mathrm{d}c(X)_{i_0,j_0}}{\mathrm{d}x_{i_1,j_2}} - \frac{\mathrm{d}c(Y)_{i_0,j_0}}{\mathrm{d}y_{i_2,j_2}}|$$

$$+ |\frac{\mathrm{d}c(X)_{i_0,j_0}}{\mathrm{d}x_{i_1,j_1}} \cdot - \frac{\mathrm{d}c(Y)_{i_0,j_0}}{\mathrm{d}y_{i_1,j_1}}| \cdot |\frac{\mathrm{d}c(Y)_{i_0,j_0}}{\mathrm{d}y_{i_2,j_2}}|$$

$$\le 10R^4 \cdot |\frac{\mathrm{d}c(X)_{i_0,j_0}}{\mathrm{d}x_{i_1,j_1}} \cdot - \frac{\mathrm{d}c(Y)_{i_0,j_0}}{\mathrm{d}y_{i_1,j_1}}|$$

$$\le O(\sqrt{nd} R^{10}) \cdot \|X - Y\|$$

where the 2nd step is by triangle inequality, the 3rd step is by Lemma F.4, the 4th step uses Lemma G.9.

For the 2nd item $U_2(X)$, we have

$$|U_2(X) - U_2(Y)| = |c(X)_{i_0,j_0} \cdot \frac{\mathrm{d}c(X)_{i_0,j_0}}{\mathrm{d}x_{i_1,j_1} x_{i_2,j_2}} - c(Y)_{i_0,j_0} \cdot \frac{\mathrm{d}c(Y)_{i_0,j_0}}{\mathrm{d}y_{i_1,j_1} y_{i_2,j_2}}|$$

$$\le |c(X)_{i_0,j_0}| \cdot |\frac{\mathrm{d}c(X)_{i_0,j_0}}{\mathrm{d}x_{i_1,j_1} x_{i_2,j_2}} - \frac{\mathrm{d}c(Y)_{i_0,j_0}}{\mathrm{d}y_{i_1,j_1} y_{i_2,j_2}}|$$

$$+ |c(X)_{i_0,j_0} - c(Y)_{i_0,j_0}| \cdot |\frac{\mathrm{d}c(Y)_{i_0,j_0}}{\mathrm{d}y_{i_1,j_1} y_{i_2,j_2}}|$$

$$\le 2R^2 \cdot |\frac{\mathrm{d}c(X)_{i_0,j_0}}{\mathrm{d}x_{i_1,j_1} x_{i_2,j_2}} - \frac{\mathrm{d}c(Y)_{i_0,j_0}}{\mathrm{d}y_{i_1,j_1} y_{i_2,j_2}}|$$

$$+ |c(X)_{i_0,j_0} - c(Y)_{i_0,j_0}| \cdot |\frac{\mathrm{d}c(Y)_{i_0,j_0}}{\mathrm{d}y_{i_1,j_1} y_{i_2,j_2}}|$$

$$\le 2R^2 \cdot |\frac{\mathrm{d}c(X)_{i_0,j_0}}{\mathrm{d}x_{i_1,j_1} x_{i_2,j_2}} - \frac{\mathrm{d}c(Y)_{i_0,j_0}}{\mathrm{d}y_{i_1,j_1} y_{i_2,j_2}}| + 5\sqrt{nd} R^4 \cdot \|X - Y\| \cdot |\frac{\mathrm{d}c(Y)_{i_0,j_0}}{\mathrm{d}y_{i_1,j_1} y_{i_2,j_2}}|$$

$$\le O(\sqrt{nd} R^{10}) \cdot \|X - Y\| + 5\sqrt{nd} R^4 \cdot \|X - Y\| \cdot |\frac{\mathrm{d}c(Y)_{i_0,j_0}}{\mathrm{d}y_{i_1,j_1} y_{i_2,j_2}}|$$

$$\leq O(\sqrt{nd}R^{10}) \cdot \|X - Y\|$$

where the 2nd step is by triangle inequality, the 3rd step uses Lemma F.2, the 4th step uses Lemma G.5, the 5th step uses Lemma G.10, the last step uses Lemma F.5.

Combining the above 2 items, we have

$$|\frac{\mathrm{d}L(X)}{\mathrm{d}x_{i_1,j_1}x_{i_2,j_2}} - \frac{\mathrm{d}L(Y)}{\mathrm{d}y_{i_1,j_1}y_{i_2,j_2}}| \leq O(n^{1.5}d^{1.5}R^{10}) \cdot \|X - Y\|$$

Then, we have

$$\begin{aligned}
\|\nabla^2 L(X) - \nabla^2 L(Y)\| &\leq \|\nabla^2 L(X) - \nabla^2 L(Y)\|_F \\
&\leq n^2 d^2 \cdot O(n^{1.5}d^{1.5}R^{10}\|X - Y\| \\
&= O(n^{3.5}d^{3.5}R^{10}) \cdot \|X - Y\|
\end{aligned}$$

where the 1st step is by matrix calculus, the 2nd is by the lipschitz for each entry of $\nabla^2 L(X)$. $\square$

## H    STRONGLY CONVEXITY

In this section, we provide proof for PSD bounds for the Hessian of Loss function.

### H.1    PSD BOUNDS FOR HESSIAN OF $c(X)_{i_0,j_0}$

**Lemma H.1** (PSD bounds for $\nabla^2 c(X)_{i_0,j_0}$). *Under following conditions,*

- *Let $c_{i_0,j_0}$ be defined as in Definition A.8*

- *Let Assumption F.1 be satisfied*

*For all $i_0 \in [n], j_0 \in [d]$, we have*

$$-36R^6 \cdot \mathbf{I}_{nd} \preceq \nabla^2 c(X)_{i_0,j_0} \preceq 36R^6 \cdot \mathbf{I}_{nd}$$

*Proof.* We prove this statement by the definition of PSD. Let $p \in \mathbb{R}^{n \times d}$ be a vector. Let $i \in [n]$, we use $p_i \in \mathbb{R}^d$ to denote the vector formed by the $(i-1) \cdot n + 1$-th term to the $i \cdot n$-th term of vector $p$.

Then, we have

$$\begin{aligned}
|p^\top \nabla^2 c(X)_{i_0,j_0} p| = |p_{i_0}^\top H_1(X)^{i_0,i_0} p_{i_0} &+ \sum_{i \in [n] \backslash \{i_0\}} p_{i_0}^\top H_2(X)^{(i_0,i)} p_i \\
&+ \sum_{i \in [n] \backslash \{i_0\}} p_i^\top H_3(X)^{(i,i_0)} p_{i_0} + \sum_{i \in [n] \backslash \{i_0\}} p_i^\top H_4(X)^{(i,i)} p_i \\
&+ \sum_{i_1 \in [n] \backslash \{i_0\}} \sum_{i_2 \in [n] \backslash \{i_0\}} p_{i_1}^\top H_5(X)^{(i_1,i_2)} p_{i_2}| \\
&\leq \max_{i \in [5]} \|H_i(X)\| \cdot \sum_{i_1 \in [n]} \sum_{i_2 \in [n]} p_{i_1}^\top p_{i_2} \\
&\leq \max_{i \in [5]} \|H_i(X)\| \cdot p^\top p \\
&\leq 36R^6 \cdot p^\top p
\end{aligned}$$

where the 1st step is by the formulation of $\nabla^2 c(X)_{i_0,j_0}$ (see Definition D.3), the 2nd and 3rd steps are from simple algebra, the 4th step uses Lemma F.5. $\square$

### H.2    PSD BOUNDS FOR HESSIAN OF LOSS

**Lemma H.2** (PSD bound for $\nabla^2 L(X)$). *Under following conditions,*

- *Let $L(X)$ be defined as in Definition A.9*

- *Let Assumption F.1 be satisfied*

*we have*

$$\nabla^2 L(X) \succeq -O(ndR^8) \cdot \mathbf{I}_{nd}$$

*Proof.* Recall in Lemma E.2, we have

$$\nabla^2 L(X) = \sum_{i_0=1}^{n} \sum_{j_0=1}^{d} \nabla c(X)_{i_0,j_0} \cdot \nabla c(X)_{i_0,j_0}^{\top} + c(X)_{i_0,j_0} \cdot \nabla^2 c(X)_{i_0,j_0} \tag{2}$$

Notice that the first term is PSD, so we omit it.

By Lemma F.2, we have

$$|c(X)_{i_0,j_0}| \le 2R^2$$

Therefore, we have

$$\nabla^2 c(X)_{i_0,j_0} \succeq -72R^8 \cdot \mathbf{I}_{nd}$$
$$i.e., \nabla^2 L(X) \succeq -72ndR^8 \cdot \mathbf{I}_{nd}$$

where the first line is by Lemma H.1 and the 2nd line is given by Eq. (2).

$\square$

## I CONVERGENCE ANALYSIS

In this section, we give the convergence analysis of the gradient-based (see Section I.1 and Hessian-based method (see Section I.2) to conduct inverse attack. We utilize the Lipschitz and strongly-convexity properties proved in previous sections.

### I.1 GRADIENT METHOD

We first state a canonical result for the convergence gradient-descent method under Lipschitz smoothness and strongly-convexity.

**Theorem I.1** (Gradient descent). *Let the following conditions hold*

- *Let $f(x)$ be a convex and twice-differentiable function on $\mathbb{R}^n$*

- *Let $\nabla f(x)$ have Lipschitz constant $L$:*
$$\|\nabla f(x) - \nabla f(y)\|_2 \le L\|x - y\|_2 \quad \forall x, y \in \mathbb{R}^n$$

- *Let $f(x)$ be strongly convex with factor $m$:*
$$\nabla^2 f(x) \succeq m\mathbf{I}_n$$

- *$f(x)$ reaches its minimum (denoted as $f^*$) at some point $x^*$*

*Then, the gradient-descent algorithm with fixed step size $t < \frac{2}{m+L}$ satisfies*

$$\|x_k - x^*\|_2 < (1 - \frac{m}{L})^{k/2} \cdot \|x_0 - x^*\|_2^2$$

*where $x_k$ is the update in $k$-th iteration.*

*In particular, it takes $O(\frac{L}{m} \cdot \log(|x_0 - x^*|_2/\epsilon))$ to find a $\epsilon$-optimal solution.*

Now, we use the above theorem to show the convergence of our regression problem.

**Theorem I.2** (Formal version of Theorem 4.3 )**.** *We assume our model satisfies the following conditions*

- *Bounded parameters: there exists $R > 1$ such that*
  - $\|W\|_F \leq R$, $\|V\|_F \leq R$
  - $\|X\|_F \leq R$
  - $\forall i \in [n], j \in [d], |b_{i,j}| \leq R$ *where $b_{i,j}$ denotes the $i, j$-th entry of $B$*

- *Regularization: we consider the following problem:*
$$\min_{X \in \mathbb{R}^{n \times d}} \|D(X)^{-1} \exp(X^\top W X) X^\top V - B\|_F^2$$
$$+ \gamma \cdot \|\operatorname{vec}(X)\|_2^2$$

*Then, for any accuracy parameter $\epsilon \in (0, 0.1)$, a gradient-descent algorithm can be employed to recover the initial data. The algorithm uses*
$$T = O(\operatorname{poly}(n, d, R) \cdot \log(|X_0 - X^*|_F / \epsilon))$$

*iterations, it outputs a matrix $\widetilde{X} \in \mathbb{R}^{d \times n}$ satisfying*
$$\|\widetilde{X} - X^*\|_F \leq \epsilon$$

*The execution time for each iteration is $\operatorname{poly}(n, d)$.*

*Proof.* Choosing $\gamma \geq O(ndR^8)$, by Lemma H.2, we have our regression problem being strongly convex with factor $O(ndR^8)$. Notice that, we proved in Lemma G.11 that the gradient of our loss function is $O(n^{1.5} d^{1.5} R^{10})$-Lipschitz continuous. Applying Theorem I.1 with $L = O(n^{1.5} d^{1.5} R^{10})$ and $m = O(ndR^8)$, we have the result in this theorem.

The execution time for each iteration is the matrix-multiplication time. $\square$

## I.2 Hessian Method

**Theorem I.3** (Formal version of Theorem 4.4, Main Result)**.** *We assume our model satisfies the following conditions*

- *Bounded parameters: there exists $R > 1$ such that*
  - $\|W\|_F \leq R$, $\|V\|_F \leq R$
  - $\|X\|_F \leq R$
  - $\forall i \in [n], j \in [d], |b_{i,j}| \leq R$ *where $b_{i,j}$ denotes the $i, j$-th entry of $B$*

- *Regularization: we consider the following problem:*
$$\min_{X \in \mathbb{R}^{n \times d}} \|D(X)^{-1} \exp(X^\top W X) X^\top V - B\|_F^2$$
$$+ \gamma \cdot \|\operatorname{vec}(X)\|_2^2$$

- *Good initial point: We choose an initial point $X_0$ such that*
$$M \cdot \|X_0 - X^*\|_F \leq O(ndR^8),$$
*where $M = O(n^3 d^3 R^{10})$.*

*Then, for any accuracy parameter $\epsilon \in (0, 0.1)$ and any failure probability $\delta \in (0, 0.1)$, an algorithm based on the Newton method can be employed to recover the initial data. The result of this algorithm guarantee within*
$$T = O(\log(|X_0 - X^*|_F / \epsilon))$$

*iterations, it outputs a matrix $\widetilde{X} \in \mathbb{R}^{d \times n}$ satisfying*
$$\|\widetilde{X} - X^*\|_F \leq \epsilon$$

*with a probability of at least $1 - \delta$. The execution time for each iteration is $\operatorname{poly}(n, d, \log(1/\delta))$.*

*Proof.* Choosing $\gamma \geq O(ndR^8)$, by Lemma H.2, we have the PD property of Hessian.

By Lemma G.12, we have the Lipschitz property of Hessian.

Since $M$ is bounded (in the condition of Theorem), then by iterative shrinking lemma (see Lemma 6.9 in Li et al. (2023c) as an example), we prove the convergence. $\square$

## J  SUPPLEMENTARY EXPERIMENTAL DETAILS

Here, we give the experimental details for our experiment as follows.

- Learning rate for fine-tuning: $\eta = 0.0001$ (for best effort).
- Learning rate for attack: $\eta = 0.001$ (default).
- Adam hyper-parameter $\beta_1 = 0.9$ (default).
- Adam hyper-parameter $\beta_2 = 0.999$ (default).
- Adam hyper-parameter $\epsilon = 1 \times 10^{-8}$ (default).
- Fine-tuning steps: 8000.
- Platform: PyTorch Paszke et al. (2019) and Huggingface Wolf et al. (2019).
- GPU device information: 1 RTX 4090 GPUs.
- Number of fine-tuning epochs 30.
- Batch size: 32 (for best effort).
- Quantization: fp16.

