# OpenReview forum: "Unmasking Transformers: A Theoretical Approach to Data Recovery via Attention Weights"
_ICLR.cc/2025/Workshop/BuildingTrust — Submitted to BuildingTrust_

### Official Review · Reviewer_yG8u · 2025-02-28

**Rating:** 6
**Confidence:** 5

**Review:**

The paper proposes an inversion attack to recover the input of a transformer model with given attention layers and outputs. The topic is of certain importance and interesting. However, the reviewer has some concerns.

1. First of all, the authors should make it clear the model output $B$ is the logits of the transformer right after Softmax because the top-k selection process and de-tokenizing are non-differentiate. The gradient-wise inverse won't work if the outputs are token IDs or words if for LLMs (e.g. GPT2 in the paper). However, it is a scarce case that outputs will be available to an adversary.
2. Continuing on point one, it is quite confusing why the weights of attention layers are needed if the logits are available as the mathematic derivation should still stand even if starting from the final logits.
3. The reason why the weights of attention layers are available while the weights of other layers such as linear MLP layers are not available is missing.
4. One common case that both outputs and weights will be leaked to attackers is federated learning, which can be mentioned in the paper

Despite these, the paper still elaborates how the gradient flow goes over during the inverse attack (gradient matching) based and raises an attention to the privatizing weights and logits of a transformer model.

---

### Official Review · Reviewer_4WdU · 2025-03-01
**Important problem, but the method and the paper has several issues**

**Rating:** 3
**Confidence:** 4

**Review:**

The paper frames the data recovery problem as an inverse regression on the transformer’s attention mechanism. It assumes the model’s weights and attention matrices for a given input are known and seek to recover the input based on this information, by directly optimizing the input vectors.

I have several concerns with the paper:

Methodology and Experimental Setup:

1. Small Fine-Tuning Dataset

The paper fine-tunes a GPT-2 model on only “hundreds” of data points, which is a trivially small scale by GPT-2 standards. This severely limits the credibility of the experiments: with such a small dataset, the model might simply be memorizing most or all of the training samples. Consequently, it is unclear whether the observed data-recovery phenomenon is due to a genuine architectural vulnerability or just the result of a grossly overfitted, memorizing model. If the latter, then the practical implications for larger-scale language models—trained on millions or billions of tokens—remain unexamined.


2. Number of Tokens Recovered

The paper showcases only one example where two masked tokens are reconstructed. Although this example confirms the feasibility of recovering a short phrase, there is no systematic evaluation on how many tokens can be recovered in practice, or how model accuracy might degrade as the number of targeted tokens grows. More studies, possibly with an increasing number of masked tokens, would be needed to determine whether the method scales or quickly fails as the masked region lengthens.


3. Performance Post Fine-Tuning

There is no discussion of whether the model maintains or loses its capabilities on the original GPT-2 tasks after this specialized fine-tuning. If the model is merely overfitted to producing memorized text strings, that might degrade other performance metrics. Without assessing generalization or broader task performance, we cannot be sure whether this “attack” is robust or simply overwriting the model with new data in a way that undermines its overall usefulness.


4. Access to Non-Masked Training Data

The approach assumes that the attacker has access to the unmasked portion of the data, plus the model’s attention mechanisms. This is a fairly strong assumption - the attack is essentially a white-box attack. Many real-world settings may not expose partial ground truth or the full set of attention weights. Hence, the practicality of the threat model is doubtful. Under less privileged conditions, the method may be far less effective. Realistically, an attacker might not know partial ground-truth tokens. This assumption can inflate attack success rates.


5. Mapping Vectors to Tokens

The authors do not explain how the final optimized continuous vectors are mapped back to discrete tokens—whether they use a nearest-neighbor approach in embedding space, an argmax over vocabulary logits, or some other heuristic. Omitting these details leaves a gap: the success of data recovery might hinge on how effectively the recovered vectors are snapped to valid tokens.

$$----------------$$

Theoretical Oversights - Ignoring Residual and Feed-Forward Layers

The Hessian analysis focuses exclusively on the self-attention sub-layer and treats the architecture as though there are no additional non-linearities (i.e., feed-forward components), normalization layers, or skip connections. Real GPT-like architectures heavily rely on these other components, which introduce significant complexity, especially in the model’s gradient and Hessian structure. Omitting them may invalidate or oversimplify any claims about invertibility in actual GPT architectures.

$$----------------$$

Related Works

1. Connection to Memorization

Large language models have been known to memorize parts of their training data, especially if the data is unique or repeated. This paper’s data recovery approach essentially exploits that memorization phenomenon, yet the authors do not mention or position their work in the context of “LLM memorization.” Understanding how memorization emerges—and how it might be mitigated—would be a vital part of any serious discussion about data leakage. Failure to cite or discuss memorization literature is a significant gap.

2. Similarities to Existing Recovery Techniques

Recovering training data from model parameters, gradients, or partial outputs has been extensively studied, especially in computer vision. Leveraging attention weights is, in principle, not too different from leveraging CNN filters or other forms of latent representations. The paper does not sufficiently acknowledge these works, leaving its “novel” position questionable.

$$--------------------$$

Scalability and Real-World Feasibility

Because the experiments are conducted only on a small set of data points, it is unclear whether this method would work at scale for real GPT-2 style models trained on massive corpora. If the method solely exploits memorization from a tiny fine-tuned dataset, it may not extend to scenarios where the model’s parameters are shaped by vast and diverse training sets.


Given these severe concerns, I cannot recommend accepting the paper in its current form.

---

### Official Review · Reviewer_LxCb · 2025-03-03
**The paper presents a theoretically sound but impractically applicable model inversion attack for extracting training data from attention weights, lacking comparative evaluation against established baselines.**

**Rating:** 5
**Confidence:** 3

**Review:**

The authors introduce a model inversion technique that extracts confidential training data by analyzing attention weights alongside model outputs. Taking a theoretical perspective, this research demonstrates that adversaries can reconstruct private input data X through a process of loss function minimization when they have obtained the model's attention weights and output values.

Strength:
This research establishes a theoretical framework for reconstructing input data through analysis of attention weights and model outputs. The authors carefully examine the mathematical underpinnings of attention mechanisms to understand how sensitive information might be recovered from inputs, which helps people's understanding of the potential risk.

Weakness:
1. The experimental evaluation lacks comparison with established baselines from the membership inference attack domain, instead only reporting loss values and corresponding success rates. While many MIA benchmarks are known to suffer from distribution shift issues, including performance metrics on these benchmarks and comparisons with existing approaches would strengthen the paper's empirical validation.
2. The threat model assumes adversarial access to training data attention outputs but not the data itself, which appears to be an impractical assumption in real-world scenarios. This undermines the practical applicability of the proposed attack method.

---

### Decision · Program_Chairs · 2025-03-04

**Decision:**

Reject

**Comment:**

Reviewers have mentioned several concerns. In particular, the memorization one is significant.